



# IPSL-Perm-LandN: improving the IPSL Earth System Model to represent permafrost carbon-nitrogen interactions

Rémi Gaillard[1,2], Patricia Cadule[3], Philippe Peylin[4], Nicolas Vuichard[4], and Bertrand Guenet[1]

[1]Laboratoire de Géologie, Ecole Normale Supérieure, CNRS, Institut Pierre-Simon Laplace, Université Paris Sciences et Lettres, Paris, France
[2]Faculty of Environment, Science and Economy, University of Exeter, Exeter, UK
[3]Laboratoire de Météorologie Dynamique, Institut Pierre-Simon Laplace, CNRS, École Normale Supérieure, Université PSL, Sorbonne Université, École Polytechnique, Paris, France
[4]Laboratoire des Sciences du Climat et de l'Environnement, LSCE/IPSL, CEA-CNRS-UVSQ, Université Paris-Saclay, Gif-sur-Yvette, France

**Correspondence:** Rémi Gaillard (gaillard@geologie.ens.fr)

**Abstract.** Permafrost soils have the potential to release large amounts of soil carbon to the atmosphere under climate change. However, in the Sixth Coupled Model Intercomparison Project (CMIP6), only two Earth System Models (ESM) represented permafrost carbon, both sharing the same land surface model. This makes future permafrost carbon dynamics highly uncertain and underscores the urgent need to include permafrost carbon in ESMs to enable more reliable future projections of climate

change and remaining carbon budget estimates. Here, we present IPSL-Perm-LandN, an improved version of the Institut Pierre-Simon Laplace (IPSL) ESM (used for CMIP6) aiming at better representing high-latitude land ecosystems. The main developments are the inclusion of an explicit nitrogen cycle and of key permafrost physical and biogeochemical processes. The latent heat associated with soil water freeze/thaw is taken into account in the energy budget, as well as soil thermal insulation by soil organic matter and a surface organic layer (e.g. litter or moss). Soil organic carbon and nitrogen are vertically resolved

with a depth-dependent decomposition dynamics, a key feature for representing the effect of gradual permafrost thaw on soil biogeochemistry. Cryoturbation is represented as a diffusion process that buries organic matter in the deeper soil layers. Compared to the previous version of the model used for CMIP6, we show that the extent of the permafrost region has improved significantly and that the simulated active layer thickness in the Arctic is in better agreement with observations. Permafrost soil carbon stocks have increased 20-fold to reach 1006 PgC in the top 3 meters of soil, which is consistent with observation-based

estimates. We simulate that the permafrost region has been a net carbon sink over the past 150 years (+0.32±0.04 PgC.yr$^{-1}$ on average between 2005 and 2014), primarily due to carbon uptake from boreal forests. This is comparable with recent pan-Arctic carbon balance estimates, when accounting for unrepresented processes in our model (fire and riverine carbon losses). Overall, the inclusion of permafrost processes has improved the response of the model to anthropogenic perturbations in high latitudes over the past century, marking a step forward in the representation of Arctic ecosystems.





## 1 Introduction

The permafrost region, located mainly in cold high-latitude areas, is home to complex interactions between physical and biogeochemical processes. It contains large amounts of thermally protected soil organic carbon that has accumulated over millennia (IPCC SROCCC Chap.3, 2019; Hugelius et al., 2014). Anthropogenic greenhouse gas emissions and the resulting climate warming lead to permafrost thawing, which threatens these vulnerable carbon stocks (Smith et al., 2022; Burke et al., 2020; Biskaborn et al., 2019). Subsequent decomposition of the newly unfrozen permafrost carbon would lead to $CO_2$ and $CH_4$ emissions, further amplifying global warming in a positive feedback loop known as the permafrost carbon-climate feedback (Schuur et al., 2015; Schaefer et al., 2014). On the other hand, rising atmospheric $CO_2$ concentrations and increased soil nitrogen availability due to soil warming could increase vegetation productivity in a negative feedback loop, partially offsetting soil carbon losses (Salmon et al., 2016; Finger et al., 2016; Koven et al., 2011). However, the timing and magnitude of these feedbacks remain highly uncertain (Schuur et al., 2022; Schädel et al., 2018). Therefore, the resulting overall response of the carbon cycle to anthropogenic emissions in permafrost regions is a major unknown in future projections of the global carbon cycle (Kleinen and Brovkin, 2018; McGuire et al., 2018; MacDougall et al., 2012).

Earth system models (ESMs) are numerical representations of the Earth system that simulate the coupled dynamics and exchanges of energy, water and carbon between the atmosphere, the ocean and continental surfaces. Based on the representation of physical and biogeochemical mechanisms at a large range of scales, they are essential tools for studying the past, present and future dynamics of the Earth's climate and carbon cycle. In particular, their use for climate projections plays a key role in informing adaptation and mitigation policies and is at the basis for IPCC Assessment Reports. Compared to simpler models, they take into account the feedbacks between the processes that control the exchange of energy, water and carbon, and are the most comprehensive representation of the Earth system currently available. They can be driven by different socio-economic and greenhouse gas emission-related scenarios to explore possible futures, and can isolate individual feedbacks to quantify their contribution to the global response (e.g. Arora et al., 2020). ESMs are therefore particularly well suited to studying the future dynamics of the permafrost carbon cycle as they provide a mechanistic description of the complex interactions between climate and the carbon cycle.

However, despite the urgent need to accurately predict the future permafrost carbon dynamics, the physical and biogeochemical mechanisms of permafrost are still not well represented in ESMs (Schädel et al., 2024; Burke et al., 2020; Slater and Lawrence, 2013). Although efforts have been made to include physical permafrost processes in land surface models (LSMs, the land component of ESMs), including soil freeze/thaw cycles and the influence of hydrology on soil thermal properties (Cuntz and Haverd, 2018; Hagemann et al., 2016; Ekici et al., 2013; Gouttevin et al., 2012), multilayer snow schemes including snow hydrological and thermal effects and snow compaction (Decharme et al., 2016; Wang et al., 2013; Ekici et al., 2013), or soil organic matter and moss insulation (Yokohata et al., 2020; Guimberteau et al., 2018; Chadburn et al., 2015; Lawrence et al., 2008), large discrepancies remain between models. Most of the CMIP6 ESMs perform poorly in simulating critical permafrost properties such as the active layer thickness (ALT, the maximum annual thaw depth) or snow insulation, partly due to shallow and poorly resolved soil profiles (Burke et al., 2020).





Furthermore, the representation of the permafrost carbon cycle in ESMs is still in its infancy. Among the CMIP6 models, only two ESMs (CESM2 and NorESM2-LR) included a vertically resolved representation of soil carbon - an essential feature for simulating permafrost carbon dynamics - and both shared the same land surface model (CLM5) (Schädel et al., 2024). The lack of such a vertical soil carbon discretisation prevents most models from representing the large soil carbon content of the permafrost region as well as the effect of gradual permafrost thaw on soil carbon dynamics and the permafrost carbon-climate feedback (Schädel et al., 2024; Varney et al., 2022). Therefore, most models used for the calculation of remaining carbon budgets do not include permafrost carbon and the permafrost contribution must be added from external estimates (Rogelj et al., 2019). The inclusion of nitrogen processes in ESMs and their coupling to the carbon cycle has been a major advance in the last decade, although only half (six out of eleven) of the CMIP6 ESMs representing the carbon cycle had an explicit representation of the nitrogen cycle (Arora et al., 2020). An accurate representation of the nitrogen cycle is particularly important for high latitudes where vegetation is generally considered to be nitrogen-limited and where mineral nitrogen release from permafrost thaw could affect both vegetation productivity and soil organic carbon decomposition (Street and Caldararu, 2022; Wooliver et al., 2019; Beermann et al., 2017; Keuper et al., 2017). The complex interactions between carbon and nitrogen in permafrost regions could lead to very different model responses and their inclusion in ESMs is therefore key to evaluating and reducing uncertainties in future projections of permafrost carbon dynamics (Burke et al., 2022; Lacroix et al., 2022; Koven et al., 2015a). This paper describes and evaluates a new version of the IPSL Earth system model - called IPSL-Perm-LandN - designed to better simulate high-latitude processes and permafrost carbon dynamics, based on the CMIP6 version IPSL-CM6A-LR (Boucher et al., 2020). New developments include coupled carbon and nitrogen cycles and key physical and biogeochemical permafrost processes in ORCHIDEE, the land surface component of the model (Vuichard et al., 2019; Guimberteau et al., 2018; Krinner et al., 2005). In particular, the model accounts for nitrogen limitation of vegetation photosynthetic activity and decomposition of soil carbon and litter (Vuichard et al., 2019). It represents permafrost freeze/thaw cycles (based on Gouttevin et al. (2012)), soil insulation by snow, soil organic matter and surface organic layers (e.g. litter, moss, Gaillard et al., 2025b), vertically resolved soil organic carbon and nitrogen with depth-dependent dynamics, thermal protection of soil organic matter when frozen and its mixing along the vertical profile (bio- and cryoturbation).

IPSL-Perm-LandN marks an important step in the representation of high-latitude ecosystems in the IPSL ESM by integrating many first-order permafrost processes. These new developments allow the simulation of permafrost physics and carbon cycle dynamics for the first time in the IPSL ESM. It is expected to be continuously improved by integrating new mechanisms (e.g. fire/permafrost interactions or abrupt thaw) and by better constraining the processes already included.

## 2 Model description

### 2.1 General presentation

IPSL-Perm-LandN is based on IPSL-CM6A-LR, the version of the Earth system model developed by the Institut Pierre-Simon Laplace (IPSL) modeling center for the 6th phase of the Coupled Model Intercomparison Project (CMIP6) (Boucher et al., 2020; Lurton et al., 2020; Eyring et al., 2016). It is composed of the atmospheric model LMDZ (version 6A-LR) (Hourdin et al.,



2020), the oceanic model NEMO and the land surface model ORCHIDEE. The ocean model includes the ocean physics NEMO-OPA (Madec et al., 2016), the sea ice dynamics and thermodynamics NEMO-LIM3 (Rousset et al., 2015; Vancoppenolle et al., 2009) and the ocean biogeochemistry NEMO-PISCES (Aumont et al., 2015) models. The coupling between the atmosphere and the surface is done every 15 minutes while the other components of IPSL-Perm-LandN are coupled at a frequency of 90 minutes. The resolution of the atmospheric model is 144x143 points in longitude and latitude, corresponding to a resolution of 2.5°x1.3° (average resolution of 157km), and 79 vertical levels extending up to 80km. The resolution of the ocean model is 1° and 75 vertical levels.

This new configuration of the IPSL Earth System Model aims to better represent high-latitude ecosystems and climate as well as permafrost physics and carbon cycle. The main modifications compared to IPSL-CM6A-LR concern the land surface model ORCHIDEE. While IPSL-CM6A-LR included ORCHIDEE-v2, a carbon-only version of the land component, IPSL-Perm-LandN uses ORCHIDEE-v3 which includes the implementation of a fully prognostic nitrogen cycle (Vuichard et al., 2019) and several key permafrost physical and biogeochemical processes (Gaillard et al., 2025b; Zhu et al., 2019; Guimberteau et al., 2018).

Sect.2.2 and Sect.2.3 briefly recall the main characteristics of the atmosphere and ocean components. A more complete description can be found in Boucher et al. (2020).

## 2.2 Atmospheric model LMDZ

The atmospheric general circulation model used is IPSL-Perm-LandN is LMDZ6A-LR (Hourdin et al., 2020). It solves the primitive equations using a finite-difference formulation (Sadourny and Laval, 1984), and advects water vapour, solid and liquid water and trace gases with a monotonic second-order finite volume scheme (Hourdin and Armengaud, 1999; Van Leer, 1977). LMDZ6A-LR physical parameterisations are based on LMDZ5B (Hourdin et al., 2013), the version of LMDZ included in IPSL-CM5B that participated in CMIP5. The turbulent scheme is based on the turbulent kinetic energy prognostic equation of Yamada (1983), a thermal plume model (Hourdin et al., 2002; Rio and Hourdin, 2008) and a parameterization of cold pools (Grandpeix and Lafore, 2010; Grandpeix et al., 2010). Convection has been improved since LMDZ5B with a better representation of the transition from stratocumulus to cumulus clouds (Hourdin et al., 2019) and the inclusion of a statistical triggering for deep convection (Rochetin et al., 2014a, b). The radiative transfer scheme includes the Rapid Radiative Transfer Model (RRTM) for thermal infrared radiation and a six-bands versions of Fouquart and Bonnel (1980) scheme for solar radiation. Gravity waves generated by mountains, convection (Lott and Guez, 2013) and fronts (de la Cámara et al., 2016; de la Cámara and Lott, 2015) are represented, as well as the quasi-biennal oscillation. Further details on the LMDZ6A model can be found in Hourdin et al. (2020).

## 2.3 Ocean model NEMO

The version 3.6 of NEMO (Nucleus for European Models of the Ocean) is the ocean component of IPSL-Perm-LandN and includes both physical and biogeochemical processes. The ocean physics is represented by NEMO-OPA (Madec et al., 2016) and is based on the Navier-Stokes equations and a nonlinear equation of state (Roquet et al., 2015). The vertical mixing of





momentum and tracers uses a turbulent energy scheme (Blanke and Delecluse, 1993; Gaspar et al., 1990) and parameterisations of mixing caused by internal tides (de Lavergne et al., 2019; de Lavergne, 2016) and submesoscale processes (Fox-Kemper et al., 2011).

Sea ice is described by the NEMO-LIM (version 3.6) model (Rousset et al., 2015; Vancoppenolle et al., 2009). NEMO-LIM uses a distribution of ice thickness (Bitz et al., 2001; Lipscomb, 2001), allowing the representation of thin to thick ice.

Sea ice can be transported horizontally and snow can accumulate above it. Vertically, two ice layers and one snow layer are represented. Within the ice layers, the ice is represented by an elastic-viscous plastic continuum (Bouillon et al., 2013; Hunke and Dukowicz, 1997). It can dynamically exchange energy and salinity with the ocean, allowing for a prognostic evolution of the coupled system. Notably, ice albedo parameters are used for model tuning as well as the snow thermal conductivity (Boucher et al., 2020).

The ocean biogeochemistry is based on PISCES-v2 (Aumont et al., 2015) and simulates the lower trophic levels of marine ecosystems, including phytoplankton and zooplankton, and the biogeochemical cycles of carbon and main nutrients (phosphorus, nitrogen, silicon and iron). The carbon cycle includes a representation of carbonate chemistry. Nutrients are supplied to the ocean by atmospheric deposition, river inputs and sediment mobilisation. Carbon compounds can be exchanged with the atmosphere through physical and biogeochemical processes, and buried at the bottom of the ocean. The parameterisation of

nitrogen fixation has been modified compared to IPSL-CM6A-LR, which has an impact on the biological carbon pump at high temperatures.

## 2.4 Land surface model ORCHIDEE

### 2.4.1 General description

ORCHIDEE-v3 (ORganizing Carbon and Hydrology in Dynamic EcosystEms) is a state-of-the art process-based land surface

model that calculates energy, water, carbon and nitrogen exchanges between the surface and the atmosphere, as well as terrestrial physical and biogeochemical processes. It is composed of two main sub-models : SECHIBA that describes exchanges of energy and water between the atmosphere, the biosphere and the soil, and STOMATE that simulates the phenology and carbon and nitrogen dynamics of the terrestrial biosphere (Vuichard et al., 2019; Zaehle and Friend, 2010; Krinner et al., 2005). Fast processes (e.g. latent and sensible heat fluxes, photosynthesis, ecosystem respiration) are computed every 15 minutes while

slow processes (e.g. carbon and nitrogen allocation) are computed daily.

Vegetation is represented by plant functional types (PFTs), i.e. groups of species sharing similar characteristics (Prentice et al., 1992). These PFTs share the same equations for most processes, but with different parameters. ORCHIDEE-v3 represents 15 PFTs, classified into forests, grasses, crops and bare soil, describing a variety of ecosystems (Table A1). PFTs can coexist in every grid box and the fraction occupied by each PFT is read from a prescribed map (which can change on a yearly basis)

(Lurton et al., 2020). For each PFT, carbon and nitrogen are contained in seven plant pools (leaves, below- and above-ground sapwood and heartwood, fruits and fine roots), five litter pools (above- and below-ground metabolic and structural, and woody litter) and three soil pools (active, slow, and passive).





ORCHIDEE-v3 represents energy exchanges between the surface and the atmosphere and takes into account shortwave and longwave radiative fluxes, turbulent latent and sensible heat fluxes, and a ground flux (Ducoudré et al., 1993). The turbulent fluxes are calculated separately for each PFT and then summed for each grid box. This coupling with the atmosphere is regulated by vegetation properties such as its albedo and its height (which impacts on surface roughness). Within the ground, heat transfers are represented by a heat diffusion equation and depend on the mineral and organic soil properties (thermal capacity, thermal conductivity, porosity) and soil hydrology. Mineral soil properties are extrapolated from the soil texture map of Zobler (1986). Soil thermal dynamics is based on an 18-layer vertical scheme, extending down to 90m (Tab.A2). The thickness of each layer increases with depth, with thinner layers near the surface. A zero flux condition is imposed at the bottom boundary.

The model also represents exchanges of water between the surface and the atmosphere. Water reaches the land through rain or snowfall, and can be lost through evaporation of water stored in the soil but also intercepted by the canopy, transpiration by vegetation, snow sublimation, surface runoff and percolation and transfer to groundwater (i.e. drainage). Internal water exchanges between land components can also occur through various mechanisms, such as snow melt, or plant root uptake. Soil moisture is resolved on a 11-layer scheme down to 2m (the same as for soil thermics) (de Rosnay et al., 2002). Water is transferred from one layer to another according to a one-dimensional Fokker-Planck equation (Ducharne et al., 2018). A free drainage condition is imposed at the bottom boundary. Vegetation has a major influence on water exchanges by regulating evapotranspiration through stomatal closure and soil water uptake.

The representation of the carbon and nitrogen cycles have already been described in detail in Vuichard et al. (2019), Zaehle and Friend (2010) and Krinner et al. (2005). The following sections are limited to the description of relevant processes for high latitudes and new developments. A more detailed description of ORCHIDEE-v3 can be found in Sect.A.

### 2.4.2 Latent heat of soil water phase change

The improvements to permafrost physics (Sect.2.4.2, 2.4.3 and 2.4.4) have been described in Gaillard et al. (2025b) and are summarised here for the sake of completeness. The ground temperature in ORCHIDEE-v3 is calculated using a one-dimensional Fourier equation with a boundary condition at the surface allowing heat exchanges with the atmosphere (eq.5 in Gouttevin et al. (2012)) :

$$c_{\mathrm{app}} \frac{\partial T}{\partial t} = \frac{\partial}{\partial z} \left( K_{\mathrm{th}} \frac{\partial T}{\partial z} \right) \tag{1}$$

where T the soil temperature (K) and $K_{\mathrm{th}}$ the soil thermal conductivity (W.m$^{-1}$.K$^{-1}$). $c_{\mathrm{app}}$ is apparent volumetric soil thermal capacity (J.K$^{-1}$.m$^{-3}$). It incorporates volumetric soil thermal capacity and a term representing the latent heat of soil water phase changes during melting and freezing :

$$c_{\mathrm{app}} = c_p - \rho_{\mathrm{ice}} L \frac{\Delta \Theta_{\mathrm{ice}}}{\Delta T} \tag{2}$$

where $c_p$ is the volumetric soil thermal capacity (J.K$^{-1}$.m$^{-3}$), $\rho_{\mathrm{ice}}$ the ice density (kg.m$^{-3}$), L the latent heat of fusion (J.kg$^{-1}$) and $\Theta_{\mathrm{ice}}$ the volumetric ice content (m$^3$.m$^{-3}$).



Taking into account the latent heat of water phase change is essential to correctly simulate the soil thermal dynamics in the permafrost region. It acts as a buffer, absorbing energy from thawing ice in spring and summer, and releasing energy when the water refreezes in autumn and winter, thus reducing the amplitude of the seasonal cycle of ground temperature.

### 2.4.3 Modifications of soil thermal properties by soil organic carbon

Soil organic carbon (SOC) has been shown to be an important driver of surface-atmosphere energy exchanges at high latitudes
and of permafrost thermal dynamics (Zhu et al., 2019; Loranty et al., 2018). Its effect is taken into account in our model by weighting the soil thermal properties by the SOC volume fraction ($f_{SOC}$). $f_{SOC}$ is calculated as:

$$f_{SOC} = \frac{C_{SOC}}{C_{SOC\,max}} \tag{3}$$

where $C_{SOC}$ is the SOC density (kgC.m$^{-3}$) and $C_{SOC\,max}$=500 kgC.m$^{-3}$ is a reference value. $C_{SOC\,max}$ has been tuned to simulate a realistic high latitude climate (Gaillard et al., 2025b), ensuring that its value remains in the range of soil carbon densities
from the SoilGrids database (Poggio et al., 2021; Batjes et al., 2019). The heat diffusion equation (eq.1) then uses the total soil thermal conductivity and capacity (mixing mineral and organic soil properties).

Solid and dry soil thermal conductivities and the dry thermal capacity are computed as weighted averages of those of mineral and organic soils (Guimberteau et al., 2018):

$$\lambda_{solid} = (1 - f_{SOC})\lambda_{solid\,mineral} + f_{SOC}\lambda_{solid\,SOC} \tag{4}$$

$$\lambda_{dry} = (1 - f_{SOC})\lambda_{dry\,mineral} + f_{SOC}\lambda_{dry\,SOC} \tag{5}$$

$$c_{dry} = (1 - f_{SOC})c_{mineral} + f_{SOC}c_{SOC} \tag{6}$$

where $c_{mineral}$ (J.K$^{-1}$.m$^{-3}$) and $\lambda_{mineral}$ (W.m$^{-1}$.K$^{-1}$) are the thermal capacities and conductivities of solid/dry mineral soils, which depend on the dominant soil texture of the grid box. Solid refers to the solid fraction of the soil (excluding pores) while the dry fraction also includes the pores filled with air (not those filled with water). The total thermal capacity is then calculated
for each soil layer as:

$$c = c_{dry} + \Theta_{liq}c_{liq} + \Theta_{ice}c_{ice} \tag{7}$$

where $\Theta_{liq}$ (unitless) and $\Theta_{ice}$ (unitless) are the volumetric liquid water and ice contents computed by the model and $c_{liq}$ and $c_{ice}$ are the thermal capacities (J.K$^{-1}$.m$^{-3}$) of liquid water and ice, respectively equal to 4.18 10$^6$ J.K$^{-1}$.m$^{-3}$ and 2.11 10$^6$ J.K$^{-1}$.m$^{-3}$. The thermal conductivity of dry organic carbon ($c_{dry}$) is fixed at 2.5 10$^6$ J.K$^{-1}$.m$^{-3}$. For each soil layer, the thermal conductivity
is computed as:

$$\lambda = Ke\lambda_{sat} + (1 - Ke)\lambda_{dry} \tag{8}$$

where:

$$\lambda_{sat} = \lambda_{solid}^{(1-\Theta_{sat})}\lambda_{liq}^{\left(\Theta_{sat}\frac{\Theta_{liq}}{\Theta_{liq}+\Theta_{ice}}\right)}\lambda_{ice}^{\left(\Theta_{sat}\frac{\Theta_{ice}}{\Theta_{liq}+\Theta_{ice}}\right)} \tag{9}$$





with $\lambda_{\text{liq}}$ and $\lambda_{\text{ice}}$ the thermal conductivities of liquid water and ice, respectively equal to 0.57 and 2.2 W.m$^{-1}$.K$^{-1}$, and $\Theta_{\text{sat}}$

(unitless) the volumetric moisture content at saturation, which depends on the dominant mineral soil texture. The thermal conductivity of dry organic carbon ($\lambda_{\text{dry}}$) is fixed at 0.25 W.m$^{-1}$.K$^{-1}$.

Ke is the Kersten number defined for unfrozen soil as:

$$
\text{Ke} = \begin{cases} \log_{10}(S_r) + 1 & \text{if } S_r > 0.1 \\ 0.7\log_{10}(S_r) + 1 & \text{if } 0.05 < S_r \leq 0.1 \\ 0 & \text{if } S_r \leq 0.05 \end{cases} \tag{10}
$$

where $S_r$ is the saturation ratio and is calculated as $S_r = \frac{\Theta_{\text{liq}}}{\Theta_{\text{sat}}}$. For (fully or partially) frozen soils, Ke=$S_r$.

The modification of soil thermal parameters by soil organic carbon creates a coupling between the carbon cycle and soil thermodynamics, eventually impacting surface-atmosphere energy transfers. Importantly, the porosity calculated by the thermal module of ORCHIDEE-v3 differs from that used in the hydrological scheme (which is equal to that of a mineral soil), which prevents a direct feedback between soil moisture and soil temperature through soil porosity.

### 2.4.4 Modification of soil thermal properties by a surface organic layer

In the Arctic, the surface organic layer (SOL) formed by litter and groundcover vegetation (moss, lichens) may significantly reduce surface-atmosphere energy exchanges through their insulative properties and therefore thermally protect permafrost soils from warmer summer air temperatures (Loranty et al., 2018; Porada et al., 2016). In IPSL-Perm-LandN, we decided to modify the thermal capacity and conductivity of the upper soil layers to mimic the effect of such a surface organic layer on soil thermal dynamics. We further assumed that the surface organic layer covers a fraction $f_{\text{SOL}}$ of each grid box containing boreal

PFTs.

The calculation of the effect of the surface organic layer on soil thermal transfers is carried out in two steps. First, a virtual column (not explicitly represented in the model) is defined over a fraction $f_{\text{SOL}}$ of the grid box, representing moss, lichen and/or decomposing litter (dashed red in Fig.A1). The thermal capacity of the virtual column is calculated as a weighted average of the surface organic layer and soil thermal capacities:

$$c_{\text{virtual column}}\text{SID} = c_{\text{SOL}}\text{SOLT} + c_{\text{soil}}\text{SID}$$

$$\Leftrightarrow c_{\text{virtual column}} = c_{\text{SOL}}\frac{\text{SOLT}}{\text{SID}} + c_{\text{soil}} \tag{11}$$

where $c_{\text{SOL}}$ is the volumetric thermal capacity of the surface organic layer (J.K$^{-1}$.m$^{-3}$), $c_{\text{soil}}$ is the volumetric soil thermal capacity (as calculated in Sect.2.4.3, J.K$^{-1}$.m$^{-3}$), $f_{\text{SOL}}$ is the fraction of the grid box that contains the surface organic layer, SOLT is the surface organic layer thickness and SID is the soil integration depth, i.e. the depth down to which the properties of

the soil organic layer are mixed with those of the soil.



Then, the total thermal capacity of the grid box ($c_{tot}$), which takes into account the fraction not covered by the surface organic layer, is calculated as the weighted average of $c_{\text{virtual column}}$ and $c_{\text{soil}}$:

$$
\begin{aligned}
c_{\text{tot}} &= f_{\text{SOL}} c_{\text{virtual column}} + (1 - f_{\text{SOL}}) c_{\text{soil}} \\
&= f_{\text{SOL}} c_{\text{SOL}} \frac{\text{SOLT}}{\text{SID}} + c_{\text{soil}}
\end{aligned}
\tag{12}
$$

The approach for thermal conductivity is similar but takes into account its intensive nature. The thermal conductivity virtual column ($\lambda_{\text{virtual column}}$) is the equivalent thermal conductivity of the surface organic layer and soil layers in series:

$$
\lambda_{\text{virtual column}} = \frac{\lambda_{\text{SOL}} \lambda_{\text{soil}} \text{SID}}{\text{SOLT} \lambda_{\text{soil}} + \text{SID} \lambda_{\text{SOL}}}
\tag{13}
$$

where $\lambda_{\text{SOL}}$ is the thermal conductivity of the soil organic layer and $\lambda_{\text{soil}}$ is the thermal conductivity of the soil.

The total thermal conductivity of the grid box is the equivalent thermal conductivity of the surface organic layer column and
the soil column in parallel:

$$
\begin{aligned}
\lambda_{\text{tot}} &= f_{\text{SOL}} \lambda_{\text{virtual column}} + (1 - f_{\text{SOL}}) \lambda_{\text{soil}} \\
&= f_{\text{SOL}} \lambda_{\text{SOL}} \lambda_{\text{soil}} \frac{SID}{\text{SOLT} \lambda_{\text{soil}} + \text{SID} \lambda_{\text{SOL}}} + (1 - f_{\text{SOL}}) \lambda_{\text{soil}}
\end{aligned}
\tag{14}
$$

Finally, the mineral soil capacity and conductivity are replaced by $c_{\text{tot}}$ and $\lambda_{\text{tot}}$ in all the soil layers between the surface and SID.

In this study, we chose $f_{\text{SOL}}=1$, SOLT=0.03m and SID=0.03m for evaluating the model. This value of SOLT is consistent with the moss thickness measured in Soudzilovskaia et al. (2013). SID was chosen small enough to allow the soil organic layer to influence surface-atmosphere energy exchanges, but to limit the modification of soil thermal properties to the very top layers.

     In addition, the thermal properties of the surface organic layer depend on its water content (Soudzilovskaia et al., 2013;
O'Donnell et al., 2009). They are parameterized using observations made on mosses, using the upper soil water content of each soil layer down to SID as a proxy for the water content of the surface organic layer. The thermal capacity of the soil organic layer is calculated as:

$$
c_{\text{SOL}} =
\begin{cases}
c_{\text{SOL dry}} + \theta(c_{\text{SOL wet}} - c_{\text{SOL dry}}) & \text{if T<2°C} \\
c_{\text{SOL dry}} + \theta[(\frac{T}{2} + 1)c_{\text{SOL wet}} - \frac{T}{2}c_{\text{SOL frozen}} - c_{\text{SOL dry}}] & \text{if -2°C}\leq\text{T}\leq\text{0°C} \\
c_{\text{SOL dry}} + \theta(c_{\text{SOL frozen}} - c_{\text{SOL dry}}) & \text{if T>0°C}
\end{cases}
\tag{15}
$$

where $c_{\text{SOL dry}}$, $c_{\text{SOL wet}}$ and $c_{\text{SOL frozen}}$ (J.m$^{-3}$.K$^{-1}$) are the thermal capacities of dry, wet and frozen surface organic layers,
respectively, and $\theta$ is the volumetric moisture content (unitless).

     The thermal conductivity of the soil organic layer is calculated as:

$$
\lambda_{\text{SOL}} = \lambda_{\text{SOL dry}} + \theta(\lambda_{\text{SOL sat}} - \lambda_{\text{SOL dry}})
\tag{16}
$$





where $\lambda_{\text{SOL dry}}$ is the thermal conductivity of a dry surface organic layer and $\lambda_{\text{SOL sat}}$ is the thermal conductivity of a saturated surface organic layer, calculated as:

$$\lambda_{\text{SOL sat}} = \lambda_{\text{SOL liq}}^{\left(\theta_{\text{sat}} \frac{\theta_{\text{liq}}}{\theta_{\text{liq}} + \theta_{\text{ice}}}\right)} \lambda_{\text{SOL frozen}}^{\left(\theta_{\text{sat}} \frac{\theta_{\text{ice}}}{\theta_{\text{liq}} + \theta_{\text{ice}}}\right)} \tag{17}$$

The values of surface organic layer thermal properties are taken from in situ measurements and laboratory experiments (Soudzilovskaia et al., 2013; O'Donnell et al., 2009). Thermal capacities are set to $c_{\text{SOL dry}}=0.29\times10^6$ J.m$^{-3}$.K$^{-1}$, $c_{\text{SOL wet}}=4.29\times10^6$ J.m$^{-3}$.K$^{-1}$ and $c_{\text{SOL frozen}}=3.26\times10^6$ J.m$^{-3}$.K$^{-1}$ (Soudzilovskaia et al., 2013; Druel et al., 2017). Thermal conductivities are equal to $\lambda_{\text{SOL dry}}=0.05$ W.m$^{-1}$.K$^{-1}$, $\lambda_{\text{SOL wet}}=0.56$ W.m$^{-1}$.K$^{-1}$ and $\lambda_{\text{SOL frozen}}=1.40$ W.m$^{-1}$.K$^{-1}$ (O'Donnell et al., 2009; Porada et al., 2016).

### 2.4.5 Snow

ORCHIDEE-v3 uses a 3-layer snow scheme of intermediate complexity with dynamic layer thickness, which was already used in IPSL-CM6A-LR. Snow strongly influences the surface-atmosphere energy transfer at high latitudes due to its insulating properties. Heat diffusion within the snowpack is accounted for by a heat-transfer equation:

$$c_p \frac{\partial T_j}{\partial t} = \frac{\partial}{\partial z}\left(\kappa_j \frac{\partial T_j}{\partial z}\right) + \frac{\partial R}{\partial z} \tag{18}$$

where $T_j$ is the snow temperature of the layer j, $c_p$ is the snow heat capacity (J.K$^{-1}$.m$^{-3}$), $\kappa_j$ is the thermal conductivity of the snow (W.m$^{-1}$.K$^{-1}$) and takes into account vapour transfer in the snow, z is the vertical coordinate and t is time. $\frac{\partial R}{\partial t}$ is the solar-radiative energy source and depends on the incoming solar radiative energy and the snow depth.

Water phase change can occur within the snowpack as snow melts or refreezes, further affecting soil hydrology and surface-atmosphere water exchange. In particular, snow can melt in the upper layer of the snowpack due to solar radiation, infiltrate down to the next layer and may refreeze, releasing latent heat and heating lower layers. Snow compaction is also represented and depends on the weight of the overlying snow. It modifies the density and thickness of snow layers over time. Finally, the snow albedo is included and depends on the snowfall rate and the liquid water content of the snowpack.

Further details on these processes and their implementation can be found in Wang et al. (2013).

### 2.4.6 Soil carbon and nitrogen dynamics

Soil organic carbon and nitrogen dynamics in ORCHIDEE follow a CENTURY-based scheme (Parton et al., 1993) which is schematised in Fig.A2 and Fig.A3. Plant residues are divided into structural and metabolic litter pools according to their lignin content. Litter decomposition follows a first-order kinetics with pool-dependent decomposition factors, and depends on temperature, moisture and lignin content. Part of the decomposed carbon is respired as $CO_2$ and the remaining flux is transferred to soil organic carbon (SOC) pools. Importantly, the model only represents $CO_2$ emissions and does not include $CH_4$ dynamics. Active, slow and passive SOC pools have different turnover times and can exchange carbon with each other, each time with an associated loss of $CO_2$ through microbial respiration. SOC decomposition also follows a first-order kinetics





with a dependence on soil temperature, moisture and texture (i.e. soil sand, silt and clay content):

$$\left.\frac{\partial C_i}{\partial t}\right|_{\text{decomposition}} = k_i \cdot f(T) \cdot f(\text{moisture}) \cdot f(\text{texture}) \cdot C_i \tag{19}$$

where $C_i$ is the carbon content of the pool i (where i corresponds to active, slow or passive) and $k_i$ is the decomposition factor.

Nitrogen is decomposed at the same rate as carbon. Nitrogen fluxes are driven by carbon fluxes and the C:N ratios of the pools (Fig.A3). The nitrogen flux between a pool A and a pool B is expressed as the product of the corresponding carbon flux and of the N:C ratio of the receiving pool:

$$f_{\text{nitrogen, A}\rightarrow\text{B}} = f_{\text{carbon, A}\rightarrow\text{B}} \cdot \text{N:C}_\text{B} \tag{20}$$

The nitrogen associated with the carbon lost by respiration is assumed to be mineralised. If the decomposed organic nitrogen cannot meet the demand of the receiving pools, mineral nitrogen is immobilised to complete the nitrogen flux. If the amount of nitrogen in the mineral pool is not sufficient, nitrogen is taken from the atmosphere to complete the required immobilisation flux. Conversely, if there is an excess of decomposed nitrogen, it is mineralised and transferred to the mineral nitrogen pool. Furthermore, decomposition rates are independent of C:N ratios. These ratios are dynamic and depend on the concentration

of soil mineral nitrogen ($NH_4^+$ and $NO_3^-$), with a lower nitrogen demand (higher C:N ratios) when mineral nitrogen is scarce, and a higher demand (lower C:N ratios) when mineral nitrogen stocks are high.

Soil mineral nitrogen follows the DNDC model which accounts for ammonium ($NH_4^+$), nitrates ($NO_3^-$), nitrogen oxides ($NO_x$) and nitrous oxide ($N_2O$) (Li et al., 1992, 2000; Zhang et al., 2002). It represents nitrification, denitrification, mineralisation and immobilisation, ammonium adsorption and desorption, plant uptake ($NH_4^+$ and $NO_3^-$ only), gaseous emissions and

leaching (Fig.A4). Plant uptake is expressed as :

$$N_{\text{up}} = v_{\text{max}} \times N_{\text{min}} \times \left( k_{\text{Nmin}} + \frac{1}{K_{\text{Nmin}} + N_{\text{min}}} \right) \times f(\text{NC}_{\text{plant}}) \times C_{\text{root}} \tag{21}$$

where $N_{\text{up}}$ is the plant nitrogen uptake (gN.m$^{-2}$.d$^{-1}$), $N_{\text{min}}$ is the amount of mineral nitrogen available ($NH_4^+$+$NO_3^-$, gN.m$^{-2}$), $v_{\text{max}}$ is the maximum rate of nitrogen uptake (gN.gC$^{-1}$.day$^{-1}$), $k_{\text{Nmin}}$ (m$^2$.gN$^{-1}$) and $K_{\text{Nmin}}$ (gN.m$^{-2}$) are Michaelis-Mentens coefficients, $C_{\text{root}}$ the root carbon mass per unit area (gC.m$^{-2}$) and $f(\text{NC}_{\text{plant}})$ the dependency of plant nitrogen uptake to $\text{NC}_{\text{plant}}$,

expressed as :

$$f(\text{NC}_{\text{plant}}) = \max\left( \frac{\text{NC}_{\text{plant}} - \text{nc}_{\text{leaf,max}}}{\text{nc}_{\text{leaf,min}} - \text{nc}_{\text{leaf,max}}}, 0 \right) \tag{22}$$

where $\text{nc}_{\text{leaf,min}}$ and $\text{nc}_{\text{leaf,max}}$ are the minimum and maximum leaf N:C ratios, respectively (PFT-dependent), and $\text{NC}_{\text{plant}}$ is defined as the mean N:C ratio of leaves, roots and labile nitrogen pools :

$$\text{NC}_{\text{plant}} = \frac{N_{\text{leaf}} + N_{\text{root}} + N_{\text{labile}}}{C_{\text{leaf}} + C_{\text{root}} + C_{\text{labile}}} \tag{23}$$

Further details can be found in Zaehle and Friend (2010) and Vuichard et al. (2019).

A major improvement from IPSL-CM6A-LR to IPSL-Perm-LandN is the vertical discretisation of soil organic carbon and nitrogen on an 18-layer scheme (the same as for soil thermal dynamics), with depth-dependent decomposition rates depending





on environmental conditions. This is particularly important in permafrost regions where the upper soil layers can thaw while deeper layers remain frozen, keeping organic matter thermally protected. Soil mineral nitrogen, however, is not vertically resolved and remains represented on a single soil layer in each grid box.

Organic carbon and nitrogen can be exchanged between soil layers through bio- or cryoturbation. This process is described by a diffusion equation:

$$\left.\frac{\partial C_i}{\partial t}\right|_{\text{cryoturbation}} = D\frac{\partial C_i^2}{\partial z^2} \tag{24}$$

where $C_i$ is the carbon or nitrogen content of the pool i at a given depth and time, and D is the diffusive mixing rate. In the permafrost region (defined as ALT $\leq$ 3m), D is set to $10^{-3}$ m$^2$.yr$^{-1}$ in the active layer and decreases linearly to zero between ALT and $3\times$ALT. Elsewhere, D is set to $10^{-4}$ m$^2$.yr$^{-1}$ in the top 2m of soil to represent bioturbation.

The depth-dependent decomposition of soil organic matter depends on environmental conditions. In particular, it is modulated as a function of temperature (f(T) in eq.19):

$$f(T) = \begin{cases} \exp\left(\log(Q_{10})\left(\frac{T-T_{\text{ref}}}{10}\right)\right) & \text{if } T>0°C \\ (T+1)\cdot\exp\left(-\log(Q_{10})\left(\frac{T_{\text{ref}}}{10}\right)\right) & \text{if } -1°C<T\leq 0°C \\ 0 & \text{if } T\leq -1°C \end{cases} \tag{25}$$

where $Q_{10}$=2 and $T_{\text{ref}}$=30°C. Above 0°C, decomposition follows a $Q_{10}$ function ($Q_{10}$=2), then decreases linearly to zero between 0°C and -1°C. Below -1°C no decomposition can take place.

Decomposition also increases monotonically with soil moisture (f(moisture) in eq.19):

$$f(\text{moisture}) = \max(0.25; -1.1\cdot\text{moisture}^2 + 2.4\cdot\text{moisture} - 0.29) \tag{26}$$

where moisture represents the humidity profile (unitless) and is between 0 and 1. Below 2m (the depth to which hydrology is resolved), a constant soil moisture profile is used, taken from the lowest layer.

Overall, for each soil layer, the organic matter dynamics follows the equation below:

$$\frac{\partial C_i(z,t)}{\partial t} = I_i(z,t) - k_i\cdot f(T)(z,t)\cdot f(\text{moisture})(z,t)\cdot f(\text{texture})\cdot C_i(z,t) + D(z,t)\frac{\partial C_i^2(z,t)}{\partial z^2} \tag{27}$$

where $I_i$ are the carbon or nitrogen inputs to the pool i, the second term corresponds to decomposition and the third term to vertical mixing.

### 2.4.7 Initialisation of soil organic carbon and nitrogen

IPSL-Perm-LandN is unable to build up the observed large permafrost carbon stocks from scratch during spinup (even covering several thousands of years) due to the constant pre-industrial climate forcing of the spinup (i.e. no glacial/interglacial cycles), the long timescales required for carbon burial, missing processes (dust deposition, peat development) and the lack of deep permafrost deposits. Consistent representation of permafrost soil carbon is critical to avoid biases in its insulating effect or





underestimation of future permafrost $CO_2$ emissions. Therefore, soil organic carbon and nitrogen pools are initialised with the contemporary observation-based product SoilGrids, which provides a global map of soil organic carbon and nitrogen with a detailed depth resolution (version 2.0, Poggio et al., 2021; Batjes et al., 2019). This allows the unfrozen soil layers to reach an equilibrium state driven by the carbon cycle and climate dynamics, while the organic matter in the frozen layers cannot be decomposed throughout the spinup. SoilGrids gathers observations from about 240 000 locations and uses more than 400 covariates. The original product has a horizontal resolution of 250m and 6 vertical layers down to 2m (0-5cm, 5-15cm, 15-30cm, 30-60cm, 60-100cm and 100-200cm). It has been conservatively regridded to the ORCHIDEE horizontal grid (2.5°x1.25°) using the CDO "remapcon" command, and vertically interpolated to the 18-layer scheme. Organic carbon and nitrogen stocks were divided into active, slow and passive fractions following the fractions given in Koven et al. (2015b) (2% in active, 29% in slow and 69% in passive pools). As there is no global gridded map of soil mineral nitrogen, the mineral nitrogen pool is initialised to zero prior to the spinup.

## 3 Methods

### 3.1 Simulations and forcings

#### 3.1.1 Spinup

Before running IPSL-Perm-LandN under varying forcings, it is necessary to bring the carbon and nitrogen pools into equilibrium. This is done by performing a spinup in pre-industrial configuration. The spinup protocol starts with a spinup using ORCHIDEE offline (i.e. not coupled to the atmosphere and the ocean) under pre-industrial conditions for 2600 years. The model is forced by a 50-year cyclic climate from the spinup of IPSL-CM6A-LR (*piControl* simulation of CMIP6), which has an atmosphere and ocean physics similar to IPSL-Perm-LandN. The PFT map (Lurton et al., 2020) and nitrogen deposition (National Center for Atmospheric Research-Chemistry-Climate Model Initiative) and fertilisation (Hurtt et al., 2020) remain at their 1850 values. Biological nitrogen fixation follows the approach of Cleveland et al. (1999) and is fixed in time (Vuichard et al., 2019). ORCHIDEE is then coupled to LMDZ (atmosphere) and NEMO (ocean) to form IPSL-Perm-LandN. The model is restarted from the offline ORCHIDEE spinup for land variables and from a spinup of IPSL-CM6A-LR for atmosphere and ocean variables. Importantly, the restart state of the ocean is from a 4000-year simulation, providing initial already equilibrated ocean physics and carbon pools. The spinup is run in concentration-driven configuration for 670 years. The land forcings remain the same and the atmospheric and oceanic forcings are fixed at their pre-industrial values. In particular, the atmospheric $CO_2$ concentration is set to 284 ppm. After the spinup, the coupled model is considered to be sufficiently close to equilibrium to avoid significant drifts in global climate variables and in the land and ocean net carbon fluxes in historical simulations (see Tab.A3).



### 3.1.2 Historical simulations

Three historical simulations (1850-2014) were performed with IPSL-Perm-LandN following the CMIP6 protocol in order to quantify the uncertainty in the simulated processes due to internal model variability. They differed only in their restart state, as the model was restarted from three distinct pre-industrial climate states (years 420, 450, and 480). These restart points were verified to be significantly different in terms of global temperature, thus providing three distinct restart states within the internal variability of IPSL-Perm-LandN. The forcings are provided by the CMIP6 input4MIP project (https:

//aims2.llnl.gov/search/input4MIPs/), including greenhouse gas concentrations, which were taken as global averages from Meinshausen et al. (2017). Tropospheric and stratospheric ozone radiative forcings came from Checa-Garcia et al. (2018) and Hegglin et al. (2016). Tropospheric aerosols were not simulated interactively by IPSL-Perm-LandN and were prescribed from a historical LMDZOR-INCA simulation (i.e. a coupled surface-atmosphere simulation with tropospheric chemistry). In addition, stratospheric (volcanic) aerosols were prescribed from the version 3 of the dataset from Thomason et al. (2018) as a

latitude-height time-varying climatology. Finally, the solar forcing is provided by Matthes et al. (2017).

Atmospheric nutrient deposition to the ocean (iron, phosphorus, and silicate) was provided by LMDZOR-INCA simulations. Wet and dry oceanic deposition of nitrogen (inorganic nitrate and ammonium) came from the National Center for Atmospheric Research-Chemistry-Climate Model Initiative nitrogen deposition rates. The river supply of biogeochemical elements to the ocean was sourced from Mayorga et al. (2010) for dissolved inorganic and organic nitrogen, dissolved inorganic and inorganic

phosphorus, and silicate. Dissolved inorganic carbon and alkalinity were provided by the simulations using the Global Erosion Model of Ludwig et al. (1996). The river supply of iron was calculated from the river supply of inorganic carbon, assuming a constant Fe/dissolved inorganic carbon ratio.

Land cover (i.e. the PFT map), wood harvest and nitrogen fertilisation are provided by the land use harmonisation database Hurtt et al. (LUH2, 2020). Nitrogen deposition is provided by th National Center for Atmospheric Research-Chemistry-Climate

Model Initiative and BNF follows the approach of Cleveland et al. (1999).

A complete description of the implementation of the forcings can be found in Lurton et al. (2020).

### 3.2 Evaluation data

Surface air temperature data is taken from ERA5 reanalysis (Copernicus Climate Change Service, 2019) for absolute values and NOAAGlobalTemp (Huang et al., 2023) and HadCRUT (Morice et al., 2021) for temperature anomalies compared to

1850-1900. Total precipitation data come from ERA5 and MSWEP (Beck et al., 2019) and snowfall data from ERA5 only. Sea surface temperature and salinity come from the World Ocean Atlas (Locarnini et al., 2024; Reagan et al., 2024). Sea ice concentration is taken from the National Snow and Ice Data Center (DiGirolamo et al., 2022). The extent of the permafrost region is taken from ESA-CCI (Westermann et al., 2024a) and active layer thickness data come from ESA-CCI and the CALM network (Westermann et al., 2024b; Brown et al., 2000). GPP comes from the FLUXCOM network (Jung et al., 2020), RH from

Bond-Lamberty and Thomson (2010) and Hashimoto et al. (2015), and NBP from the 2023 Global Carbon Budget (GCB2023, Friedlingstein et al., 2023) and the CAMS inversion product (Chevallier et al., 2023). Ocean net air-sea carbon flux come from



GCB2023. Gridded data of vegetation biomass is taken from the ESA-CCI product (Santoro and Cartus, 2021) and soil carbon comes from HWSD (Wieder et al., 2014), SoilGrids (Poggio et al., 2021; Batjes et al., 2019) and NCSCD (Hugelius et al., 2013). Anthropogenic fossil emissions are from GCB2023 (Friedlingstein et al., 2023).

Data for C4MIP models has been retrieved from the IPSL ESGF node (https://esgf-node.ipsl.upmc.fr/projects/esgf-ipsl/) at the time of the study. For each model, the first 10 members are used, except for UKESM1-0-LL and NorESM2-LM where only 4 and 3 members were available, respectively. For IPSL-CM6A-LR, the 33 members are used.

### 3.3    Evaluation metrics

### 3.3.1    Permafrost region

A necessary but tricky step in the study of permafrost modeling is to clearly define permafrost in the model. A first clarification is needed to avoid the common confusion between the *permafrost region* and the *permafrost area* (Obu, 2021). The *permafrost region* is defined as the total area covered by permafrost zones (continuous, discontinuous, sporadic and isolated patches). However, each permafrost zone is not completely underlain by permafrost and the actual area underlain by permafrost is smaller than the *permafrost region*. This area actually underlain by permafrost is called the *permafrost area*, and takes into

account, for example, that there is more permafrost in the continuous than in the sporadic zone. Many observation products provide both the *permafrost region* and the *permafrost area* (Obu, 2021; Obu et al., 2019; Gruber, 2012). In Earth System Models, however, each pixel of the grid either contains permafrost or does not. A finer description of permafrost would require the representation of sub-grid land surface heterogeneity and the estimation of a permafrost fraction for each pixel, which is not the case in current ESMs despite promising developments (Shirley et al., 2022; Beer, 2016; Fowler et al., 2024; Torres-

Rojas et al., 2022). Thus ESMs can only represent the *permafrost region* as the total area where grid boxes contain permafrost. However this modeled *permafrost region* is slightly different from the one estimated from observations. As the ESMs represent the dominant environmental conditions over each grid box, areas with small amounts of permafrost are likely to be missing permafrost. On the contrary, in areas with observed permafrost fractions greater than 50%, the majority of the area is underlain by permafrost and the models should consider them as pixels containing permafrost. Thus, continuous and discontinuous

permafrost zones (>50% of permafrost) should be similar between models and observations while disagreement is expected for sporadic permafrost and isolated patches (<50% of permafrost).

Apart from this, a second source of uncertainty comes from the way in which is decided whether a model grid box contains permafrost or not. Comparing 10 different definitions of permafrost in ESMs, Steinert et al. (2024) found large differences within each model of the CMIP6 ensemble and showed that the spread due to permafrost definition could even be larger than

the inter-model spread. Among the classical permafrost definitions, those based on ground-air temperature coupling show a better agreement between models but miss the complexity introduced by ground thermodynamics by implicitly assuming the same ground thermodynamics for all models. More relevant definitions are based on ground thermal properties and are closer to the original definition of permafrost. A direct application of this definition in models would be to define the zero annual amplitude depth ($D_{zaa}$) as the minimum soil depth at which the temperature variation within a year is less than 0.1°C. If the





temperature at $D_{zaa}$ is less than or equal to 0°C for at least two consecutive years, there is assumed to be permafrost in the
grid box (Burke et al., 2020). However the $D_{zaa}$ can be deep, especially in models with a deep soil column such as IPSL-
Perm-LandN. With this definition, if deep permafrost is modeled, the grid box is marked as containing permafrost. This can
be problematic if the lower soil layers are poorly represented. For instance, the lower ground boundary condition in IPSL-
Perm-LandN does not represent the heat coming from the Earth's mantle, resulting in an incorrect geothermal gradient. This

can cause deep ground to remain unrealistically frozen and to overestimate the area of permafrost using this definition. This is
why in this study, we chose another commonly used permafrost definition, based on the active layer thickness (ALT) (McGuire
et al., 2018; Koven et al., 2013). If the ALT is less than 3m, i.e. if the annual maximum thaw depth is less than 3m, the grid box
is said to contain permafrost. This definition includes surface permafrost but excludes deep permafrost (i.e. below 3m), which
is fine for two reasons :

–  IPSL-Perm-LandN poorly represents deep soil temperature profile and focusing on surface permafrost avoids overesti-
mating the permafrost region.

   –  The vast majority of soil organic carbon is in the top 3m of soil in IPSL-Perm-LandN and soil carbon decomposition
following permafrost thaw would occur within the top 3m of soil.

Thus we chose to define the permafrost region ($\mathcal{R}_{permafrost}$) as the total area where ALT<3m, i.e. :

$$\mathcal{R}_{permafrost} = \sum_{ilon=1}^{144} \sum_{ilat=1}^{143} \delta(ilon,ilat) \cdot \mathcal{A}(ilon,ilat) \cdot f_{land}(ilon,ilat)$$

with $f_{land}(ilon,ilat)$ the fraction of land in the grid box, $\mathcal{A}(ilon,ilat)$ the grid box area and

$$\delta(ilon,ilat) = \begin{cases} 1 & \text{if ALT<3m} \\ 0 & \text{otherwise} \end{cases}.$$

The permafrost region is calculated for IPSL-Perm-LandN, IPSL-CM6A-LR, C4MIP models and the ESA-CCI observation
product (Westermann et al., 2024a). As some C4MIP models have a poorly resolved soil thermal profile, a linear interpolation at

3m depth is performed instead of taking the temperature of the nearest soil layer. If the interpolated 3m-temperature is less than
or equal to 0°C, the ALT is less than 3m and the grid box contains permafrost. For IPSL-Perm-LandN, the yearly maximum
ALT is directly available and is used to calculate the size of the permafrost region (*altmax*<3m). The ESA-CCI observation
product provides the permafrost fraction ($f_{perm}$) for each pixel, which allows the calculation of the permafrost region (area
where $f_{perm}$>0), the permafrost area (area weighted by $f_{perm}$) and the region of continuous and discontinuous permafrost (area

where $f_{perm}$>0.5).

### 3.3.2   Active layer thickness

The spatially-averaged time evolution of the active layer thickness is computed using a mask of the permafrost region. This
mask is defined as the simulated 2005-2014 permafrost region, using the definition ALT<3m.





### 3.3.3 Compatible $CO_2$ emissions

Instead of prescribing anthropogenic $CO_2$ emissions to IPSL-Perm-LandN, the historical simulations are run with an imposed atmospheric $CO_2$ concentration. This prevents the simulated land and ocean carbon fluxes from feeding back onto climate, removing a source of uncertainty for the study of atmospheric processes, despite the use of a spatially homogeneous $CO_2$ concentration with no vertical gradient. However, these fluxes can be used in addition to atmospheric $CO_2$ changes to calculate the fossil fuel emissions that are compatible with the prescribed $CO_2$ concentration scenarios. The rate of compatible fossil fuel

emissions is equal to the sum of the rate of atmospheric $CO_2$ change, the net atmosphere-land and atmosphere-ocean fluxes, i.e. :

$$E_{FF} = G_{ATM} + F_{A-O} + F_{A-L} \tag{28}$$

with $E_{FF}$ the rate of anthropogenic fossil fuel emissions (PgC.yr$^{-1}$), $G_{ATM}$ the rate of change of atmospheric $CO_2$ concentration (PgC.yr$^{-1}$), $F_{A-O}$ the net atmosphere-ocean flux (PgC.yr$^{-1}$, positive for ocean uptake) and $F_{A-L}$ the net atmosphere-land flux

(PgC.yr$^{-1}$, positive for land uptake). Land-use change emissions are included in the NBP, and therefore in $F_{A-L}$.

## 4 Results and discussion

### 4.1 Atmosphere physics

Over the period 1940-2014, the mean annual land surface air temperature (SAT) is about 1.5°C colder than the ERA5 reanalysis (Fig.1 (a)). During the last decade of the simulation (2005-2014), the mean land SAT of IPSL-Perm-LandN is 13.46±0.14°C

while ERA5 has a warmer land SAT of 14.84°C. IPSL-Perm-LandN is consistently very close to IPSL-CM6A-LR as both share the same radiative scheme, and is at the lower bound of the C4MIP range, although the models generally tend to correctly simulate warming rates rather than absolute temperatures. The cold bias in IPSL-Perm-LandN is mainly due to underestimated tropical and mid-latitude temperatures while the Arctic land SAT is closer to ERA5 estimates (Fig.1 (b)). Although the absolute land temperature is too cold, the land SAT anomaly relative to 1850-1900 is close to observations. Over land (emerged

land excluding Greenland and Antarctica), IPSL-Perm-LandN has warmed by +1.60±0.14°C while the observations show a warming of +1.40°C for NOAAGlobalTemp and +1.16°C for HadCRUT (Fig.1 (c)). In contrast to the absolute temperature, the land SAT change compared to the 1850-1900 average is at the upper limit of the range of the C4MIP models. This relatively high warming comes mainly from the tropics and the Arctic (Fig.1 (d) and A5 (a)). In particular, the Arctic amplification is overestimated in IPSL-Perm-LandN with a high latitude warming twice as large as in the observations. This Arctic warming

bias was already present in IPSL-CM6A-LR and is amplified in IPSL-Perm-LandN. In addition, when including the oceans to compute the global surface air temperature (GSAT) anomaly, IPSL-Perm-LandN deviates from the observations and starts to warm faster from 1990 onwards, driven by a strong oceanic warming in the Arctic ocean (Fig.A6). The mean global warming for 2005-2014 relative to 1850-1900 is +1.27±0.12°C for IPSL-Perm-LandN and +0.84°C (NOAAGlobalTemp) and +0.80°C (HadCRUT) for observation-based datasets. This departure from observations in the recent period was already present in IPSL-





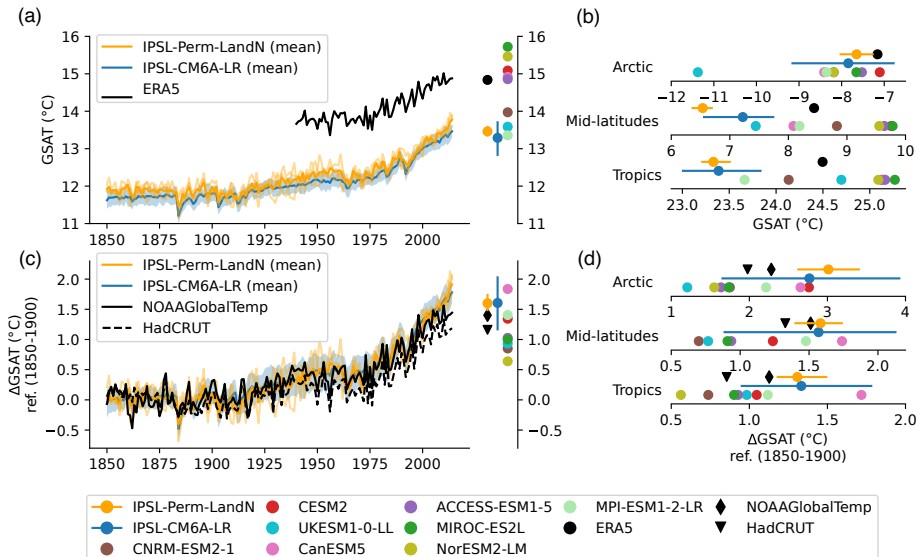

**Figure 1. Historical surface temperature over land**. **(a)** Mean global surface air temperature (GSAT) over land over the historical period for IPSL-Perm-LandN, IPSL-CM6A-LR and ERA5 reanalysis. Colored dots represent the mean GSAT (2005-2014) for IPSL-Perm-LandN, IPSL-CM6A-LR, ERA5 and C4MIP models. Light orange lines represent the three historical members for IPSL-Perm-LandN. The light blue envelope corresponds to one standard deviation between members of IPSL-CM6A-LR. **(b)** Mean land GSAT (2005-2014) over the Arctic (>60°N), mid-latitudes (30°S-60°S and 30°N-60°N) and the tropics (30°S-30°N) for IPSL-Perm-LandN and C4MIP models. **(c)** Anomaly of mean land GSAT relative to 1850-1900 for IPSL-Perm-LandN, IPSL-CM6A-LR, NOAAGlobalTemp and HadCRUT reanalyses. Colored dots represent the mean GSAT anomaly (2005-2014) for IPSL-Perm-LandN, IPSL-CM6A-LR, NOAAGlobalTemp, HadCRUT and C4MIP models. Light orange lines represent the three historical members for IPSL-Perm-LandN. The light blue envelope corresponds to one standard deviation between members of IPSL-CM6A-LR. **(d)** Mean land GSAT anomaly over the Arctic (>60°N), mid-latitudes (30°S-60°S and 30°N-60°N) and the tropics (30°S-30°N) for IPSL-Perm-LandN, IPSL-CM6A-LR and C4MIP models. The products NOAAGlobalTemp and HadCRUT provide global mean surface temperature (GMST) anomaly, defined as land surface air temperature anomaly over land and sea surface temperature anomaly over the ocean. GMST and GSAT differ by at most 10% (IPCC AR6 WGI Chap.2, 2021).

CM6A-LR and depends on the reference period used to compute the anomaly (Boucher et al., 2020). The Arctic amplification is strongly overestimated as it was the case for IPSL-CM6A-LR. Most of this bias is due to strong oceanic warming at high latitudes (Fig.A6 (f)).

The mean total precipitation (liquid+solid) in IPSL-Perm-LandN for the period 2005-2014 is shown in Fig.2 (a). The latitudinal distribution of precipitation is very close to the observations in the Arctic and mid-latitudes (Fig.2 (a) and (c)). In
the tropics, although the model correctly represents the ITCZ, it has a pronounced peak at 5°S, which is much lower in the observations. Such a double ITCZ is a known bias in many CMIP6 models and could be due to the representation of deep convection as well as model resolution (Ma et al., 2023). The mean total snowfall is represented on Fig.2 (b). Its latitudinal distribution follows that of ERA5 and, in particular, the Arctic snowfall is well represented in the recent period (Fig.2 (b) and





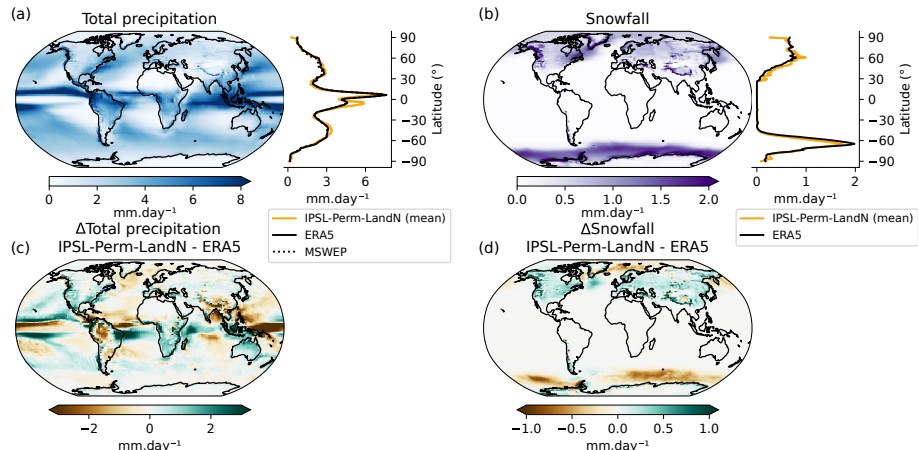

**Figure 2. Historical global precipitation and snowfall**. **(a)** Left : map of mean total precipitation (liq+sol) over 2005-2014 for IPSL-Perm-LandN. Right : zonal mean of total precipitation over 2005-2014 for IPSL-Perm-LandN, ERA5 reanalysis and MSWEP observation product. **(b)** Left : map of mean snowfall (2005-2014) for IPSL-Perm-LandN. Right : zonal mean of snowfall (2005-2014) for IPSL-Perm-LandN and ERA5. **(c)** Difference in mean total precipitation (liq+sol) between IPSL-Perm-LandN and ERA5 over 2005-2014. **(d)** Difference in mean snowfall between IPSL-Perm-LandN and ERA5 over 2005-2014.

(d)). However, the good agreement between IPSL-Perm-LandN and ERA5 masks a slight overestimation of Arctic snowfall

over land and a slight underestimation over the ocean. In addition, the mean seasonality of both total precipitation and snowfall is well captured by the model in the Arctic (Fig.A7). In the mid-latitudes, the seasonal cycle of snowfall is well represented while total precipitation is overestimated by up to 0.16 mm.day⁻¹, except in late summer. Although total precipitation has a double ITCZ in the tropics, the amplitude and phase of its seasonal cycle are in agreement with observations. In general, both total precipitation and snowfall are close to those of IPSL-CM6A-LR.

**4.2  Ocean physics**

The sea surface temperature (SST) mean pattern computed over the historical period in IPSL-Perm-LandN is quite similar to that of IPSL-CM6A-LR, as the same version of the ocean model NEMOv3.6 was used. The main bias in IPSL-Perm-LandN is a negative SST anomaly in the North Atlantic ocean compared to observations from the World Ocean Atlas over the period 2005-2014, which is associated with the position of the North Atlantic drift (Fig.3 (a)). This bias was already present in IPSL-

CM6A-LR but was less pronounced (Boucher et al., 2020) (Fig.A8 (a)). The maximum temperature negative anomaly around 45°N (in the box 60-15°W, 40-55°N) for the period 2005-2014 is -7.2°C in IPSL-Perm-LandN while it was -5.5°C for IPSL-CM6A-LR. Such a cold bias is a common feature of CMIP6 models and is stronger in winter (Zhang et al., 2023). Other classical SST biases of CMIP6 models are present in IPSL-Perm-LandN : warm biases in eastern ocean borders (although not very strong along South America), cold mid-latitudes and a warm bias near Antarctica (Zhang et al., 2023; Boucher et al.,

2020). Sea surface salinity (SSS) also shows similar patterns as IPSL-CM6A-LR (Fig.3 (b) and Fig.A8 (b)). A negative salinity




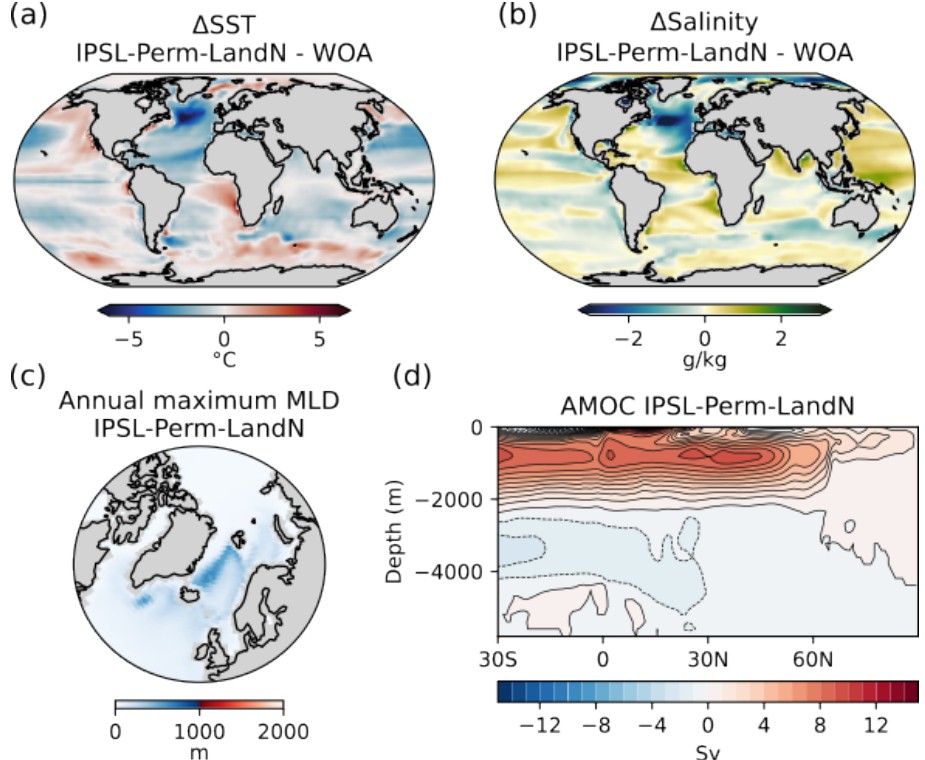

**Figure 3. Historical ocean physics for IPSL-Perm-LandN**. Difference in annual mean sea surface **(a)** temperature and **(b)** salinity between IPSL-Perm-LandN the World Ocean Atlas (2005-2014). **(c)** Mean annual maximum mixed layer depth (2005-2014). **(d)** Atlantic meridional overturning stream function, on average over 2005-2014.

anomaly is observed in the North Atlantic - in the same region as the cold SST bias - but has been reduced in IPSL-Perm-LandN, although exact reasons are yet unclear. As in IPSL-CM6A-LR , the eastern equatorial Pacific ocean is too salty compared to the World Ocean Atlas. This could be due to an underestimation of precipitation in the area, which would reduce the dilution effect (Fig.2 (c)). Similarly, positive and negative salinity biases are consistent with precipitation biases, suggesting that SSS
biases could be driven by precipitation.

The Atlantic Meridional Overturning Circulation (AMOC) cell has a very similar latitudinal extent and a maximum around 40°N but its strength is lower than for IPSL-CM6A-LR (Fig.3 (d) and Fig.A8 (d)). The sign of the AMOC stream function changes around 2200m depth while it changes around 2500m for IPSL-CM6A-LR. In the short observational dataset available, this change is diagnosed to occur around 4500m. This shallow AMOC cell is a known bias of the IPSL model (Boucher et al.,
2020). The maximum mixed layer depth (MLD) is maximum in the Labrador and Nordic seas, indicating areas of dense water production (Fig.3 (c) and Fig.A8 (c)). The location of the MLD maxima is consistent with observations in spite of a large variability among members (Boucher et al., 2020). The MLD of IPSL-Perm-LandN is shallower than that of IPSL-CM6A-LR, which is consistent with a weaker AMOC and suggests a reduced production of dense water in the northern North Atlantic.





The March sea ice extent - generally the annual sea ice maximum extent - is overestimated by the IPSL-Perm-LandN when

compared to NSIDC observations (Fig.A9). Over the historical period, the March sea ice extent decreases from 20.3 Mkm$^2$ (1850-1900) to 16.7 Mkm$^2$ (2005-2014), while observations show a slower decrease over the last decades and yet a weaker total sea ice extent of 14.8 Mkm$^2$ (2005-2014). In the last years of the historical simulation, the model comes closer to the satellite observations. On the contrary, the March sea ice extent was very close to the observations in IPSL-CM6A-LR. Almost all of the difference is explained by the presence of sea ice at the Labrador sea-Atlantic junction in winter with fractions close

to 1 in IPSL-Perm-LandN, while this area is almost ice-free in IPSL-CM6A-LR (Fig.A9 (a) and (b)). This is consistent with the strong cold SST bias, the strong reduction of MLD in the Labrador sea and the weakening of the AMOC previously observed in IPSL-Perm-LandN. The annual minimum sea ice area (in September) is also slightly overestimated by IPSL-Perm-LandN, but less than for winter sea ice. The decreasing trend in the simulations is consistent with observed trends.

### 4.3   Permafrost physics

In IPSL-Perm-LandN, the permafrost region covers 16.5 Mkm$^2$ at the end of the historical simulation (2005-2014) (Fig.4 (a)). This is higher than the ESA-CCI mean permafrost area (regridded to the resolution of IPSL-Perm-LandN) (14.0 Mkm$^2$), but just below the upper limit of uncertainty, and lower than the ESA-CCI permafrost region (mean 19.3 Mkm$^2$). This was expected as the ESA-CCI permafrost area represents the area underlain by permafrost, that the model cannot represent and which is smaller than the permafrost region. In addition, as the ESA-CCI permafrost region is the region covered by all permafrost

zones, it results in a larger estimate than the models that cannot capture sporadic permafrost and isolated patches. However, the simulated permafrost region is slightly higher than the ESA-CCI continuous and discontinuous permafrost region (permafrost fraction>50%, mean 14.17 Mkm$^2$) that the model is expected to simulate, mainly due to overestimated permafrost extent over the Tibetan Plateau. The modeled permafrost region is also within the range of C4MIP models estimates, although they have not been regridded and the permafrost representation of each model is superimposed to the effect of its spatial resolution.

Higher resolution models should, in principle, be closer to observations as they capture finer permafrost patterns. Notably, there is a clear improvement in the representation of permafrost compared to IPSL-CM6A-LR which had an extremely small permafrost region. This is mainly due to the inclusion of the latent heat of soil water phase change in IPSL-Perm-LandN. Its absence in IPSL-CM6A-LR resulted in overestimated ALT and underestimated permafrost region (Steinert et al., 2024). In the recent period, the permafrost region is very close for all three simulation members, with only small differences at the

southern permafrost edges (Fig.4 (b)). Overall, there is a very good agreement between IPSL-Perm-LandN and the ESA-CCI product (permafrost fraction>50%). In Eurasia, the permafrost region compares well with the 50% permafrost contour from ESA-CCI observations, with a slight overestimation over the southern boundary. As expected, the model also predicts too much permafrost over the Tibetan Plateau, which has a known cold bias in surface air temperature (Boucher et al., 2020). In North America, simulated permafrost in IPSL-Perm-LandN is present in the north, but is absent at the southern edge, in Canada,

which is a known bias in many CMIP6 models (Burke et al., 2020). Overall, the permafrost region has decreased by 2.4 Mkm$^2$ (-15.0%) over the historical period compared to 1850-1900.





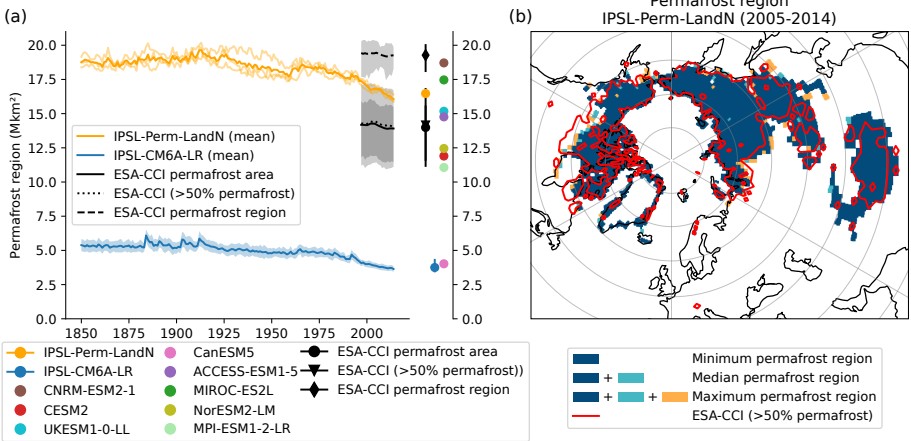

**Figure 4. Historical permafrost region**. **(a)** Permafrost region in the northern hemisphere over the historical period for IPSL-Perm-LandN, IPSL-CM6A-LR and C4MIP models, and permafrost area and permafrost region for ESA-CCI observation product. Colored dots represent the mean permafrost region (2005-2014) for IPSL-Perm-LandN, IPSL-CM6A-LR and C4MIP models and the mean permafrost area, permafrost region and region of >50% permafrost for ESA-CCI. Light orange lines represent the three historical members for IPSL-Perm-LandN. The light blue envelope corresponds to one standard deviation between members of IPSL-CM6A-LR. **(b)** Map of the permafrost region in IPSL-Perm-LandN (2005-2014). Dark blue : all three members diagnose permafrost. Light Blue : two members diagnose permafrost. Orange : only one member diagnoses permafrost. Red contour : 50% permafrost fraction (continuous and discontinuous) from ESA-CCI.

The simulated mean ALT is in good agreement with CALM observations in eastern and northern Canada, and northern and eastern Siberia (Fig.5). However, it is too deep in western Siberia, western Alaska and along the MacKenzie river in western Canada. It also compares well to the ESA-CCI product over most of the permafrost region. At the southern edge of
Canadian permafrost, there is no permafrost in IPSL-Perm-LandN and the ALT is unsurprisingly too deep. Within the modeled permafrost region, the simulated ALT is also too deep in Western Alaska and Western Siberia, the latter being partly due to the underestimation of ALT in this area by the ESA-CCI product (Fig.A10).

## 4.4 Global land carbon cycle dynamics

### 4.4.1 Growth Primary Production (GPP)

On a global scale, gross primary production (GPP) increases slowly until the 1960's and much faster thereafter (Fig.6 (a)). As in other ESMs, this bent curve is mainly driven by the fertilisation effect caused by an increase in anthropogenic $CO_2$ emissions (Piao et al., 2009; Schimel et al., 2015) as well as increased nitrogen atmospheric deposition and fertilisation (Huntzinger et al., 2017; O'Sullivan et al., 2019). The change in the slope around the 1960s is more pronounced than in IPSL-CM6A-LR, primarily driven by the explicit representation of the nitrogen cycle in IPSL-Perm-LandN and its effect in the tropics and mid-latitudes.
In IPSL-Perm-LandN, the global GPP reaches 132 PgC/yr in the last decade of the simulation, higher than estimates from Jung et al. (2020) but within the range of C4MIP ESMs, although there is a large variability across models. GPP is overestimated in





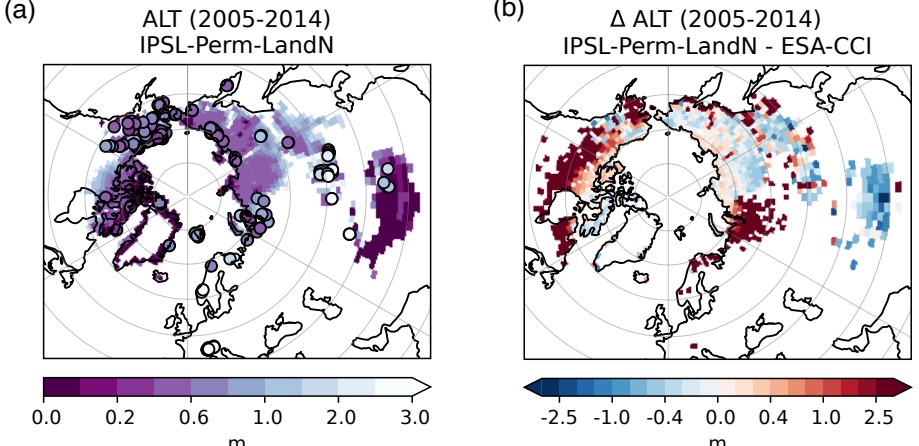

**Figure 5. Active layer thickness (2005-2014). (a)** Background : map of ALT for IPSL-Perm-LandN (2005-2014). Colored circles : CALM observations. **(d)** Background : difference of ALT between IPSL-Perm-LandN and ESA-CCI (2005-2014).

the Arctic and mid-latitudes compared to data-driven products, and within the observational range in the tropics (Fig.6 (b) and Fig.A11 (a)). Compared to IPSL-CM6A-LR, GPP has largely increased in the Arctic (+3.3 PgC/yr) and mid-latitudes (+15.4 PgC/yr), and decreased in the tropics (-10.1 PgC/yr), resulting in an overall global increase of 8.6 PgC/yr. The seasonal cycle

was improved in the tropics compared to IPSL-CM6A-LR, with a seasonality closer to data-driven estimates (Fig.A11 (a)). In the northern mid-latitudes, the shape of the seasonal cycle is consistent with the observations but its amplitude is too large. IPSL-Perm-LandN captures the onset of vegetation growth well, but overestimates GPP during the summer peak and vegetation senescence in autumn. In contrast, IPSL-CM6A-LR was very close to data-driven products throughout the year. In the Arctic, the model overestimates the amplitude of the seasonal cycle, but also shows a delayed decrease in GPP in late summer and

autumn. This was already the case for IPSL-CM6A-LR and is partly due to a warm autumn bias in the Arctic which allows vegetation to survive later in the season (Fig. A12 and A13). These differences are explained by the fact that IPSL-CM6A-LR has been largely tuned using different data sources (FLUXNET, atmospheric $CO_2$, NDVI, Peylin et al., 2016), while the new model including the nitrogen cycle has not been extensively calibrated.

### 4.4.2 Soil heterotrophic respiration (RH)

Soil heterotrophic respiration (RH) follows the same bent shape as GPP over the historical period (Fig.7). This was expected as enhanced GPP leads to increased litter and soil carbon, resulting in higher RH. In the last decade of the simulation, RH reaches 47.4 PgC/yr, close to IPSL-CM6A-LR (45.5 PgC/yr) and data-driven products (49.1 PgC/yr for Bond-Lamberty and Thomson (2010) and 51.9 PgC/yr for Hashimoto et al. (2015)). Similar to IPSL-CM6A-LR, IPSL-Perm-LandN is one of the ESMs with the globally simulated RH value that is closest to estimates over the recent period (Guenet et al., 2024). However,

even if the global RH is close to IPSL-CM6A-LR, the use of a discretised soil carbon profile, the inclusion of permafrost and of an explicit nitrogen cycle in IPSL-Perm-LandN leads to very different regional RH patterns. As with GPP, RH has increased in





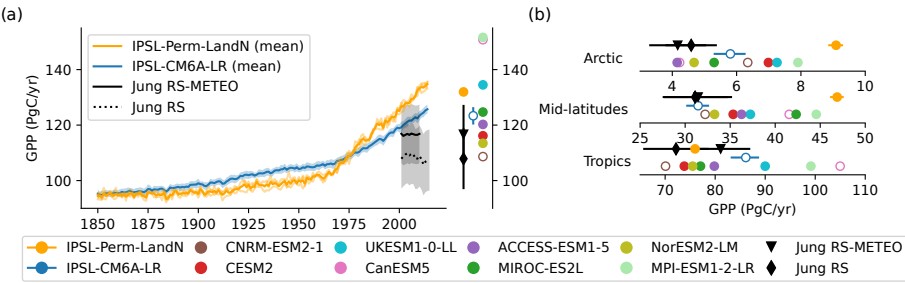

**Figure 6. GPP over the historical period**. **(a)** Global GPP over the historical period for IPSL-Perm-LandN, IPSL-CM6A-LR, C4MIP models, Jung-RS and Jung-RSMETEO observation products (Jung et al., 2020). Colored dots represent the mean GPP (2005-2014) for IPSL-Perm-LandN, IPSL-CM6A-LR, C4MIP models and observations products. Plain (resp. empty) circles represent models with (resp. without) an explicit nitrogen cycle. Light orange lines represent the three historical members for IPSL-Perm-LandN. The light blue envelope corresponds to one standard deviation between members of IPSL-CM6A-LR. **(b)** Total GPP (2005-2014) over the Arctic (>60°N), mid-latitudes (30°S-60°S and 30°N-60°N) and the tropics (30°S-30°N) for IPSL-Perm-LandN, IPSL-CM6A-LR and C4MIP models.

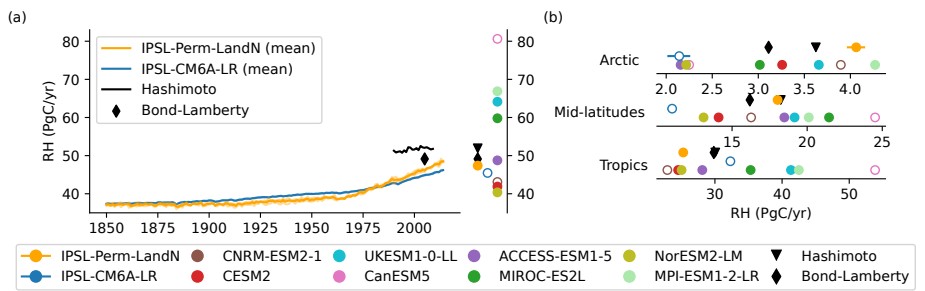

**Figure 7. Soil heterotrophic respiration over the historical period**. **(a)** Global soil heterotrophic respiration (RH) over the historical period for IPSL-Perm-LandN, IPSL-CM6A-LR, C4MIP models, Bond-Lamberty and Thomson (2010) and Hashimoto et al. (2015) observation products. Colored dots represent the mean RH (2005-2014) for IPSL-Perm-LandN, IPSL-CM6A-LR, C4MIP models and observations. Plain (resp. empty) circles represent models with (resp. without) an explicit nitrogen cycle. Light orange lines represent the three historical members for IPSL-Perm-LandN. The light blue envelope corresponds to one standard deviation between members of IPSL-CM6A-LR. **(b)** Total RH (2005-2014) over the Arctic (>60°N), mid-latitudes (30°S-60°S and 30°N-60°N) and the tropics (30°S-30°N) for IPSL-Perm-LandN, IPSL-CM6A-LR and C4MIP models.

the Arctic and mid-latitudes, and decreased in the tropics compared to IPSL-CM6A-LR. The modeled RH is in good agreement with the Hashimoto and Bond-Lamberty products globally, but is slightly overestimated over forests (Fig.A14). In tropical and mid-latitude grassland ecosystems, RH tends to be underestimated. Both GPP and RH show the same regional biases when

confronted with independent observational products, reinforcing our confidence in these results.





### 4.4.3 Net land-atmosphere carbon flux (NBP)

The net land-atmosphere carbon flux (NBP, positive for land uptake), including land-use change emissions ($E_{LUC}$), is negative until the 1970's, mainly because of the negative contribution of land-use change (Tharammal et al., 2019). Thereafter, NBP increases, driven by $CO_2$ fertilisation and nitrogen fertilisation to reach 1.83±0.34 PgC/yr in the last decade (2005-2014),

making the land a net carbon sink over the last 50 years (Fig.8 (a)). This value is very close to estimates from the 2023 Global Carbon Budget (1.86±1.13 PgC/yr), which uses offline Dynamic Global Vegetation Models (DGVMs) for the land carbon sink and bookkeeping models for land-use change emissions. The NBP is slightly larger for IPSL-Perm-LandN than IPSL-CM6A-LR (1.52±0.78 PgC/yr) due to increased net carbon uptake in the Arctic and mid-latitudes, while the tropical NBP remains similar (Fig.8 (b)). The inverse modeling approach used by the Copernicus Atmosphere Monitoring Service (CAMS) shows

a higher global NBP (2.89 PgC/yr) and a different latitudinal distribution. This is due to the fact that atmospheric inversions account for lateral carbon fluxes (between the land and the ocean) while land surface models (and hence ESMs) do not. Subtracting the contribution of lateral fluxes from the inversions generally reconciles both approaches and leads to comparable values of NBP (Ciais et al., 2021). In the tropics, the CAMS product diagnoses a net carbon source while all ESMs rather show a positive to near-neutral NBP. At mid- and high-latitudes, the NBP is positive and much larger in the inversion than in

models, indicating a large net carbon sink that more than compensates for the tropical net carbon source. Such discrepancies between models and inversions are a known knowledge gap and an area of active research (Friedlingstein et al., 2023; Bastos et al., 2020). Recently, the work of O'Sullivan et al. (2024) has shown the key role of forest disturbances at mid-high latitude to reconcile the estimates of the northern carbon sink between atmospheric inversions and DGVMs. The seasonal cycle of NBP for IPSL-Perm-LandN is consistent with that of CAMS despite differences in amplitude (Fig.A15 (a)). In general, IPSL-Perm-

LandN has a smaller amplitude than CAMS in the tropics and a larger amplitude in the extra-tropics. This difference is greater during periods of negative NBP, especially during autumn and winter of the northern hemisphere. Over the last decade, the mean NBP is positive over most of the globe, with the notable exception of regions of high deforestation (eastern and southern Brazil, equatorial African forest, Indonesia) (Fig.A15 (b)). Large sinks are simulated over Europe, Amazonian forest, western African forest, eastern China and the boreal forests of Canada, Alaska and Siberia. By removing the contribution of land-use

change emissions in the NBP, we can estimate the land carbon sink ($S_{LAND}$ in GCB2023), which is positive almost everywhere with deforested areas close to neutrality (Fig.A15 (c)). However, we can only approximate $S_{LAND}$ as it is calculated using fixed pre-industrial vegetation in GCB2023, whereas the vegetation evolves over time in our simulations. Therefore, the large spread in $E_{LUC}$ hinders a more precise assessment of the land carbon sink in our simulations (Bastos et al., 2021; Friedlingstein et al., 2023).

### 650 4.4.4 Land carbon stocks

The global vegetation biomass (above- and below-ground, averaged over 2005-2014) amounts to 479 PgC, which is lower than the ESA-CCI observation-based product (607 PgC, estimated in 2010), mainly due to the lower tropical biomass, and close to the mean of C4MIP models (Fig.9). Compared to IPSL-CM6A-LR, the tropical biomass remained almost unchanged





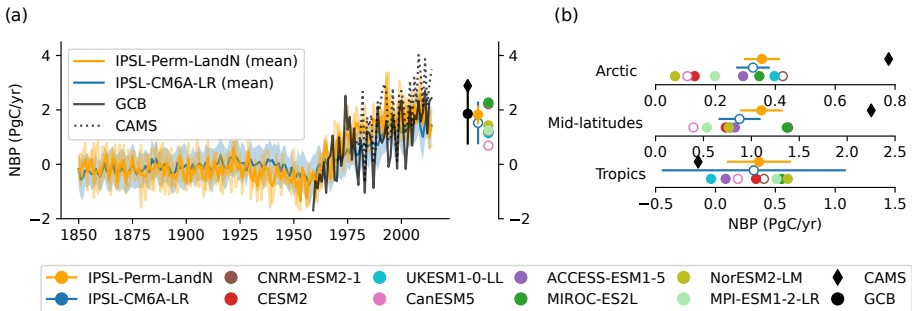

**Figure 8. Net land-atmosphere carbon flux over the historical period**. **(a)** Global net land-atmosphere carbon flux (NBP) over the historical period for IPSL-Perm-LandN, IPSL-CM6A-LR, C4MIP models, CAMS inversion product and the Global Carbon Budget 2023. Colored dots represent the mean NBP (2005-2014) for IPSL-Perm-LandN, IPSL-CM6A-LR, C4MIP models, CAMS and GCB2023. Plain (resp. empty) circles represent models with (resp. without) an explicit nitrogen cycle. Light orange lines represent the three historical members for IPSL-Perm-LandN. The light blue envelope corresponds to one standard deviation between members of IPSL-CM6A-LR. **(b)** Total NBP (2005-2014) over the Arctic (>60°N), mid-latitudes (30°S-60°S and 30°N-60°N) and the tropics (30°S-30°N) for IPSL-Perm-LandN, IPSL-CM6A-LR, C4MIP models and CAMS.

while it doubled at mid- and high-latitudes. Vegetation in the permafrost region remains limited (23 PgC) and smaller than
the ESA-CCI estimate (37 PgC). Comparison with other models is provided for information, but it should be noted that the
permafrost mask used here is that of IPSL-Perm-LandN while the permafrost region may differ between models. The total
amount of litter carbon has remained similar since CMIP6 (108 PgC for IPSL-Perm-LandN and 107 PgC for IPSL-CM6A-LR)
but its distribution has changed, with less carbon in the tropics and more in mid- and high-latitudes. However, there are no
global scale observations and the spread across models is large, making it difficult to assess the performance of IPSL-Perm-
LandN. Finally, IPSL-Perm-LandN simulates a total amount of SOC of 1985 PgC in 0-1m (3001 PgC in 0-3m), distributed
between the tropics (521 PgC for 0-1m, 639 PgC for 0-3m), mid-latitudes (934 PgC for 0-1m, 1376 PgC for 0-3m) and the
Arctic (530 PgC for 0-1m, 985 PgC for 0-3m). Compared to IPSL-CM6A-LR (total SOC of 550 PgC), it has largely increased
in all latitudes, with the highest increases in the mid-latitudes and the Arctic. These changes are due to the discretisation of
SOC along a vertical profile, the initialisation of the soil organic carbon and nitrogen pools by observation-based products and
the representation of an explicit nitrogen cycle. Observed total SOC is 1204 PgC for HWSD and 2498 PgC for SoilGrids in
0-1m (3384 PgC in 0-3m). The large spread across these products and the resulting uncertainty in soil organic carbon content
hampers a constrained assessment of ESMs on a global scale. IPSL-Perm-LandN is naturally closer to SoilGrids, which was
chosen to initialise the SOC and SON pools due to the large number of observations, the robustness of the machine learning
algorithm and the availability of gridded SOC and SON on 6 soil layers. Furthermore, the choice of a product with a high
amount of SOC seems justified as global SOC gridded datasets tend to underestimate SOC content when compared to field
data (e.g. Tifafi et al., 2018). Compared to CMIP6 models contributing to C4MIP (Arora et al., 2020), IPSL-Perm-LandN is the
model with the highest amount of SOC, mainly due to large pools in the mid-latitudes and the Arctic. Permafrost SOC amounts
to 511 PgC in the first meter of soil (1006 PgC in 0-3m) and has largely increased compared to IPSL-CM6A-LR (46 PgC),





which is a significant improvement of IPSL-Perm-LandN. It is again similar to the SoilGrids product (760 PgC in 0-1m, 1028

PgC in 0-3m) and larger than both NCSCD (282 PgC in 0-1m, 668 PgC in 0-3m) and HWSD (186 PgC). It is also very close to

specific estimates of permafrost SOC stocks from Mishra et al. (2021) ($1014^{+186}_{-170}$ PgC) and Hugelius et al. (2014) (1035±150

PgC), both assessing the amount of SOC in the first 0-3m, which is also what IPSL-Perm-LandN aims to represent. However,

IPSL-Perm-LandN does not represent the carbon stored in Yedoma and Arctic river deltas, missing an additional 327-466 PgC

and 96±55 PgC, respectively (Schuur et al., 2022). Deep deposits outside Yedoma or the carbon stored in subsea permafrost

are also not represented by the model, but remain challenging to estimate (Schuur et al., 2022; Sayedi et al., 2020).

## 4.5   Permafrost carbon dynamics

The permafrost region is a carbon sink over the historical period, with a net land uptake of 0.32±0.04 PgC/yr over the last

decade (2005-2014) (Fig.10). The NBP is higher for IPSL-Perm-LandN than IPSL-CM6A-LR (0.24±0.04 PgC/yr), which is

mainly due to differences in the initial state (similar temporal evolution), with IPSL-Perm-LandN being a small carbon sink

and IPSL-CM6A-LR a small carbon source. C4MIP models are divided into three groups and show a wide spread, partly

because the permafrost region may differ between models. A first group shows a small land carbon sink, including CESM2 and

NorESM2-LM, both of which share the land surface model CLM5, as well as CanESM5 which is known to have a small land

NBP (Swart et al., 2019). On the other hand, a second group including UKESM1-0-LL, MIROC-ES2L and CNRM-ESM2-1

has a strong NBP over the permafrost region. IPSL-Perm-LandN belongs to the third group with a moderate permafrost sink,

and which includes MPI-ESM1-2-LR, ACCESS-ESM1-5 and IPSL-CM6A-LR. The net land sink simulated in IPSL-Perm-

LandN contradicts a recent study based on the upscaling of flux measurements, which concludes that the carbon cycle in the

permafrost region is close to neutrality (Ramage et al., 2024). However, the main processes contributing to $CO_2$ emissions in

this study are boreal fires (-0.10 PgC/yr) and carbon losses from rivers (-0.16 PgC/yr), two processes that are not represented

in IPSL-Perm-LandN. In contrast, Ramage et al. (2024) find boreal forests to be the main contributor to carbon uptake with a

net flux of 0.27 PgC/yr, which is close to the NBP of IPSL-Perm-LandN, although the region considered is slightly different.

Therefore, the NBP of IPSL-Perm-LandN could be explained by the lack of important high-latitude $CO_2$-emitting processes

in the model that cannot counterbalance the carbon uptake by boreal forests, which is of the correct order of magnitude.

The persistent carbon uptake in IPSL-Perm-LandN leads to an accumulation of land carbon (+17.0 PgC over the historical

period, Fig.A16 (a) and (b)). Most of this carbon enters the soil - especially the medium SOC pool - and is partly buried by

cryoturbation. Instead, in boreal areas outside the permafrost region, most of the carbon uptake is stored in vegetation (Fig.A16

(f)). The difference with the permafrost region is particularly striking and is likely due to warmer temperatures, increased soil

nitrogen uptake and an abrupt deepening of ALT outside the permafrost region (Fig.A17), and a change in dominant vegetation

type (Fig.A18). The increase in land carbon in the permafrost region over the historical period is also found in the majority of

C4MIP models, except CESM2 and NorESM2-LM which show a net carbon loss, and UKESM1-0-LL and CanESM5 which

show almost no change (Fig.A16 (d)).

The introduction of a vertical discretisation for SOC in IPSL-Perm-LandN allows a better representation of soil carbon

dynamics, especially in permafrost soils. The global SOC profile is very close to observations from SoilGrids below 0.5m but



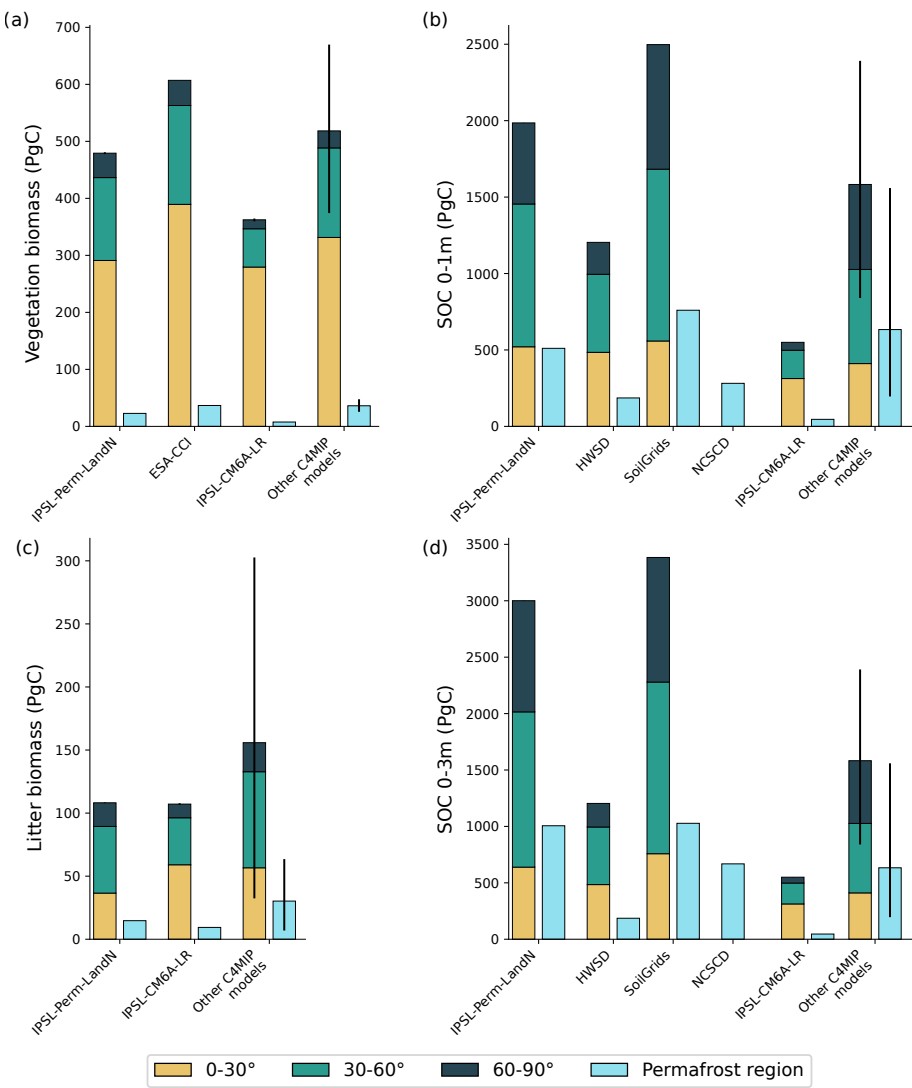

**Figure 9. Carbon stocks at the end of the historical period (2005-2014)**. **(a)** Mean vegetation biomass, **(b)** mean SOC 0-1m, **(c)** mean litter biomass and **(d)** mean SOC 0-3m for IPSL-Perm-LandN, IPSL-CM6A-LR and other C4MIP models over the tropics (30°S-30°N), mid-latitudes (30°S-60°S and 30°N-60°N), the Arctic (>60°N) and the permafrost region. The vegetation biomass observation product is from ESA-CCI and SOC observation products are HWSD, SoilGrids and NCSCD. The other C4MIP models ensemble is composed of CNRM-ESM2-1, CESM2, UKESM1-0-LL, CanESM5, ACCESS-ESM1-5, MIROC-ES2L, NorESM2-LM and MPI-ESM1-2-LR. The error bar shows the full range of C4MIP models.

shows lower soil carbon in the upper 0.5m, probably due to overly high turnover rates of the fast and medium carbon pools (Fig.A19 (a)). The agreement between IPSL-Perm-LandN and SoilGrids at deeper levels is partly due to the model initialisation
by this observation-based product. However, the proportions of fast, medium and slow differ from their initial value and vary



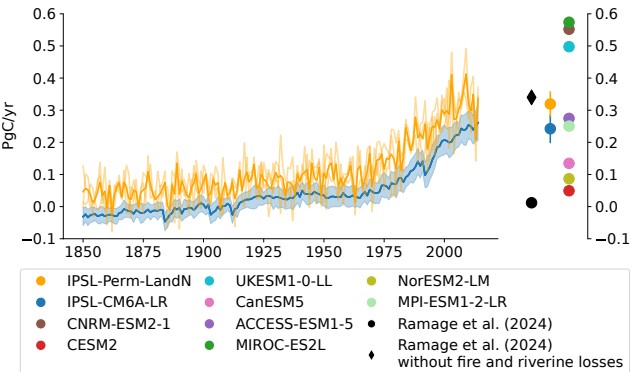

**Figure 10. Permafrost net land-atmosphere carbon flux (NBP) over the historical period** for IPSL-Perm-LandN, IPSL-CM6A-LR and C4MIP models. A positive NBP corresponds to a net land sink. Colored dots represent the mean NBP over the last decade (2005-2014). Black dots correspond to estimates from Ramage et al. (2024). Light orange lines represent the three historical members for IPSL-Perm-LandN. The light blue envelope corresponds to one standard deviation between members of IPSL-CM6A-LR.

with depth, while the total SOC concentration remains close to SoilGrids. In general, surface SOC contains a higher proportion of fast and medium soil carbon, which tends to decrease with depth and to switch to higher slow carbon fractions. This is also consistent with observations showing older carbon in deeper soil layers because of the time required for SOC to be buried by bioturbation or cryoturbation, leaving only the most stable fraction (Balesdent et al., 2018). In the permafrost region, the SOC

vertical profile is flatter than SoilGrids, with less carbon in the upper layers and more at depth (Fig.A19 (b)). As SoilGrids was used to initialise IPSL-Perm-LandN, this simulated SOC profile shows the efficiency of cryoturbation in burying soil carbon and increasing its concentration at depth. Although the absolute SOC concentration is larger, the shape of the SOC profile is closer to NCSCD, which is specifically designed for Arctic regions. Both the observation products and the simulated SOC profile show significant amounts of soil carbon that could lead to carbon emissions as permafrost thaws. In particular, the fast

and medium SOC fractions are larger in the permafrost region than globally and represent a reservoir of reactive carbon on timescales of days to centuries. Finally, the grid boxes of the model can be grouped by classes (bins) of active layer thickness and the mean SOC profile is calculated for each group (Fig.A19 (c)). For regions of shallow ALT (in purple), the ground remains frozen for most of the year, with only surface layers thawing in summer. In this case, the SOC profile is very similar to its initial value as decomposition is almost non-existent. Conversely, in areas of deep ALT, the soil is mainly unfrozen and

the profile is representative of the carbon cycle dynamics of the model. In particular, the profile is flattened compared to the initialisation and shows the effect of cryoturbation, with greater deeper soil SOC concentration. In between, for intermediate ALT, the deep SOC is still close to its initial value while the upper soil responds to the carbon cycle dynamics.



## 4.6 Ocean carbon cycle

The total net ocean-atmosphere carbon flux (*fgco2*) of IPSL-Perm-LandN increases slightly until the 1950's and more rapidly
thereafter, to reach a mean value of 2.16±0.05 PgC/yr over the 2005-2014 period (Fig.A20). This is close to the lower bound
of GCB2023 estimates (2.52±0.4 PgC/yr). *fgco2* is also lower than in IPSL-CM6A-LR (2.55±0.16 PgC/yr over 2005-2014),
due to the effect of the initial state (IPSL-CM6A-LR slightly out of equilibrium with a pre-industrial *fgco2* of 0.25 PgC/yr
compared to 0.045 PgC/yr for IPSL-Perm-LandN). This difference is mainly due to the equatorial oceans with larger $CO_2$
degassing in IPSL-CM6A-LR. The pattern of $CO_2$ fluxes is consistent with observations (e.g. Fay et al., 2024) with degassing
in equatorial ocean and carbon uptake in mid-to high latitudes. Compared to IPSL-CM6A-LR, there is an enhancement of
*fgco2* pattern in the southern mid- and high-latitudes (i.e. larger uptake in areas of $CO_2$ uptake and larger release in areas of
$CO_2$ release). Large compensating differences are also evident in the North Atlantic with a reduced carbon sink in the Labrador
sea and an increased carbon uptake in the Norwegian and Greenland seas. This is broadly consistent with the observed changes
in ocean dynamics in the North Atlantic.

## 740 4.7 Compatible $CO_2$ emissions

After a slow but steady increase from 1850 to 1950, simulated fossil fuel compatible emissions rose much faster during the
second half of the 20[th] century and beyond, reaching 8.3 PgC/yr in the last decade (2005-2014) (Fig.11 (a)). They are very
close to the fossil fuel emissions diagnosed by the Global Carbon Budget 2023 from different emission datasets, except for
the simulated plateau in the 1940s. This plateau is due to the stabilisation of the atmospheric $CO_2$ concentration during this
period (Rubino et al., 2013), leading to a decrease of $G_{ATM}$ and a stagnation of $E_{FF}$. However no plateau is observed in GCB
estimates, which implies a concomitant increase in carbon sinks (Liddicoat et al., 2021). The models do not represent such an
increase and the dynamics of carbon sink in this period is still not fully understood (Bastos et al., 2016). Overall, hypotheses
on the origin of this plateau are a decadal variability in the ocean carbon sink not accounted for in reconstructions, a terrestrial
sink missing from land surface model estimates, or land-use change processes not included in current datasets (Bastos et al.,
2016).

The cumulative compatible fossil fuel emissions of IPSL-Perm-LandN from 1850 to 2014 are 406 PgC, which is very close
to GCB estimates (404 PgC) (Fig.11 (b)). Cumulative compatible emissions are overestimated between 1850 and 1950 but the
plateau in compatible emissions in the 1940s allows GCB estimate to catch up with IPSL-Perm-LandN. Over the second half
of the 20[th] century and the 21[th] century, the model is comparable to GCB. This shape is typical of most of C4MIP models
with a slowdown of the rate of increase of cumulative emissions in the 1940s and an acceleration from the 1960s onwards
(Liddicoat et al., 2021). Cumulative emissions are lower for IPSL-Perm-LandN than for IPSL-CM6A-LR (446 PgC), and most
of this difference results from lower compatible emissions from 1850 to 1950 (mostly due to a lower ocean uptake) and from
a stronger plateau in the 1940s (due to higher land losses).

Inter-model differences in compatible emissions arise from the representation of the land and ocean sinks. The total sink of
IPSL-Perm-LandN over the last historical decade (3.98 PgC/yr) is lower than the mean GCB estimate (4.57 PgC/yr) but within





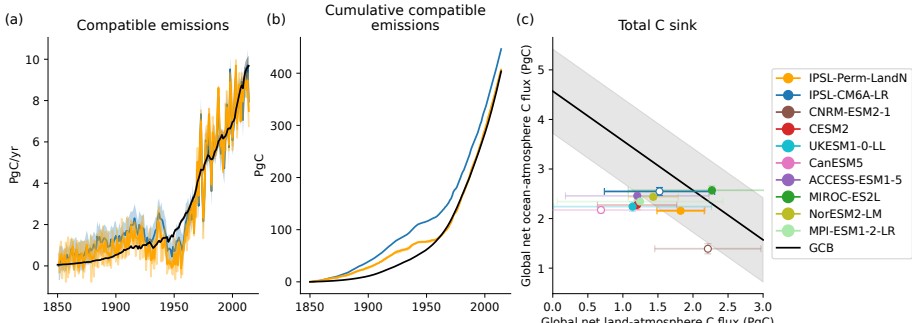

**Figure 11. Historical compatible CO$_2$ emissions**. **(a)** Compatible CO$_2$ emissions over the historical period for IPSL-Perm-LandN, IPSL-CM6A-LR and GCB2023. Light orange lines represent the three historical members for IPSL-Perm-LandN. The light blue envelope corresponds to one standard deviation between members of IPSL-CM6A-LR. **(b)** Cumulative compatible CO$_2$ emissions over the historical period for IPSL-Perm-LandN, IPSL-CM6A-LR and GCB2023. **(c)** Total carbon sink for IPSL-Perm-LandN, IPSL-CM6A-LR, C4MIP models and GCB2023. The land carbon sink is plotted on the x axis and the ocean carbon sink is on the y axis. The black line corresponds to the total CO$_2$ sink estimated by the Global Carbon Budget 2023 (2005-2014). The envelope corresponds to one standard deviation. Plain (resp. empty) circles represent models with (resp. without) an explicit land nitrogen cycle.

its range of uncertainty, with the land and ocean taking up carbon at a similar rate (Fig.11 (c)). Compared to IPSL-CM6A-LR, the ocean sink has been reduced while the land sink has increased, resulting in a comparable total sink in the last decade. Overall, ESMs generally underestimate the total carbon sink, either because of low land or ocean carbon sinks, or both. In particular, CanESM5 is known to have a low land carbon sink (Swart et al., 2019) while CNRM-ESM2-1 has a low ocean

sink, due to a legacy drift in the net air-sea carbon flux from the spinup. A group of models, including CESM2, UKESM1-0-LL, ACCESS-ESM1-5 and MPI-ESM1-2-LR, has moderate land and ocean sinks, resulting in a slightly underestimated total carbon sink. NorESM2-LM, MIROC-ES2L and IPSL-CM6A-LR are within the range of uncertainty of the total carbon sink from GCB. The reasons for the general underestimation of the total sink by ESMs are very model dependent, but the lack of representation of forest dynamics and demography, the representation of land use change and of the nutrient cycles could

explain part of this underestimation (O'Sullivan et al., 2022).

## 5 Conclusions

This work describes IPSL-Perm-LandN, an ESM aiming at better representing the physics and biogeochemistry of high latitudes, and its response to natural and anthropogenic forcings during the historical period. Compared to IPSL-CM6A-LR - the previous version of the model -, the permafrost region has greatly extended and is now close to observations. Soil thermal

dynamics has also improved, as shown by the good agreement of the model's active layer thickness with field measurements. Permafrost now holds much larger amounts of soil organic carbon, with a vertical profile close to observations, which is a prerequisite for assessing future permafrost carbon emissions under climate change. In the historical period, the permafrost

region is a net carbon sink in IPSL-Perm-LandN, whereas more recent estimates rather show a neutral net flux. However, this is consistent with the processes represented in our model, which does not yet include boreal fires and riverine carbon losses.

Overall, the representation of physical and biogeochemical permafrost has greatly improved the response of the model in the Arctic during the historical period.

*Code and data availability.* All model code and data are available on Zenodo (https://doi.org/10.5281/zenodo.16739216; Gaillard et al., 2025a). This DOI contains two files. Gaillard-GMD-2025-Model.tar.gz is the code of the IPSL-Perm-LandN model used to perform all the simulations of this study. Gaillard-GMD-2025-Data.tar.gz contains the model outputs of the three ensemble members for the historical

simulation.

We give in the following more references for the code used. LMDZ, XIOS, NEMO and ORCHIDEE are released under the terms of the CeCILL license. OASIS-MCT is released under the terms of the Lesser GNU General Public License (LGPL). IPSL-Perm-LandN is composed of the following model components (SVN branches and tags):

- LMDZ: LMDZ6/trunk, Tag: 4515

- NEMO: branches/2015/nemo_rev3_6_STABLE/NEMOGCM, Tag: 9455

- ORCA1: trunk/ORCA1_LIM3_PISCES, Tag: 318

- ORCHIDEE: branches/ORCHIDEE_3/ORCHIDEE, Tag: 8336

- IPSLCM6: CONFIG/UNIFORM/v6/IPSLCM6.3, Tag: 6703

- OASIS: CPL/oasis3-mct/branches/OASIS3-MCT_2.0_branch, Tag: 4775

- IOIPSL: IOIPSL/tags/v2_2_5, Tag:6273

- libIGCM: trunk/libIGCM, Tag: 1599

- XIOS: XIOS2/trunk, Tag: 2439

The code modifications made in ORCHIDEEv3 are described in this paper.

*Author contributions.* R.G., P.P., P.C., and B.G. designed the research; R.G. performed the simulations; R.G., P.P., P.C. N.V. and B.G.

analyzed data; R.G. wrote the paper and all authors reviewed the paper and proposed improvements.

*Competing interests.* The authors declare no competing interests.

*Acknowledgements.* This project was provided with computer and storage resources by Grand équipement national de calcul intensif at Très Grand Centre de Calcul du CEA thanks to the Grant 2024-A0140107732 on the supercomputer Joliot Curie's Rome partition. The





IPSL-CM6 team of the IPSL Climate Modeling Centre (https://cmc.ipsl.fr, last access: 15 May 2025) is acknowledged for having developed,
tested, evaluated, and tuned the IPSL climate model IPSL-CM6A-LR. This study benefited from the ESPRI computing and data center (https:
//mesocentre.ipsl.fr, last access: 15 May 2025), which is supported by CNRS, Sorbonne Université, Ecole Polytechnique and Centre national
d'études spatiales, as well as by national and international grants. We acknowledge funding from the European Union's Horizon Europe
research and innovation programme under OptimESM (Grant Agreement No. 101081193). We thank all contributors to the ORCHIDEE-
MICT branch (supervised by Philippe Ciais) for providing the initial parameterization of permafrost. We thank Josefine Ghattas, Frédérique
Cheruy, Vladislav Bastrikov, Dan Zhu, Olivier Torres, Julie Dehayes and Juliette Mignot for their help during model development and
calibration.



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

# Appendix A: Additional description of ORCHIDEE

## A1 Carbon assimilation

Carbon assimilation by photosynthesis is based on the scheme proposed by Yin and Struik (2009), which is an extension
of the model of Farquhar et al. (1980), developed for C3 plants. It calculates carbon assimilation as the minimum of the
rubisco-limited rate of $CO_2$ assimilation and the electron-transport-limited rate of $CO_2$ assimilation. Both the maximum rate of
rubisco-limited carboxylation ($V_{c, max}$, $\mu$mol $CO_2$.m$^{-2}_{[leaf]}$.s$^{-1}$) (i.e. unstressed photosynthetic capacity at optimum temperature)





and the maximum rate of electron-transport under saturated light ($J_{max}$, $\mu$mol e$^-$.m$^{-2}_{[leaf]}$.s$^{-1}$) follow the formulation of Kattge
et al. (2009).

Photosynthetic activity depends on the leaf nitrogen content and can be reduced under nitrogen starvation. Thus, the intro-
duction of an explicit nitrogen cycle allows the model to represent nitrogen limitation of photosynthesis. However, the leaf C:N
ratio is dynamic and varies within a limited range as a function of root nitrogen supply and biomass allocation requirements,
preventing a strict nitrogen limitation. The C:N ratio of all other vegetation nitrogen pools is determined by the leaf C:N ratio
multiplied by a pool-dependent factor.

The dependence of GPP on leaf nitrogen content introduced in IPSL-Perm-LandN replaces the downregulation of maximum
photosynthetic capacity as a function of $CO_2$ used in IPSL-CM6A-LR. In this earlier version of the model, GPP was artificially
reduced at high $CO_2$ concentrations to mimic a nutrient limitation effect. This downregulation mechanism was modeled as a
logarithmic function of the $CO_2$ concentration relative to 380 ppm, following Sellers et al. (1996).

**A2   Carbon allocation**

The allocation of carbon to the different tissues of the plant (leaves, roots, sapwood, heartwood, and fruits) follows the pipe
model theory (Shinozaki et al., 1964), which states that a unit of leaf mass is associated with the downward continuation of non-
photosynthetic tissue that has a constant cross-sectional area (Lehnebach et al., 2018). In other words, the production of one
unit of leaf mass requires a proportional amount of sapwood to transport water and nutrients from the roots to the leaves, and
a proportional amount of roots to take up the water and nutrients from the soil. The allocation scheme dynamically simulates
the leaf area depending on the cost of maintaining a unit leaf area, which takes into account the effects of external stresses
such as water and nitrogen availability. For instance, more carbon is allocated to roots compared to leaves in case of drought,
or nitrogen limitation. The total nitrogen required to sustain the carbon assimilation is then allocated to the different tissues.
If nitrogen uptake is insufficient to sustain the carbon uptake, the leaf C:N ratio of the newly growing tissues increases within
a certain range. If nitrogen is still deficient, carbon uptake is reduced proportionally to match nitrogen availability. Only the
leaf nitrogen concentration is explicitly simulated, while nitrogen is allocated to other tissues in proportion to the leaf nitrogen
content.

**A3   Autotrophic respiration**

Based on Ruimy et al. (1996), autotrophic respiration is divided into maintenance and growth respiration. Maintenance respi-
ration represents the respiration of the biomass already present, and therefore depends on the amount of biomass of each PFT.
It also varies linearly with temperature, with a PFT-dependent coefficient. Maintenance respiration is subtracted from photo-
synthetic carbon assimilation before allocation, up to a certain threshold (80% of GPP). If maintenance respiration is higher
than this threshold, carbon is taken directly from the tissues. Maintenance respiration also increases with the amount of leaf
nitrogen, as in Sitch et al. (2003). In addition, a prescribed fraction of the resulting allocatable carbon (i.e. after maintenance
respiration) is lost through growth respiration, which represents the respiration of newly assimilated carbon. The remaining
carbon after maintenance and growth respiration (i.e. NPP) is allocated to plant tissues.





## A4   Calibration of soil organic matter decomposition and mineral nitrogen losses

When running a spinup under pre-industrial conditions with ORCHIDEE in offline mode, more than 60% of the initial global soil carbon content (initialised with the SoilGrids product) was lost in 2000 years, with significant losses from the high latitudes. Such low soil organic carbon stocks would lead to a low insulation effect and possible underestimation of soil carbon losses under warming. Therefore the decomposition constant of the passive pool - which contains more than 2/3 of the initial carbon - was decreased by a factor of four.

This did indeed reduced soil carbon losses during spinup but also led to the immobilisation of large amounts of nitrogen, eventually resulting in a strong nitrogen limitation of photosynthesis. Global GPP decreased up to 50 PgC.yr$^{-1}$ under pre-industrial conditions, while it was 95 PgC.yr$^{-1}$ for IPSL-CM6A-LR, and no less than 85 PgC.yr$^{-1}$ for the C4MIP models. To reduce the strength of the nitrogen limitation, we increased the soil mineral nitrogen content available for plant uptake by reducing $NH_4^+$ and $NO_3^-$ losses through nitrification and gaseous emissions. This was done by changing the values of the following parameters : N2O_NITRIF_P=0.0004 gN-$N_2$O.gN-$NO_3^{-1}$, NO_NITRIF_P=0.0016 gN-NO.gN-$NO_3^{-1}$, CHEMO_0=19 (unitless), EMM_FAC=0.125 (unitless), CTE_BACT=9.10$^{-5}$ (unitless) and K_NITRIF=1.25 day$^{-1}$. The long turnover times of organic matter prevented a comprehensive statistical optimisation and a manual optimisation of critical parameters had to be performed using a limited number of simulations (with an offline ORCHIDEE configuration). The reduction of mineral nitrogen losses was also motivated by a study showing the overestimation of losses by denitrification in CMIP6 models (Feng et al., 2023).

Overall, with a decreased decomposition constant of the passive carbon and nitrogen pools and reduced mineral nitrogen losses, soil organic carbon and nitrogen losses during the spinup are limited and these pools approach equilibrium faster, while GPP remains close to the value of IPSL-CM6A-LR. Calibration of the model is difficult due to the long turnover times of soil carbon and nitrogen dynamics, and the feedbacks between processes controlling them, and is therefore a source of uncertainty.

## Appendix B:  Equilibrium state after spinup

In this section, we analyse the equilibrium state reached at the end of the spinup, ensuring that climate and carbon cycle drifts are reasonably small. After about 400 years of coupled spinup, the model is considered to be close enough to equilibrium to start historical simulations. The three historical members were started in years 419, 449 and 479 of the coupled spinup, from different phases of the internal variability of the model. The metrics presented in this section are averaged over 150 years surrounding the start years of the historical simulations, to look for potential drifts in the pre-industrial state that could affect these simulations. A detailed description of the model state after spinup can be found in Tab.A3.

The global mean surface temperature (GMST) after spinup is 12.28±0.12°C (mean±std over the three simulation members) and is at equilibrium (trend of +0.0003°C.yr$^{-1}$). IPSL-Perm-LandN is slightly colder than the IPSL-CM6A-LR piControl simulation used for CMIP6, which has a GMST of 12.54±0.12°C, and this is consistent at all latitudes. No significant regional trends are observed, indicating that equilibrium is reached everywhere (not shown). In the permafrost region, the mean tem-





perature is -12.10±0.33°C, significantly colder than IPSL-CM6A-LR (-10.30±0.51)°C. Global mean precipitation (2.95±0.01

mm.day$^{-1}$) and snowfall (0.260±0.004 mm.day$^{-1}$) also show negligible trends.

The global net air-sea carbon flux *fgco2* is 0.045±0.086 PgC/yr (Fig.A21 (a)). This positive value corresponds to a small remaining oceanic carbon sink after the spinup. Although full equilibrium is not reached, the net air-sea carbon flux is much lower than for IPSL-CM6A-LR (0.25±0.09 PgC/yr). The resulting bias in the historical period simulations is consequently one order of magnitude lower than IPSL-CM6A-LR. However this well-balanced global net air-sea flux masks regional variability,

with an oceanic carbon source in the tropics (-1.09±0.04 PgC/yr) that is counterbalanced by carbon sinks in mid- and high-latitudes (1.13±0.07 and 0.01±0.02 PgC/yr respectively). This is an expected behaviour as the large-scale oceanic circulation induces $CO_2$ outgassing in the tropics, and an oceanic $CO_2$ sink in cooling poleward flowing subtropical surface waters as well as in equatorward flowing subpolar surface waters.

The global net land-atmosphere carbon flux (NBP) after the spinup is 0.038±0.510 PgC/yr, which is equal to 1% of the

present-day land carbon sink (Friedlingstein et al., 2023) (Fig.A21 (b)). Over the historical period (1850-2014), this drift is responsible for a cumulative land carbon accumulation of 6.3 PgC, which is negligible compared to land carbon changes during this period. On the contrary, in the IPSL-CM6A-LR piControl simulation, the land is a carbon source with a negative NBP of -0.19±0.64 PgC/yr. The absolute value of NBP is an order of magnitude smaller in IPSL-Perm-LandN, indicating that the model is closer to equilibrium. In addition, the net land-atmosphere carbon exchange is close to equilibrium at all latitudes

with a small carbon sources in the tropics (-0.029±0.440 PgC/yr) and small carbon sinks at mid- (0.044±0.250 PgC/yr) and high latitudes (0.023±0.070PgC/yr). The permafrost region is also a small sink, with a net carbon flux of 0.034±0.48 PgC/yr. Approximately three quarters of the remaining imbalance in the global net land carbon flux is due to a drift in soil carbon *cSoil* and one quarter to a drift in vegetation carbon *cVeg*, both slowly increasing over time. The positive drift in total soil carbon results from opposite trends in the model carbon pools, with the fast and medium soil carbon pools gaining carbon

(resp. +0.004 and +0.06 PgC/yr) and the slow pool losing carbon (-0.03 PgC/yr). Soil organic nitrogen trends are similar for individual pools with the fast and medium pools gaining nitrogen (resp. +0.0002 and +0.001 PgN/yr) and the slow pool losing nitrogen (-0.02 PgN/yr), but resulting in a net soil organic nitrogen loss of -0.02 PgN/yr.

At the end of the spinup, the land stores 3567 PgC, distributed into 485 PgC in vegetation, 100 PgC in the litter and 2982 PgC in the soil (Fig.A22). Most of the vegetation biomass is found in the tropics (302 PgC), followed by mid-latitudes (147

PgC) and the Arctic (37 PgC). Soil organic carbon is divided into 641 PgC in the tropics, 1364 PgC in mid-latitudes and 845 PgC in high latitudes. Most of the soil carbon is found in the so-called slow pool (85%), the rest being stored in the medium pool (14%) and a tiny fraction in the active pool (1%). Compared to IPSL-CM6A-LR, which has a downregulation of GPP with $CO_2$, litter and vegetation stocks are slightly higher but the main difference is the almost 6-fold increase in soil carbon, especially in mid- and high latitudes. In particular, permafrost soil carbon was almost non-existent in IPSL-CM6A-LR and

now amounts to 1006 PgC. Total land nitrogen is 225 PgN, most of which is stored in the soil in organic form (215 PgN). In addition, the vegetation contains 8 PgN, the litter 1 PgN and the soil also stores 1 PgN of mineral nitrogen, which cannot be compared to IPSL-CM6A-LR which did not include a representation of the nitrogen cycle.





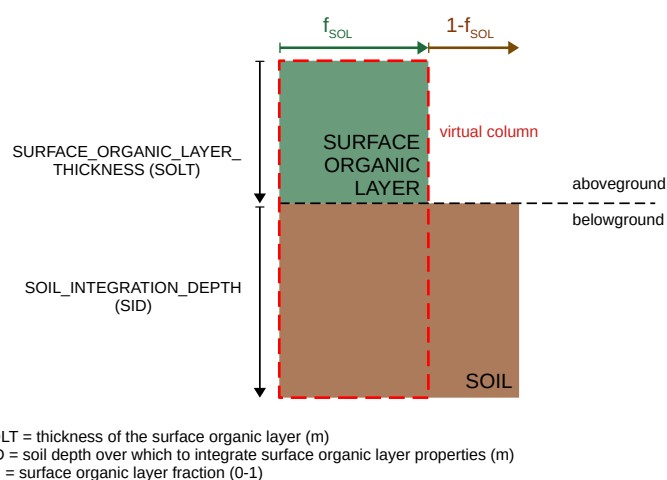

SOLT = thickness of the surface organic layer (m)
SID = soil depth over which to integrate surface organic layer properties (m)
$f_{SOL}$ = surface organic layer fraction (0-1)

**Figure A1. Scheme of the integration of surface organic layer thermal properties.** Soil thermal properties are modified to include a surface organic layer over a fraction fSOL of the soil surface.





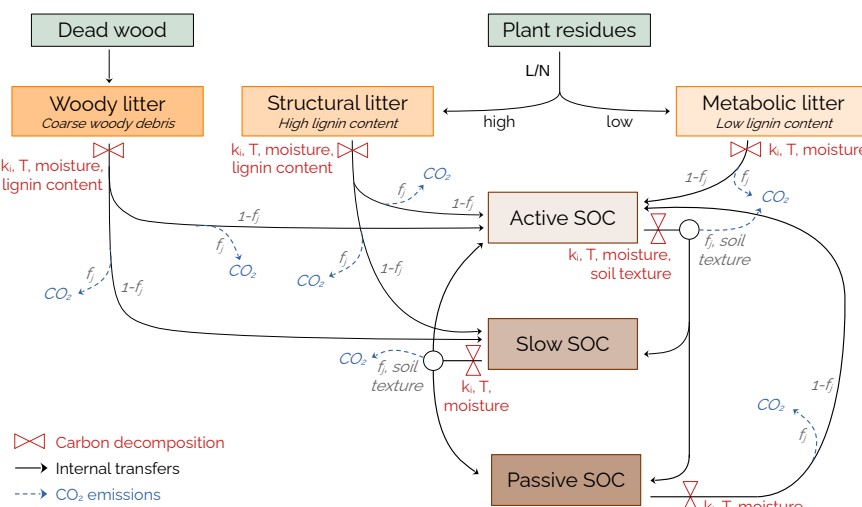

**Figure A2. Schematic of the soil organic carbon dynamics in ORCHIDEE.** Red sandglasses correspond to organic carbon decomposition. The red text shows the associated drivers where the indices i refers to the associated pool. Black arrows show internal organic carbon transfers between pools. Blue arrows show $CO_2$ emissions. Grey text corresponds to the fractions of carbon fluxes that are transferred to another pool or lost as $CO_2$, and depends on a fixed factor $f_j$ (j corresponds to the associated pool) and soil texture. L/N corresponds to the lignin to nitrogen ratio of plant residues. Litter is decomposed into below- and above-ground pools in the model but for clarity, they have been grouped together on this schematic.





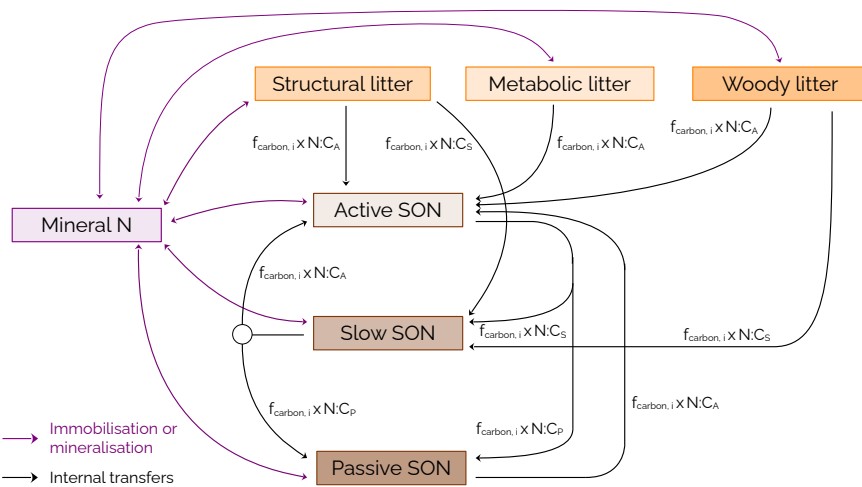

**Figure A3. Schematic of the soil organic nitrogen in ORCHIDEE.** Black arrows show internal organic nitrogen transfers between pools. Purple arrows show exchanges between soil organic and mineral nitrogen pools (immobilisation or mineralisation). $f_{carbon, i}$ is the corresponding carbon flux between the associated soil organic carbon pools and $N:C_X$ is the N:C ratio of the receiving pool, with X=A, S or P (active, slow or passive). Litter is decomposed into below- and above-ground pools in the model but for clarity, they have been grouped together on this schematic.





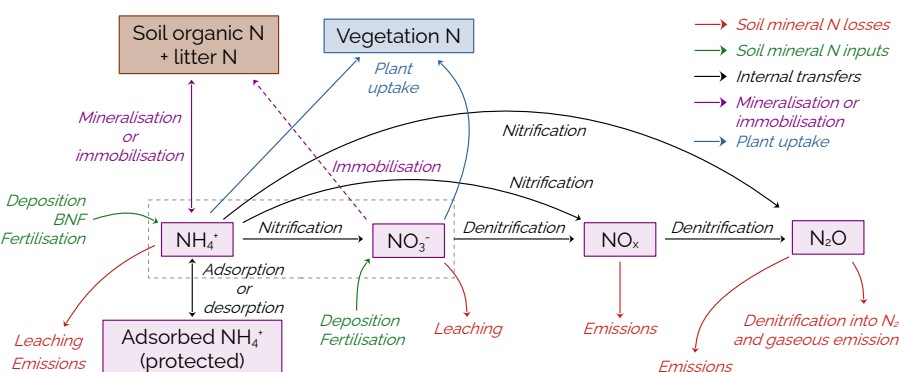

**Figure A4. Schematic of the soil mineral nitrogen dynamics in ORCHIDEE.** Red (resp. green) arrows represent soil mineral nitrogen losses (resp. inputs). Black arrows represent internal land nitrogen transfers. Purple arrows show exchanges between the mineral and organic soil nitrogen pools (immobilisation or mineralisation). The blue arrow shows plant nitrogen uptake. The dashed grey box represents the mineral nitrogen species available for plant uptake ($NH_4^+$ and $NO_3^-$).



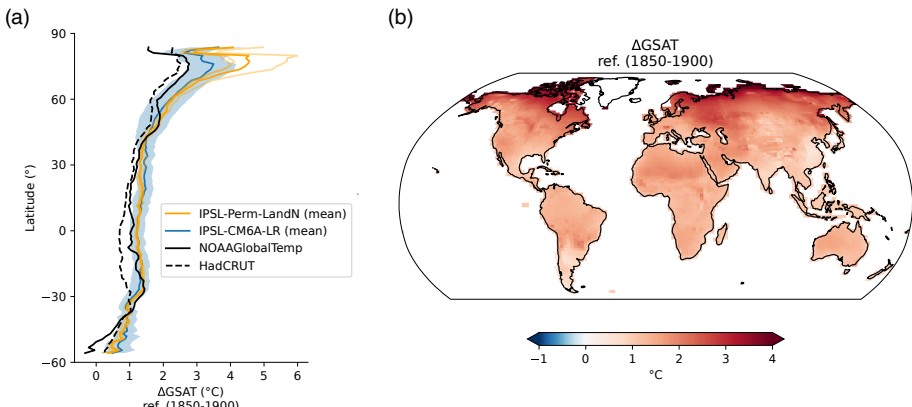

**Figure A5. Historical surface temperature over land**. **(a)** Zonal mean of mean land GSAT anomaly (2005-2014) relative to 1850-1900 for IPSL-Perm-LandN, IPSL-CM6A-LR, NOAAGlobalTemp and HadCRUT. Light orange lines represent the three historical members for IPSL-Perm-LandN. The light blue envelope corresponds to one standard deviation between members of IPSL-CM6A-LR. **(b)** Map of mean land GSAT anomaly (2005-2014) relative to 1850-1900 for IPSL-Perm-LandN. Greenland and Antarctica have been excluded for all panels to only account for non-glaciated land. The products NOAAGlobalTemp and HadCRUT provide global mean surface temperature (GMST) anomaly, defined as land surface air temperature anomaly over land and sea surface temperature anomaly over the ocean. GMST and GSAT differ by at most 10% (IPCC AR6 WGI Chap.2, 2021).



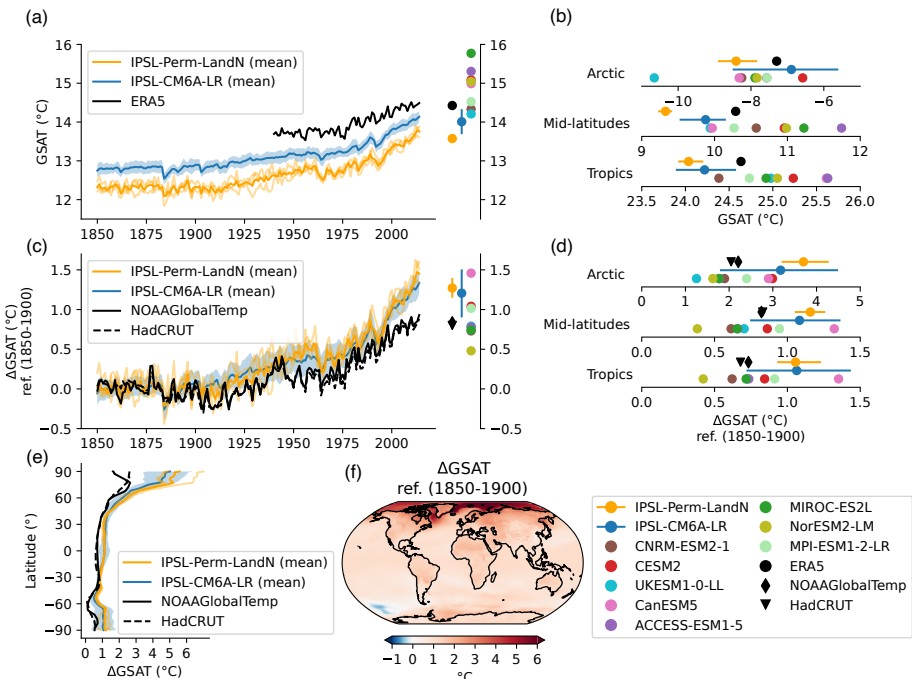

**Figure A6. Historical global surface temperature**. **(a)** Mean global surface air temperature (GSAT) over the historical period for IPSL-Perm-LandN, IPSL-CM6A-LR and ERA5 reanalysis. Colored dots represent the mean GSAT (2005-2014) for IPSL-Perm-LandN, IPSL-CM6A-LR, ERA5 and C4MIP models. Light orange lines represent the three historical members for IPSL-Perm-LandN. The light blue envelope corresponds to one standard deviation between members of IPSL-CM6A-LR. **(b)** Mean global GSAT (2005-2014) over the Arctic (>60°N), mid-latitudes (30°S-60°S and 30°N-60°N) and the tropics (30°S-30°N) for IPSL-Perm-LandN, IPSL-CM6A-LR and C4MIP models. **(c)** Anomaly of mean GSAT relative to 1850-1900 for IPSL-Perm-LandN, IPSL-CM6A-LR, NOAAGlobalTemp and HadCRUT reanalyses. Colored dots represent the mean GSAT anomaly (2005-2014) for IPSL-Perm-LandN, IPSL-CM6A-LR, NOAAGlobalTemp, HadCRUT and C4MIP models. Light orange lines represent the three historical members for IPSL-Perm-LandN. The light blue envelope corresponds to one standard deviation between members of IPSL-CM6A-LR. **(d)** Mean GSAT anomaly over the Arctic (>60°N), mid-latitudes (30°S-60°S and 30°N-60°N) and the tropics (30°S-30°N) for IPSL-Perm-LandN, IPSL-CM6A-LR and C4MIP models. **(e)** Zonal mean of mean GSAT anomaly (2005-2014) relative to 1850-1900 for IPSL-Perm-LandN, IPSL-CM6A-LR, NOAAGlobalTemp and Had-CRUT. Light orange lines represent the three historical members for IPSL-Perm-LandN. The light blue envelope corresponds to one standard deviation between members of IPSL-CM6A-LR. **(f)** Map of mean GSAT anomaly (2005-2014) relative to 1850-1900 for IPSL-Perm-LandN. The products NOAAGlobalTemp and HadCRUT provide global mean surface temperature (GMST) anomaly, defined as land surface air temperature anomaly over land and sea surface temperature anomaly over the ocean. GMST and GSAT differ by at most 10% (IPCC AR6 WGI Chap.2, 2021).



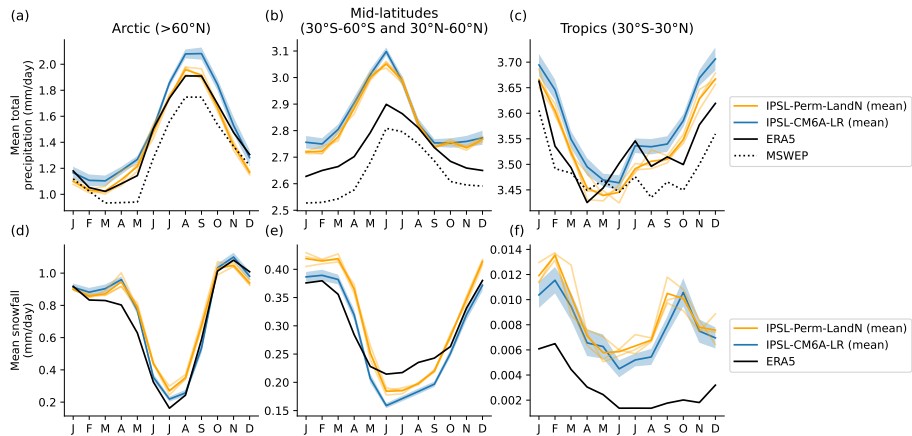

**Figure A7. Seasonal cycle of historical precipitation and snowfall**. Seasonal cycle (2005-2014) of mean total precipitation for IPSL-Perm-LandN, IPSL-CM6A-LR, ERA5 and MSWEP in **(a)** the Arctic, **(b)** mid latitudes, and **(c)** the tropics. Seasonal cycle (2005-2014) of mean snowfall for IPSL-Perm-LandN, IPSL-CM6A-LR, ERA5 and MSWEP in **(d)** the Arctic, **(e)** mid latitudes, and **(f)** the tropics. Light orange lines represent the three historical members for IPSL-Perm-LandN. Light blue envelopes correspond to one standard deviation between members of IPSL-CM6A-LR.

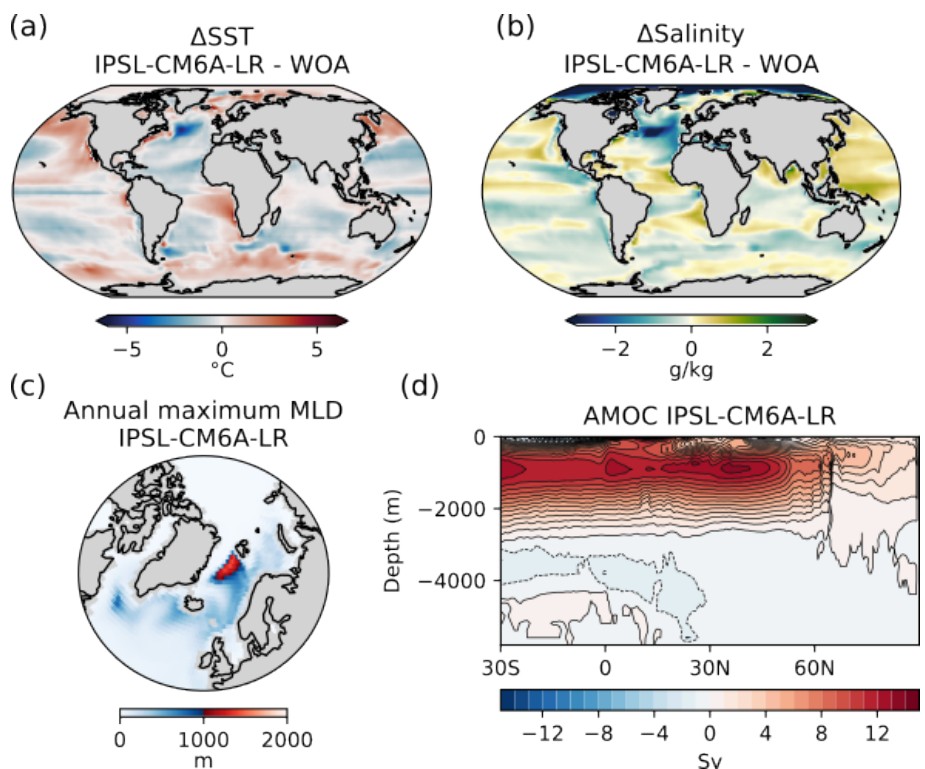

**Figure A8. Historical ocean physics for IPSL-CM6A-LR**. Difference in annual mean sea surface **(a)** temperature and **(b)** salinity between IPSL-Perm-LandN the World Ocean Atlas (2005-2014). **(c)** Mean annual maximum mixed layer depth (2005-2014). **(d)** Atlantic meridional overturning stream function, on average over 2005-2014.



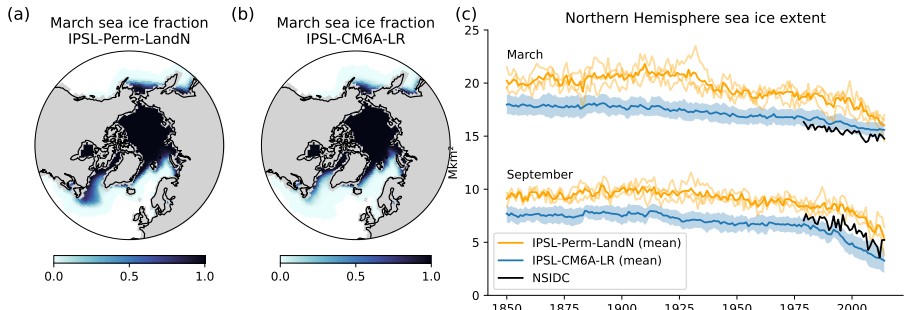

**Figure A9. Historical sea ice**. Mean march sea ice fraction (2005-2014) for **(a)** IPSL-Perm-LandN and **(b)** IPSL-CM6A-LR. **(c)** Time series of sea ice extent (total area enclosed within the 15% sea ice fraction) over the Northern Hemisphere for IPSL-Perm-LandN, IPSL-CM6A-LR and NSIDC observations. The upper and lower curves represent March and September sea ice extents, respectively. Light orange lines represent the three historical members of IPSL-Perm-LandN. The light blue envelope corresponds to one standard deviation between members of IPSL-CM6A-LR.



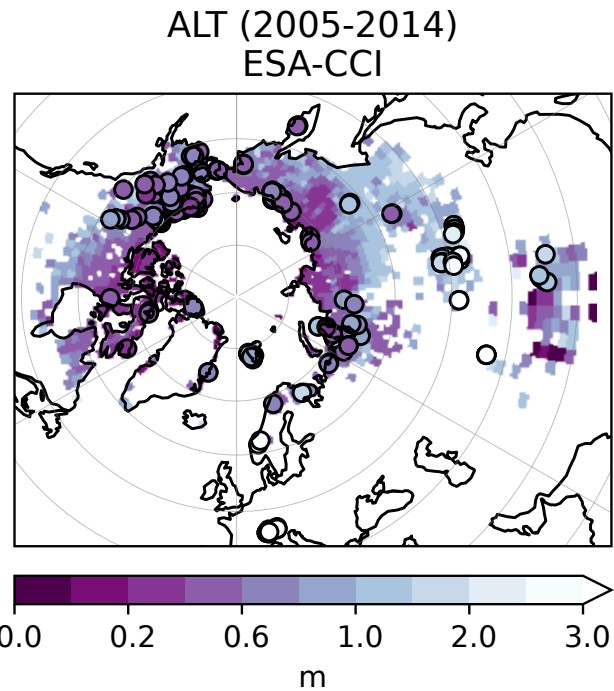

**Figure A10. Active layer thickness for CALM and ESA-CCI (2005-2014)**. Background : map of ALT observation from ESA-CCI (2005-2014). Colored circles : CALM observations.





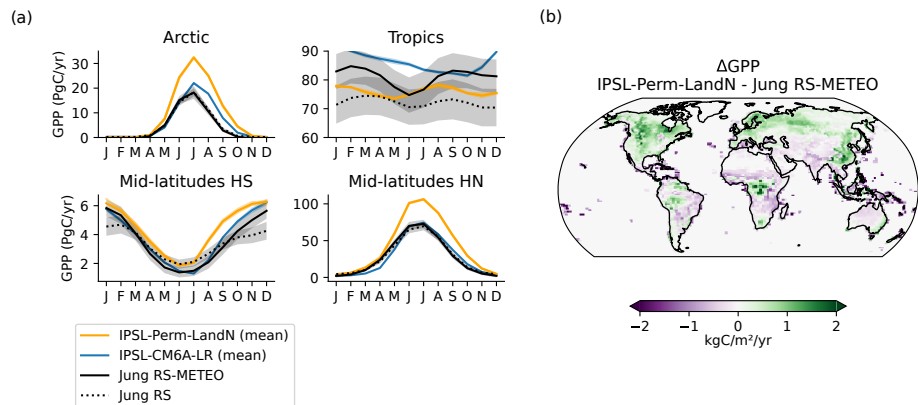

**Figure A11. GPP over the historical period. (a)** Mean seasonal cycle (2005-2014) of total GPP over the Arctic (>60°N), the northern hemisphere mid-latitudes (30°N-60°N), the tropics (30°S-30°N) and the southern hemisphere mid-latitudes (30°S-60°S). Orange: IPSL-Perm-LandN. Blue: IPSL-CM6A-LR. Plain dark: Jung RS-METEO. Dotted dark: Jung RS. **(b)** Mean GPP difference between IPSL-Perm-LandN and Jung-RSMETEO product (2005-2014).





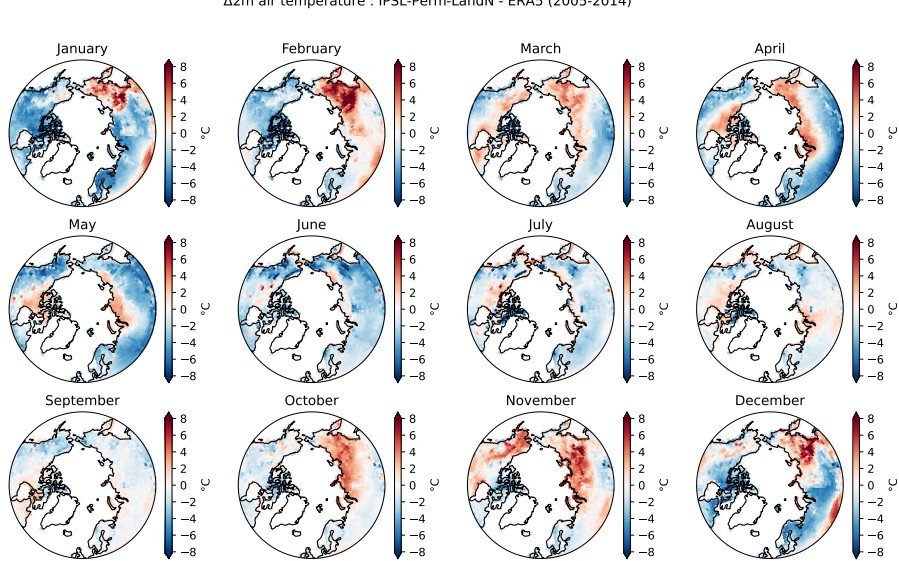

**Figure A12. Arctic surface air temperature bias of IPSL-Perm-LandN (2005-2014).** Mean monthly difference in 2m air temperature between IPSL-Perm-LandN and ERA5 for the period 2005-2014.





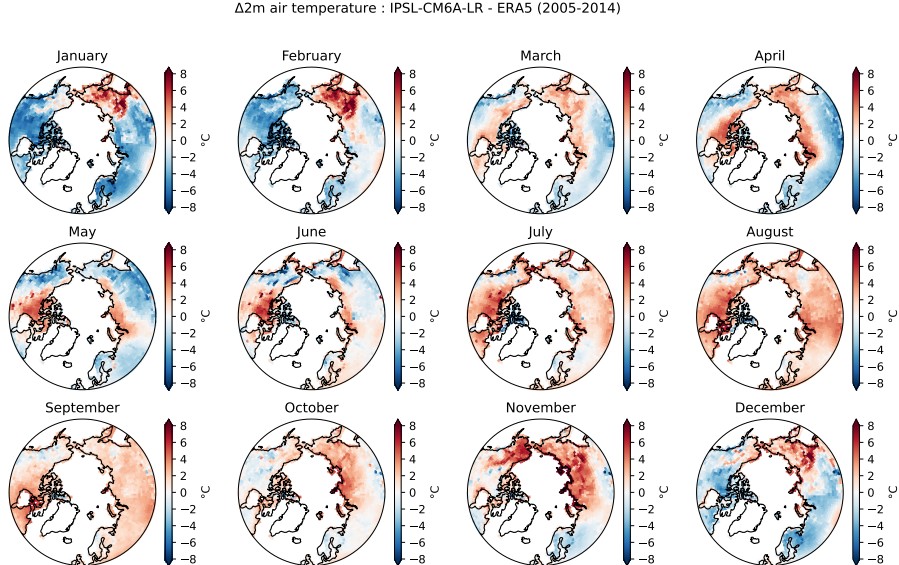

**Figure A13. Arctic surface air temperature bias of IPSL-CM6A-LR (2005-2014).** Mean monthly difference in 2m air temperature between IPSL-CM6A-LR and ERA5 for the period 2005-2014.





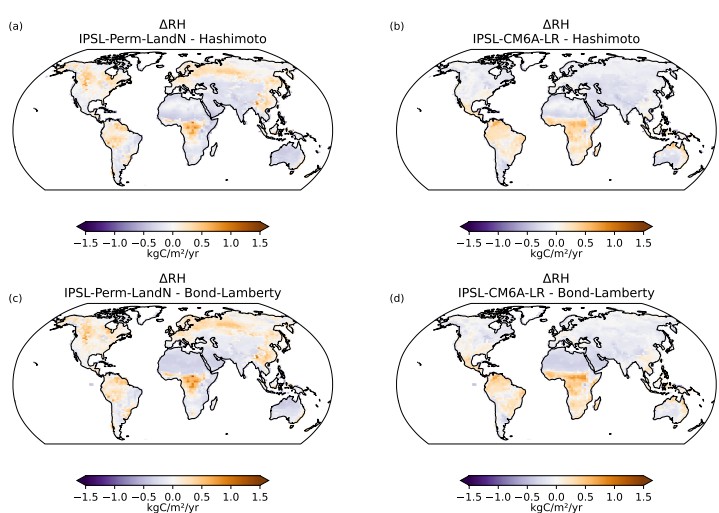

**Figure A14. Maps of soil heterotrophic respiration over the historical period.** Mean RH difference (2005-2014) between IPSL-Perm-LandN and **(a)** the Hashimoto product and **(b)** the Bond-Lamberty product. Mean RH difference (2005-2014) between IPSL-CM6A-LR and **(c)** the Hashimoto product and **(d)** the Bond-Lamberty product.





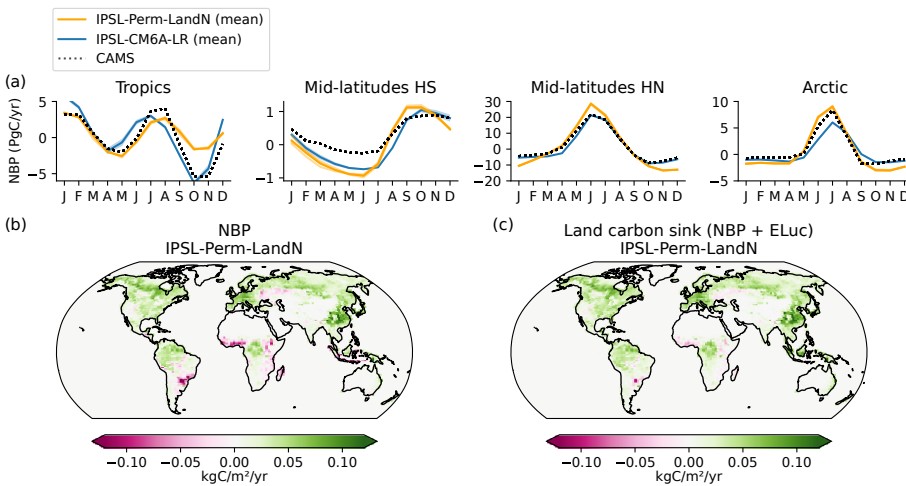

**Figure A15. NBP over the historical period. (a)** Mean seasonal cycle (2005-2014) of total NBP over the tropics (30°S-30°N), the southern hemisphere mid-latitudes (30°S-60°S), the northern hemisphere mid-latitudes (30°N-60°N) and the Arctic (>60°N) for IPSL-Perm-LandN (orange), IPSL-CM6A-LR (blue) and CAMS (2005-2014, dotted dark). **(b)** Map of IPSL-Perm-LandN NBP (2005-2014). It corresponds to $S_{LAND}$-$E_{LUC}$ in GCB2023. **(c)** Land carbon sink (sum of NBP and land use change emissions) for IPSL-Perm-LandN (2005-2014). It corresponds to $S_{LAND}$ in GCB2023.



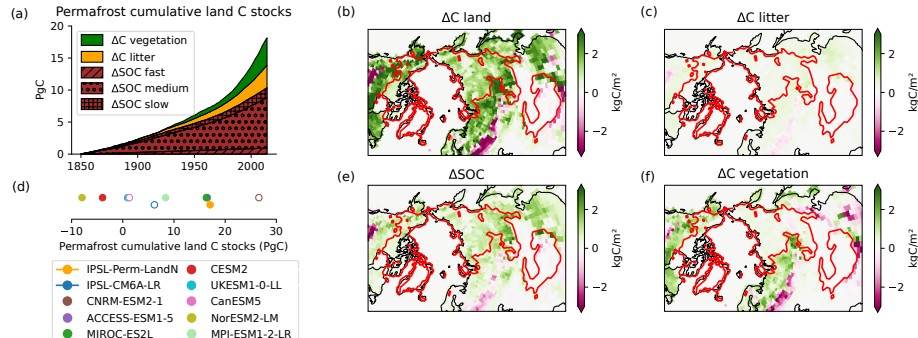

**Figure A16. Changes in permafrost carbon stocks over the historical period. (a)** Permafrost cumulative land carbon stocks since 1850 for IPSL-Perm-LandN over the historical period. A positive value corresponds to a land carbon gain. **(b)** Change in total land C in IPSL-Perm-LandN compared to 1850-1900. A positive value corresponds to a land carbon gain. The red contour shows the limits of the permafrost region in IPSL-Perm-LandN. **(c)** Change in total litter C in IPSL-Perm-LandN compared to 1850-1900. A positive value corresponds to a carbon gain by the litter. **(d)** Permafrost cumulative land C stocks (2005-2014) compared to 1850-1900 for IPSL-Perm-LandN and C4MIP models. **(e)** Change in total SOC in IPSL-Perm-LandN compared to 1850-1900. A positive value corresponds to a SOC gain. **(f)** Change in total vegetation C in IPSL-Perm-LandN compared to 1850-1900. A positive value corresponds to a carbon gain by the vegetation.



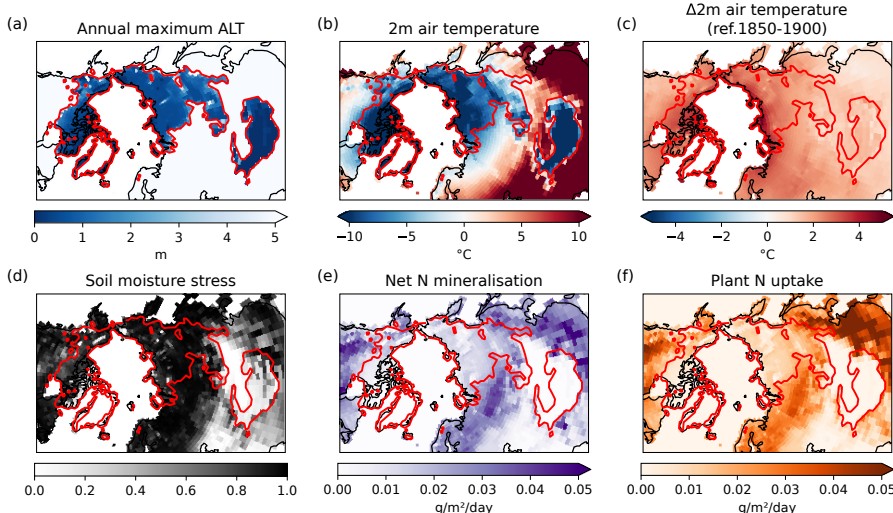

**Figure A17. Physical and biogeochemical potential drivers of vegetation change in IPSL-Perm-LandN (2005-2014)**. **(a)** Annual maximum active layer thickness, **(b)** 2m air temperature, **(c)** change in 2m air temperature compared to 1850-1900, **(d)** soil moisture stress, **(e)** net soil nitrogen mineralisation and **(f)** plant nitrogen uptake, averaged over 2005-2014. The red contour shows the limits of the permafrost region in IPSL-Perm-LandN.





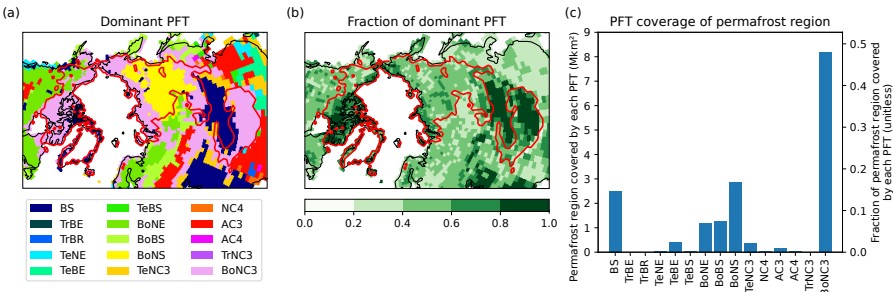

**Figure A18. PFT coverage of the permafrost region** in IPSL-Perm-LandN for the 2005-2014 period. **(a)** Dominant PFT for each grid cell. See Tab.A1 for a description of PFTs. **(b)** Fraction of the grid cell occupied by the dominant PFT. The red contour shows the limits of the permafrost region in IPSL-Perm-LandN. **(c)** Area and fraction of the permafrost region occupied by each PFT.



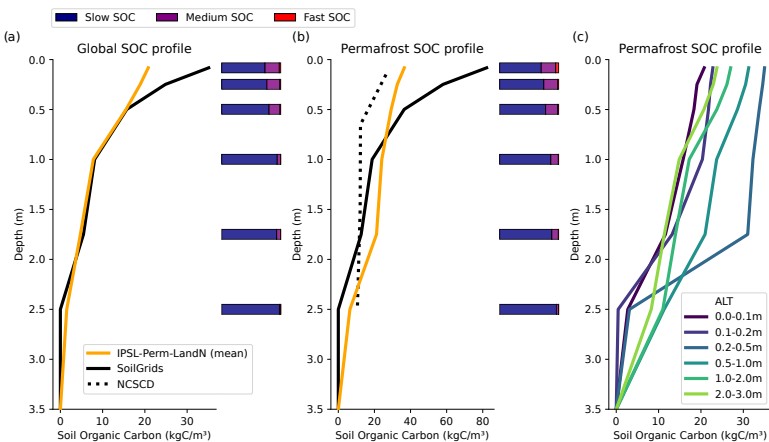

**Figure A19. SOC historical profile**. **(a)** Mean global SOC profile (2005-2014) for IPSL-Perm-LandN and SoilGrids. Horizontal bars represent the proportion of slow (blue), medium (purple) and fast (red) SOC in each soil layer. **(b)** Mean permafrost SOC profile (2005-2014) for IPSL-Perm-LandN, SoilGrids and NCSCD. **(c)** Mean permafrost SOC profile (2005-2014) binned by ALT. For all profiles, the first seven soil layers have been averaged.




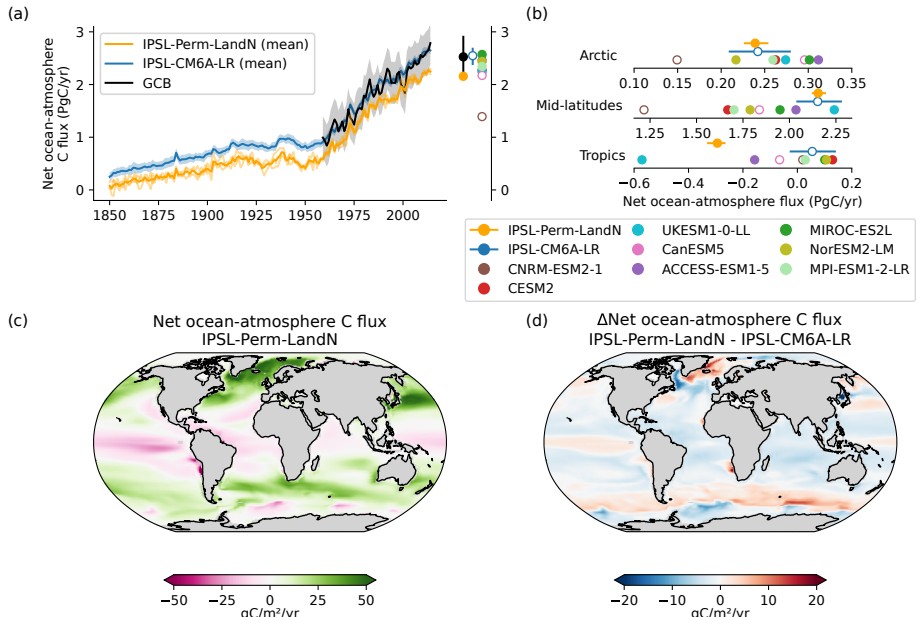

**Figure A20. Net ocean-atmosphere carbon flux over the historical period**. **(a)** Global net ocean-atmosphere carbon flux (*fgco2*) over the historical period for IPSL-Perm-LandN, IPSL-CM6A-LR, C4MIP models and the Global Carbon Budget 2023. Colored dots represent the mean *fgco2* (2005-2014) for IPSL-Perm-LandN, IPSL-CM6A-LR, C4MIP models and GCB2023. Plain (resp. empty) circles represent models with (resp. without) an explicit land nitrogen cycle. Light orange lines represent the three historical members of IPSL-Perm-LandN. The light blue envelope corresponds to one standard deviation between members of IPSL-CM6A-LR. **(b)** Total *fgco2* (2005-2014) over the Arctic (>60°N), mid-latitudes (30°S-60°S and 30°N-60°N) and the tropics (30°S-30°N) for IPSL-Perm-LandN, IPSL-CM6A-LR and C4MIP models. **(c)** Map of IPSL-Perm-LandN mean *fgco2* (2005-2014). **(d)** Difference in mean *fgco2* between IPSL-Perm-LandN and IPSL-CM6A-LR.





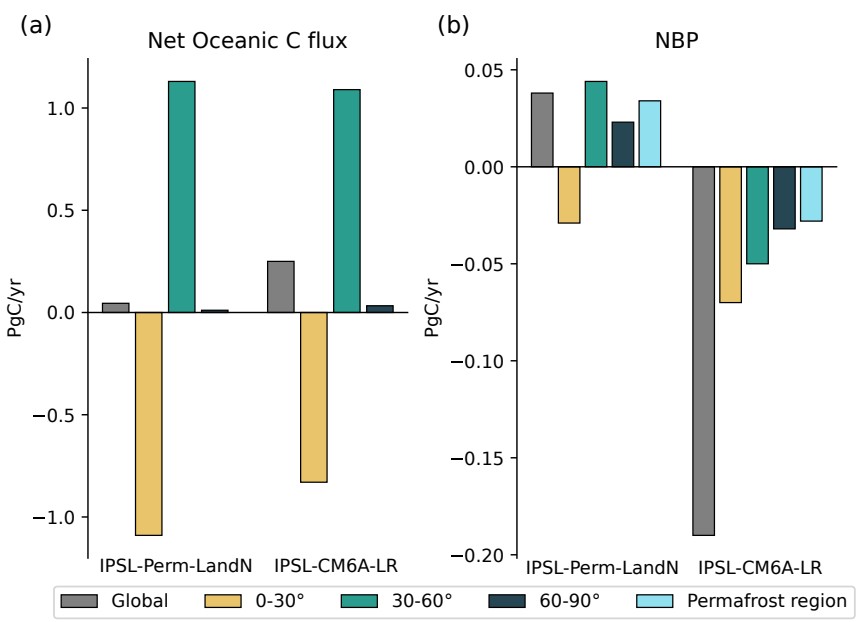

**Figure A21. Ocean and land net carbon fluxes after the spinup**. **(a)** Net sea-air carbon flux, globally and by latitudinal bands. **(b)** Net land-atmosphere carbon flux (Net Biome Production, NBP), globally, by latitudinal bands and over the permafrost region. Fluxes are averaged over 150 years of the piControl simulation surrounding the start years of the historical simulations for IPSL-Perm-LandN, and over the last 150 years of the piControl simulation for IPSL-CM6A-LR. Positive (resp. negative) fluxes correspond to a land or oceanic carbon sink (resp. source).



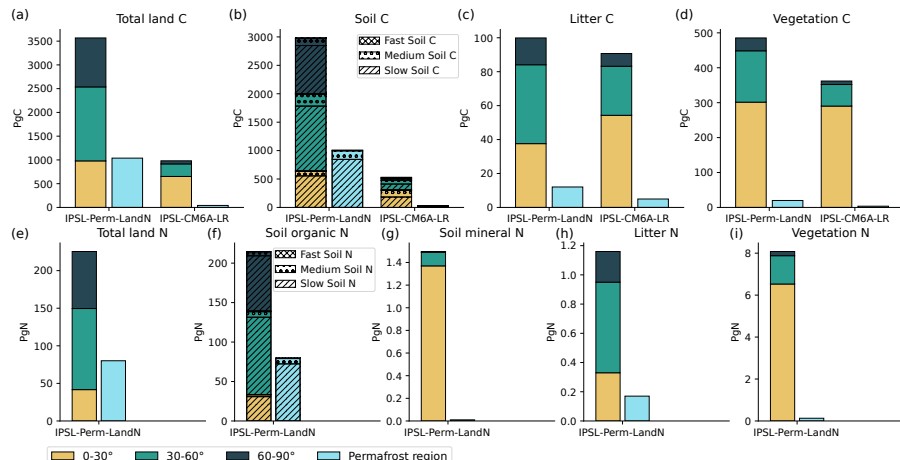

**Figure A22. Land carbon and nitrogen stocks after the spinup**. **(a)** Total land carbon, including soil, litter and vegetation carbon pools. **(b)** Soil carbon. Hatching shows the three soil carbon pools (active, slow, passive) for each latitudinal band and permafrost area. **(c)** Litter carbon. **(d)** Vegetation C. **(e)** Total land nitrogen, including soil organic and mineral nitrogen, litter and vegetation. **(f)** Soil organic nitrogen. Hatching shows the three soil organic nitrogen pools (active, slow, passive) for each latitudinal band and permafrost area. **(g)** Soil mineral nitrogen. **(h)** Litter nitrogen. **(i)** Vegetation nitrogen. Stocks are averaged over 150 years of the piControl simulation surrounding the start years of the historical simulations for IPSL-Perm-LandN, and over the last 150 years of the piControl simulation for IPSL-CM6A-LR. Stocks are given by latitudinal band and over the permafrost area.





**Table A1.** PFTs and their dominant locations in ORCHIDEE.

| PFT number | PFT name | Dominant location |
|---|---|---|
| 1 | Bare Soil | Deserts (Sahara, Australia, Middle East, Gobi) |
| 2 | Tropical Broadleaf Evergreen trees | Tropical South America, Equatorial Africa, Southeastern Asia |
| 3 | Tropical Broadleaf Raingreen trees | Tropical Africa |
| 4 | Tropical Needleleaf Evergreen trees | Japan, North American coasts |
| 5 | Temperate Broadleaf Evergreen trees | China, Southern Brazil, Chile, Australian coasts |
| 6 | Temperate Broadleaf Summergreen trees | Eastern USA, Northern Argentina, Balkans, Zambia |
| 7 | Boreal Needleleaf Evergreen trees | Central Canada, Alaska, Scandinavia, Northeastern Russia |
| 8 | Boreal Broadleaf Summergreen trees | Eastern and Western Russia |
| 9 | Boreal Needleleaf Summergreen trees | Eastern Siberia |
| 10 | Temperate C3 grass | Europe, Central and Western USA, Southern South America, Southern Australia, New Zealand, Central Asia |
| 11 | C4 grass | Southern and Eastern Africa, Southern border of Sahara, Eastern and Western Australia, Western Brazil, Southern USA |
| 12 | Agricultural C3 plants | India, Eastern China, Europe |
| 13 | Agricultural C4 plants | India |
| 14 | Tropical C3 grass | Southeastern Asia, Australia, Western Brazil, Southern border of Sahara |
| 15 | Boreal C3 grass | Northern Canada, Northern Siberia, Tibetan Plateau, Central Asia |



**Table A2.** Soil layer structure. Layer node depth ($z_{n,i}$), thickness ($\Delta z_i$) and depth at layer interface ($z_{l,i}$). All in meter.

| Layer | $z_{n,i}$ | $\Delta z_i$ | $z_{l,i}$ |
|:-----:|:---------:|:------------:|:---------:|
| 1  | 0.0005 | 0.001  | 0.001  |
| 2  | 0.002  | 0.003  | 0.004  |
| 3  | 0.006  | 0.006  | 0.010  |
| 4  | 0.014  | 0.012  | 0.022  |
| 5  | 0.029  | 0.023  | 0.045  |
| 6  | 0.061  | 0.047  | 0.092  |
| 7  | 0.123  | 0.094  | 0.186  |
| 8  | 0.248  | 0.188  | 0.374  |
| 9  | 0.498  | 0.375  | 0.749  |
| 10 | 0.999  | 0.751  | 1.500  |
| 11 | 1.750  | 0.500  | 2.000  |
| 12 | 2.500  | 1.001  | 3.001  |
| 13 | 3.501  | 1.501  | 4.502  |
| 14 | 5.503  | 3.003  | 7.505  |
| 15 | 9.507  | 6.006  | 13.511 |
| 16 | 17.515 | 12.012 | 25.523 |
| 17 | 33.531 | 24.023 | 49.546 |
| 18 | 65.562 | 40.454 | 90.000 |

**Table A3.** Global equilibrium values and trends of main model variables after spinup. Variables are divided into six sections : land carbon fluxes, land nitrogen fluxes, land carbon stocks, land nitrogen stocks, atmospheric physics and oceanic carbon fluxes. IPSL-CM6A-LR values and trends are presented for comparison. Means and standard deviations are calculated over a 150-year period centered on the start of the historical simulations. Trends are calculated by linear regression (least squares method) over the same period. The years chosen as starting points for the historical simulations are shown for completeness.

| Variable (unit) | IPSL-Perm-LandN | | | | | | IPSL-CM6A-LR | | |
| --- | --- | --- | --- | --- | --- | --- | --- | --- | --- |
| | Mean value (unit) | Standard deviation (unit) | Trend (unit/year) | Value at the beginning of historical member (unit) | | | Mean value (unit) | Standard deviation (unit) | Trend (unit/year) |
| | | | | 1 | 2 | 3 | | | |
| NBP (PgC/yr) | 0.038 | 0.51 | -0.0003 | -0.17 | 0.33 | 0.39 | -0.19 | 0.64 | -0.0009 |
| GPP (PgC/yr) | 94.3 | 1.1 | 0.0044 | 92.6 | 94.0 | 94.7 | 94.5 | 1.2 | -0.005 |
| RA (PgC/yr) | 57.0 | 0.62 | 0.0026 | 56.1 | 56.4 | 57.1 | 55.1 | 0.6 | -0.003 |
| RH (PgC/yr) | 37 | 0.32 | 0.021 | 36.4 | 37 | 37 | 37.2 | 0.29 | -0.001 |
| fVegLitter (PgC/yr) | 37 | 0.29 | 0.002 | 36.5 | 36.8 | 37.1 | 37.2 | 0.39 | -0.001 |
| fLitterSoil (PgC/yr) | 17.4 | 0.12 | 0.0008 | 17.1 | 17.4 | 17.4 | 17.7 | 0.15 | -0.0005 |
| fNnetmin (TgN/yr) | 585.4 | 7.6 | 0.038 | 573.4 | 588.8 | 584.3 | | | |
| fNgas (TgN/yr) | 99.3 | 1.78 | 0.008 | 97.0 | 95.9 | 96.6 | | | |
| fNup (TgN/yr) | 531.5 | 6.94 | 0.031 | 518.7 | 536 | 533.6 | | | |
| fNleach (TgN/yr) | 20.4 | 0.63 | 0.0008 | 20.9 | 21.1 | 19.6 | | | |
| fNVegLitter (TgN/yr) | 528.1 | 5.03 | 0.031 | 520.6 | 526.1 | 530.3 | | | |
| fNLitterSoil (TgN/yr) | 1239.2 | 10.7 | 0.036 | 1224.2 | 1242.4 | 1238.1 | | | |
| cVeg (PgC) | 485.5 | 0.9 | 0.01 | 485.5 | 484.2 | 485.5 | 361.7 | 0.73 | 0.001 |
| cLitter (PgC) | 99.8 | 0.35 | 0.004 | 100.1 | 100.2 | 100.0 | 90.8 | 0.24 | -0.0002 |
| cSoil (PgC) | 2982 | 1.4 | 0.03 | 2981 | 2982.5 | 2983.3 | 528 | 0.23 | 0.002 |
| cSoilFast (PgC) | 26.1 | 0.21 | 0.004 | 26.1 | 26.07 | 26.2 | 7.58 | 0.042 | -0.00009 |
| cSoilMedium (PgC) | 413.9 | 2.6 | 0.06 | 411.8 | 414.4 | 416.1 | 204.9 | 0.22 | 0.002 |
| cSoilSlow (PgC) | 2542 | 1.48 | -0.034 | 2543 | 2542 | 2542 | 316 | 0.008 | 0.0001 |



(*continued*)

| Variable (unit) | IPSL-Perm-LandN | | | | | | IPSL-CM6A-LR | | |
| | Mean value (unit) | Standard deviation (unit) | Trend (unit/year) | Value at the beginning of historical member (unit) | | | Mean value (unit) | Standard deviation (unit) | Trend (unit/year) |
| | | | | 1 | 2 | 3 | | | |
| nVeg (PgN) | 8.08 | 0.091 | 0.001 | 8.03 | 8.07 | 8.12 | | | |
| nLitter (PgN) | 1.16 | 0.0043 | 0.00007 | 1.16 | 1.16 | 1.16 | | | |
| nSoil (PgN) | 214.7 | 0.88 | -0.02 | 215.3 | 214.7 | 214.1 | | | |
| nSoilFast (PgN) | 1.68 | 0.012 | 0.0002 | 1.68 | 1.68 | 1.68 | | | |
| nSoilMedium (PgN) | 14.28 | 0.064 | 0.001 | 14.23 | 14.3 | 14.34 | | | |
| nSoilSlow (PgN) | 198.8 | 0.95 | -0.022 | 199.4 | 198.8 | 198.1 | | | |
| nMineral (PgN) | 1.50 | 0.006 | 0.0001 | 1.49 | 1.50 | 1.50 | | | |
| t2m (°C) | 12.28 | 0.12 | 0.0003 | 12.13 | 12.19 | 12.34 | 12.54 | 0.12 | -0.00002 |
| precip (mm/day) | 2.95 | 0.012 | 0.00004 | 2.94 | 2.94 | 2.94 | 2.97 | 0.01 | -0.000003 |
| snow (mm/day) | 0.26 | 0.0042 | -0.000009 | 0.26 | 0.26 | 0.25 | 0.25 | 0.004 | 0.000005 |
| fgco2 (PgC/yr) | 0.045 | 0.086 | -0.00007 | -0.0003 | 0.21 | 0.14 | 0.25 | 0.091 | -0.0001 |

**Table A4.** Equilibrium values and trends of main model variables after spinup in permafrost region. Variables are divided into five sections : land carbon fluxes, land nitrogen fluxes, land carbon stocks, land nitrogen stocks and atmospheric physics. IPSL-CM6A-LR values and trends are presented for comparison. Means and standard deviations are calculated over a 150-year period centered on the start of the historical simulations. Trends are calculated by linear regression (least squares method) over the same period. The years chosen as starting points for the historical simulations are shown for completeness.

| Variable (unit) | IPSL-Perm-LandN | | | | | | IPSL-CM6A-LR | | |
| | Mean value (unit) | Standard deviation (unit) | Trend (unit/year) | Value at the beginning of historical member (unit) | | | Mean value (unit) | Standard deviation (unit) | Trend (unit/year) |
| | | | | 1 | 2 | 3 | | | |
| NBP (PgC/yr) | 0.034 | 0.048 | 0.00004 | -0.0014 | 0.12 | -0.0046 | -0.028 | 0.022 | -0.00002 |
| GPP (PgC/yr) | 3.85 | 0.13 | 0.0005 | 3.83 | 3.96 | 3.78 | 1.73 | 0.075 | -0.0003 |
| RA (PgC/yr) | 1.64 | 0.06 | 0.0002 | 1.65 | 1.68 | 1.63 | 0.9 | 0.004 | -0.0002 |
| RH (PgC/yr) | 2.17 | 0.051 | 0.0002 | 2.17 | 2.16 | 2.15 | 0.85 | 0.019 | -0.0001 |
| fVegLitter (PgC/yr) | 2.2 | 0.054 | 0.0002 | 2.1 | 2.24 | 2.18 | 0.85 | 0.0023 | -0.0001 |
| fLitterSoil (PgC/yr) | 1.03 | 0.02 | 0.00009 | 1.01 | 1.02 | 1.03 | 0.41 | 0.009 | -0.00005 |
| fNnetmin (TgN/yr) | 45.3 | 1.5 | 0.004 | 46.8 | 45.5 | 44 | | | |
| fNgas (TgN/yr) | 10.2 | 0.43 | 0.0005 | 10.1 | 10.3 | 10.2 | | | |
| fNup (TgN/yr) | 36.5 | 1.19 | 0.004 | 37.4 | 36.8 | 35.2 | | | |
| fNleach (TgN/yr) | 1.46 | 0.059 | 0.0002 | 1.5 | 1.46 | 1.51 | | | |
| fNVegLitter (TgN/yr) | 36.7 | 0.082 | 0.003 | 35.8 | 37.1 | 36.7 | | | |
| fNLitterSoil (TgN/yr) | 70.9 | 1.3 | 0.004 | 70.6 | 70.7 | 70.6 | | | |
| cVeg (PgC) | 19.9 | 0.08 | -0.0002 | 19.92 | 19.88 | 19.78 | 3.52 | 0.032 | -0.0006 |
| cLitter (PgC) | 12.05 | 0.05 | 0.0005 | 12.0 | 12.0 | 12.0 | 4.96 | 0.027 | -0.0003 |
| cSoil (PgC) | 1005.8 | 1.43 | 0.03 | 1004.8 | 1005.8 | 1006.8 | 33.4 | 0.04 | -0.0009 |
| cSoilFast (PgC) | 11.1 | 0.13 | 0.003 | 11 | 11 | 11.1 | 0.6 | 0.006 | -0.00003 |
| cSoilMedium (PgC) | 149.4 | 1.09 | 0.025 | 148.6 | 149.5 | 150.1 | 12.7 | 0.03 | -0.0007 |
| cSoilSlow (PgC) | 845.4 | 0.21 | 0.005 | 845.2 | 845.4 | 845.5 | 20.2 | 0.0059 | -0.0001 |





*(continued)*

| Variable | IPSL-Perm-LandN | | | | | | IPSL-CM6A-LR | | |
| | Mean value | Standard deviation | Trend | Value at the beginning of historical member (unit) | | | Mean value | Standard deviation | Trend |
| (unit) | (unit) | (unit) | (unit/year) | 1 | 2 | 3 | (unit) | (unit) | (unit/year) |
|---|---|---|---|---|---|---|---|---|---|
| nVeg (PgN) | 0.1 | 0.00099 | 0.00001 | 0.13 | 0.13 | 0.13 | | | |
| nLitter (PgN) | 0.2 | 0.00075 | 0.000008 | 0.17 | 0.17 | 0.17 | | | |
| nSoil (PgN) | 79.8 | 0.024 | -0.0005 | 79.8 | 79.8 | 79.8 | | | |
| nSoilFast (PgN) | 0.8 | 0.0077 | 0.0001 | 0.75 | 0.75 | 0.75 | | | |
| nSoilMedium (PgN) | 6.8 | 0.019 | 0.0004 | 6.73 | 6.75 | 6.76 | | | |
| nSoilSlow (PgN) | 72.3 | 0.05 | -0.001 | 72.4 | 72.3 | 72.3 | | | |
| nMineral (PgN) | 0.010 | 0.0003 | 0.000003 | 0.010 | 0.011 | 0.010 | | | |
| t2m (°C) | -12.1 | 0.33 | -0.0004 | -11 | -12.4 | -11.8 | -10.3 | 0.51 | -0.002 |
| precip (mm/day) | 1.37 | 0.032 | 0.00004 | 1.36 | 1.33 | 1.37 | 1.52 | 0.004 | 0.000001 |
| snow (mm/day) | 0.89 | 0.024 | 0.00002 | 0.87 | 0.85 | 0.92 | 0.85 | 0.04 | 0.00004 |