# Peer review of "IPSL-Perm-LandN: improving the IPSL Earth System Model to represent permafrost carbon-nitrogen interactions"

_EGUsphere, 2025_

## Author Response (AR1)

**Author's response**

**Anonymous referee #1**

**General comments**

**This manuscript presents IPSL-Perm-LandN, an improved version of IPSL-CM6A-LR (used in CMIP6), incorporating a representation of key permafrost physical and biogeochemical processes and an explicit nitrogen cycle, for a better representation of high-latitude land ecosystems. The authors present the methodological details on the implementation of the model modifications for the relevant processes. In addition to relevant metrics and evaluations regarding permafrost dynamics and land carbon-cycle dynamics, the manuscript also provides an assessment and evaluation of atmospheric physics (here limited to temperature and precipitation) and ocean physics.**

**The manuscript provides relevant and novel results with regard to permafrost physics and carbon dynamics in historical simulations based on model modifications that address an important research gap. As the majority of CMIP climate models lack an adequate representation of permafrost physics, an assessment of the permafrost carbon-climate feedback was limited in CMIP6 and therefore AR6. The next iteration of climate models is in need of the inclusion of the relevant processes. This manuscript provides a description of including these processes in one of the participating CMIP models, which is not only a step towards a better representation of permafrost in CMIP, but could also serve as a blueprint for other climate models to consider the relevant processes.**

We thank the reviewer for highlighting the relevance of our work within the CMIP context.

**I do have a few questions and comments, which I consider overall minor, that I would like the authors to address. The manuscript is well written, structured and generally reads fluently. While the manuscript is quite long, the structuring and flow of the text help its understanding. However, several of the sentences in the Results Section (often the first sentence of the paragraph) do not explicitly state that they refer to the IPSL-Perm-LandN version (which I assume from the context and interpreting the figures), and therefore leave it up to interpretation for the reader. I suggest to carefully go through the Results Section to add these clarifications. I have indicated a few in my 'Specific Comments', but there may be more occurrences.**

We agree that references to IPSL-Perm-LandN are missing. We have corrected this in the Results section by systematically referring to the name of the models when needed.

**I commend the effort to assemble and present all model results in comparison to observations, and also evaluate the atmosphere and ocean to contextualize the results on permafrost modeling. Beyond that, the manuscript provides a valuable and concisely presented reference for other studies to compare their results to, with respect to both modeling results and observational products.**

We thank the reviewer for this comment.

**The methodological modifications presented here and the assumptions taken generally look good to me. Minor comments towards the methods are also specified in 'Specific Comments'.**

**I appreciate that the simulations presented here are run over the historical period, also because this lends itself to the evaluation with respect to observational products. Nonetheless, the efforts to improve the wider suite of CMIP models with regard to permafrost representation are often made with the motivation to address future changes of the climate. A comment in that direction was made in the, somewhat short, Conclusions Section, but I would appreciate some more discussion on how the improvements made in IPSL-Perm-LandN could/will improve the future assessment of permafrost changes and the associated permafrost carbon-climate feedback. Further, have the authors done any future experiments of note that could be presented here? Or will those be addressed in a separate follow-up paper?**

We thank the reviewer for this suggestion. Further experiments have been performed with IPSL-Perm-LandN, including idealised 1pctCO2 and future SSP simulations. They will be presented in a following paper aiming at disentangling the different feedbacks in the future permafrost carbon cycle response to increasing $CO_2$ emissions.

We have added the following paragraphs to the conclusion : *"The model developments presented in this study are essential for evaluating potential future permafrost physical and biogeochemical changes. In particular, the vertical discretisation of soil carbon and nitrogen and related soil biogeochemical processes enables the assessment of the permafrost carbon-climate feedback associated with gradual thaw in IPSL-Perm-LandN. Such a feedback analysis under future climate change will be conducted in a forthcoming article."* and *"A number of other processes are currently under development, including boreal fire disturbance, peatlands, lake and river biogeochemistry and Arctic vegetation. Medium to long-term developments include the representation of abrupt thaw and associated carbon emissions, excess ice and permafrost small-scale heterogeneity. Collectively, these processes would provide a more comprehensive and realistic picture of future permafrost changes and allow to capture the more complex dynamics of permafrost ecosystems beyond gradual thaw."*

**Will the modifications of IPSL-Perm-LandN be incorporated in the next IPSL model version (i.e., for CMIP7)? And how does this operationally affect the spin-up/initialization strategy presented here, given that soil carbon initialization was done offline here?**

Most of the modifications of IPSL-Perm-LandN will be included in the version of the IPSL ESM for CMIP7 Fast Track (CMIP7-FT), including the latent heat of soil water phase change, soil insulation by soil carbon and surface organic layers, and an explicit nitrogen cycle (which is a significant step forward regarding the representation of land biogeochemistry in the IPSL ESM). However, the vertically-resolved soil biogeochemistry scheme might not be included because it requires a long spin-up phase that is not compatible with the relatively short deadline of the CMIP7-FT simulations (to be completed in 2026). Some developments are ongoing to solve this difficulty and vertically-resolved soil biogeochemistry should be included for the more general phase of CMIP7.

With a single-layer soil carbon scheme, a built-in pseudo-analytical spin-up procedure is used to accelerate this phase, following Lardy et al. (2011).

The following lines have been added to the conclusion: *"Most of the new permafrost processes described in this study will be integrated into the IPSL ESM for CMIP7 Fast Track (CMIP7-FT), including the latent heat of soil water phase change, soil insulation by soil carbon and surface organic layers, and the explicit nitrogen cycle. The vertically-resolved soil biogeochemistry will likely only be included for the broader CMIP7 phase, due to the long spinup required and the time constraints of CMIP7-FT."*

**Going into the manuscript, I expected an evaluation of the effect of the model changes on the permafrost carbon-climate feedback (even in the historical period). However, this assessment is not (directly) possible in the CO2-concentration driven simulations conducted here. What was the reasoning behind choosing to run CO2-concentration driven simulations? And can the results presented in Section 4.7 help to infer any of these results? Equally, with regards to this, the reader may appreciate some more discussion in the Conclusions section.**

We agree that emission-driven simulations would be more relevant for assessing the permafrost carbon-climate feedback as they enable feedbacks between the carbon cycle and climate. However, the emission-driven configuration at the time of the study was in construction and not ready to be used. It will be ready for CMIP7 but will use a previous and simpler land surface model that does not represent permafrost processes nor an explicit nitrogen cycle (ORCHIDEE-v2 as used for CMIP6). This choice has been made mostly because we wanted to have an ORCHIDEE version close to CMIP6 to be able to better understand the effect of the emission driven simulations.

Section 4.7 assesses global compatible emission and could, in principle, be used to infer the strength of the permafrost carbon-climate feedback ($\Delta T$, °C) by multiplying the change in cumulative compatible emissions (EgC) between IPSL-Perm-LandN and IPSL-CM6A-LR by the transient climate response to cumulative emissions (TCRE, °C EgC$^{-1}$) of IPSL-Perm-LandN (assuming negligible change in the carbon-concentration feedback from the inclusion of permafrost). However, differences between IPSL-Perm-LandN and IPSL-CM6A-LR arise from both the inclusion of new permafrost processes and of an explicit representation of the nitrogen cycle. This prevents the direct inference of the permafrost carbon-climate feedback as the response of the tropics and mid-latitudes also differ from IPSL-CM6A-LR and impact the global land carbon sink (Fig.8). However, individual permafrost feedbacks (carbon-concentration, carbon-climate, impact of permafrost-nitrogen coupling) will be evaluated in a following paper following a more robust protocol (modified from the C4MIP protocol).

We have added the following discussion to Section 4.7 : *"In principle, the difference in cumulative compatible emissions (EgC) between IPSL-Perm-LandN and IPSL-CM6A-LR could be multiplied by the transient climate response to cumulative emissions (TCRE, °C EgC$^{-1}$) of IPSL-Perm-LandN to infer the strength of the permafrost carbon-climate feedback ($\Delta T$, °C) (assuming negligible change in the carbon-concentration feedback from the inclusion of permafrost). However, differences between IPSL-Perm-LandN and IPSL-CM6A-LR arise from both the inclusion of new permafrost processes and an explicit nitrogen cycle, leading to superimposed effects in the permafrost region, and to different carbon cycle dynamics in the tropics and mid-latitudes. Therefore, differences in cumulative emissions*

*and TCRE between both versions are not solely due to the inclusion of permafrost in IPSL-Perm-LandN, which prevents a direct assessment of the historical permafrost carbon-climate feedback."*

We have also added the following paragraph to the conclusion : *"Additionally, an emission-driven version of IPSL-Perm-LandN is under development and will enable the strength of the permafrost carbon-climate feedback to be properly assessed."*

**Specific comments**

- **Line 28: 'vegetation productivity in a negative feedback loop' – Please add a short explanation of this feedback, as this is otherwise unclear to the reader. Also, it seems that two mechanisms are mixed here: 1) increased veg carbon sink from CO2 fertilization that decreases the atmospheric CO2 concentration (and therefore offsets permafrost carbon loss), and 2) the provision of nutrients from permafrost thaw increases veg productivity. However, many models show that the inclusion of a nitrogen cycle rather leads to a limitation of nitrogen (at least in future simulations), suppressing vegetation productivity.**

We thank the reviewer for this remark. We have modified this paragraph into : *"On the other hand, increased $CO_2$ fertilisation from rising atmospheric $CO_2$ concentrations and longer growing seasons caused by warming could increase vegetation productivity in negative feedback loops, partially offsetting the positive climate feedback from warming-induced soil carbon losses (Gier et al., 2024; Arora et al., 2020). Nitrogen also impacts carbon cycle feedbacks in both directions. It can reduce vegetation productivity through nitrogen limitation (positive feedback, Street and Caldararu, 2022; Davies-Barnard et al., 2020), but can also increase plant carbon uptake through increased soil nitrogen availability due to soil warming and permafrost thaw (negative feedback Burke et al., 2022, Salmon et al., 2016; Finger et al., 2016; Koven et al., 2011). "*

- **Line 29: 'soil carbon losses' – rather the positive climate feedback from such soil carbon losses and therefore some of the subsequent loss that would occur in the absence of the negative feedback (Arora et al. 2020, Gier et al. 2024). Also, Gier et al. 2024 should be cited in appropriate places.**

We agree that this sentence lacked clarity. We have corrected it (see previous comment). We have also cited Gier et al. (2024) at l.59.

- **Lines 45-46: Should also cite Natali et al. (2021)**

We have added this citation.

- **Line 48: Should also cite de Vrese et al. (2023) and Steinert et al. (2021)**

We have added these citations.

- **Line 62: 'only half (six out of eleven)' – is this number based on the models used in Arora et al. (2020)? … because they don't use all available CMIP models (just those that provide specific 1pctCo2-bgc and -rad simulations).**

We agree this needs clarification. We consider that a CMIP6 model is an ESM if it includes an explicit representation of the land and ocean carbon cycles, following the IPCC AR6 (Table 5.4 of AR6 WGI

Chap.5). This also corresponds to the models used in Arora et al. (2020). We have clarified : *"only half of the CMIP6 ESMs representing the carbon cycle had an explicit representation of the nitrogen cycle (six out of the eleven ESMs from Arora et al. (2020))."*.

- **Line 71: 'coupled carbon and nitrogen cycles' – worth pointing out that these are vertically resolved**

We have modified the text accordingly : *"New developments include vertically resolved coupled carbon and nitrogen cycles..."*

- **Lines 79-80: Sentence starting with 'These new…' – Technically, this was possible with ORCHIDEE-v2, just not as good, right?**

This is correct as permafrost is defined as ground having a temperature lower or equal to 0°C for at least consecutive years, which was possible to simulate in ORCHIDEE-v2. However, many first-order permafrost processes/features were missing (latent heat of soil water phase change, soil insulation, vertically resolved soil biogeochemistry), leading to a large underestimation of the permafrost region by IPSL-CM6A-LR. We have modified this sentence : *"These new developments significantly improve the simulation of permafrost physics and carbon cycle dynamics in the IPSL ESM."*

- **Line 166: 'Soil moisture is resolved on a 11-layer scheme down to 2m' – Is this down to the bedrock level? And where is the bedrock level located?**

The bedrock level is not explicitly represented in ORCHIDEE-v3 as its position differs for soil thermics and hydrology. Hydrology is only resolved down to 2 m, which corresponds to bedrock, with a free drainage bottom boundary condition. However, soil thermics is resolved down to 90 m, therefore assuming a much deeper bedrock. Below 2 m, the calculation of thermal properties uses the water content of the deepest layer of the soil hydrology scheme. We have modified this paragraph : *"Soil moisture is resolved on a 11-layer scheme (the same as for soil thermics) down to 2m, where a free drainage bottom boundary condition is imposed (de Rosnay et al (2002)). Therefore, the bedrock differs between soil thermics (90m) and hydrology (2m). Note that these depths are flexible and can be easily changed by the user for specific configurations."*

- **Line 168: 'bottom boundary' – Does this refer to the 2m of soil moisture levels?**

This indeed refers to the 2m maximum depth of the soil hydraulic scheme. We have clarified this sentence (see previous comment).

- **Line 168: 'buffer' – also known as the zero-curtain effect.**

We assumed the reviewer refers to l.186. We have modified the text accordingly : *"It acts as a buffer - also called zero-curtain effect – absorbing..."*

- **Lines 227-230: I can see that these assumptions are sensible, but beyond that, are they supported by any literature? If yes, it would strengthen these assumptions by citing these.**

Chadburn et al. (2015) (in JULES) and Wu et al. (2016) (in CLASS v3.6) also included the thermal effect of moss in the first soil layer. We have added : *"This assumption is made in some land surface*

*models (Wu et al. (2016), Chadburn et al. (2015)), whereas some other models explicitly represent an organic layer on top of the soil column (Park et al. 2018, Porada et al. (2016)."*

The presence of moss and lichens is almost ubiquitous in Arctic ecosystems and can significantly alter soil thermal dynamics. This is why we chose to represent their effect in grid boxes containing Arctic PFTs. We have modified the sentence l.229 to justify this choice : *"We further assumed that the surface organic layer covers a fraction $f_{SOL}$ of each grid box containing boreal PFTs, as bryophytes are widespread in these ecosystems (Lewis et al. 2017, Barry et al. 2013)."*

- **Line 245: Please clarify what 'intensive nature' means here.**

We agree this needs clarification. We have modified this sentence : *"The approach for thermal conductivity is similar but takes into account that it is an intensive property (i.e. its value is independent of the size of the system)."*

- **Line 373: 'similar to' – I understand that these are identical not similar?**

This is correct. We have replaced "similar" by "identical".

- **Line 420: C4MIP has not been introduced. Also, does this mean, when considering CMIP models herein, this always refers to C4MIP models? (I see that it does in the results, would be good to specify this here).**

This is correct. We have added the following text at the beginning of the paragraph : *"IPSL-Perm-LandN is compared to ESMs from the Coupled Climate-Carbon Cycle Model Intercomparison Project (C4MIP, Jones et al., 2016), which are part of the broader CMIP6 ensemble (C4MIP models are listed in Arora et al., 2020). These models represent interactive land and ocean carbon cycle and can therefore represent carbon cycle feedbacks."*

- **Line 434: Should also cite Cai et al. (2020).**

We have added this citation.

- **Line 469: 'linear interpolation' – technically an exponential vertical interpolation would be more appropriate given that vertical heat distribution is assumed to be exponential, but I guess this has only minor impact on the estimated temperature at 3m?**

We thank the reviewer for this suggestion. We agree that an exponential interpolation would be more appropriate, even though the impact of the interpolation method remains limited at a 3m depth, as noted by the reviewer. We have changed the linear to an exponential interpolation method, taking the log of the temperature profile, interpolating linearly and taking the exponential of the interpolated value at 3m, as described in Koven et al. (2013). We have modified Fig.4 and the main text accordingly. We can confirm that both interpolation methods give very close results.

- **Line 493: Add that this refers to IPSL-Perm-LandN. Further clarifications are needed in Lines 582, 590 (see below), 593, 604 (see below), 617 (see below), 622 (see below), 682 (see below).**

We thank the reviewer for pointing out missing references to IPSL-Perm-LandN. We have clarified these lines.

- **Line 504: 'high warming comes mainly from the tropics' – Four sentences prior, it was said that the tropics are anomalously cold. Or does this refer to the warming rates? Or does this imply that over land the model warms more than expected, and therefore that relative absolute decreased temperatures appear over the ocean? (Though, further down the (global) ocean seems to be much warmer as well). Please revise this paragraph and more explicitly include which metric (land/ocean, global/regional, absolute/rate) you refer to.**

We thank the reviewer for this comment. The first part of the paragraph refers to absolute temperatures, which are underestimated by IPSL-Perm-LandN, especially in the tropics and mid-latitudes (Fig.1 (a) and (b)). The second part of the paragraph refers to the mean warming over the 2005-2014 period compared to 1850-1900. We have modified this paragraph and hope it is now clearer : *"Over the period 1940-2014, the mean annual land surface air temperature (SAT) of IPSL-Perm-LandN is about 1.5°C colder than the ERA5 reanalysis (Fig.1 (a)). During the last decade of the simulation (2005-2014), the mean land SAT of IPSL-Perm-LandN is 13.46±0.14°C while ERA5 has a warmer land SAT of 14.84°C. IPSL-Perm-LandN is consistently very close to IPSL-CM6A-LR as both share the same radiative scheme, and is at the lower bound of the C4MIP range, although the models generally tend to correctly simulate temperature changes (i.e. ΔSAT) rather than absolute temperatures. The cold land SAT bias in IPSL-Perm-LandN is mainly due to underestimated tropical and mid-latitude temperatures while the Arctic land SAT is closer to ERA5 estimates (Fig.1 (b)). Although the absolute land temperature is too cold, the land SAT anomaly relative to 1850-1900 is close to observations. Over land (emerged land excluding Greenland and Antarctica), IPSL-Perm-LandN has warmed by +1.60±0.14°C (mean 2005-2014 warming compared to 1850-1900) while the observations show a warming of +1.40°C for NOAAGlobalTemp and +1.16°C for HadCRUT (Fig.1 (c)). In contrast to the absolute temperature, the land SAT change compared to 1850-1900 is at the upper limit of the range of the C4MIP models. This relatively high warming mainly comes from the tropics and the Arctic where land SAT change (ref. 1850-1900) is overestimated compared to both NOAAGlobalTemp and HadCRUT (Fig.1 (d) and A5 (a))."*. We have also clarified the reference period in the legend of Fig.1 (d).

- **Lines 504-505: Sentence starting with 'In particular,…' – Could be moved to line 498 as the previous sentence implies that.**

We think the reviewer may have misunderstood this paragraph due to its lack of clarity (see previous comment). The sentence starting with "In particular…" refers to the Arctic amplification, and therefore to temperature change (or warming, compared to 1850-1900). The first part of the paragraph (until l.498) refers to the absolute mean land surface temperature. We think this sentence should remain in the second part of the paragraph assessing temperature changes, which has been clarified.

- **Lines 510-511: Repetition. Remove.**

These sentences have been removed.

- **Line 522: '0.16 mm.day-1' – would be helpful to also give this in percent.**

We have added *"...up to 0.16 mm.day$^{-1}$ (~6%)"*

- **Line 529: 'which is associated with the position of the North Atlantic drift' – As discussed further below, also because of a weaker AMOC. Might be worth to already mention here to connect these thoughts.**

We have completed this sentence : *"...which is associated with the position of the North Atlantic drift and due to a weaker AMOC than IPSL-CM6A-LR (Fig.3 (a) and Fig.3 (d))"*

- **Line 585: 'unsurprisingly' – better use 'expectedly'**

We have replaced "unsurprisingly" by "expectedly".

- **Line 590: I assume this is a general statement that applies to both model versions?**

We have specified *"for both IPSL-Perm-LandN and IPSL-CM6A-LR"*.

- **Lines 596-597: Sentence starting with ''GPP is…' – Is it because Arctic amplification is high? I presume not, since both models are quite similar in that regard, but they are not for GPP. So, is it all due to nitrogen fertilization? Or just tuning? (Is this what you mean by 'These differences' in the last sentence of this paragraph?**

We agree that the reasons for GPP overestimation in the Arctic and mid-latitudes should be given. This is due to 1) the inclusion of the nitrogen cycle in IPSL-Perm-LandN that replaces an empirical GPP downregulation in IPSL-CM6A-LR (limitation of Vcmax under increasing atmospheric CO2 to mimic nutrient limitation without explicitly representing it), 2) the combined effect of nitrogen fertilisation and climate warming, in particular in permafrost regions (this point will be detailed in a forthcoming paper), and 3) the fact that IPSL-Perm-LandN has not been as extensively calibrated as IPSL-CM6A-LR regarding GPP (although the GPP of ORCHIDEE-V3 has been calibrated and evaluated offline in Vuichard et al. 2019, tuning has been adjusted in IPSL-Perm-LandN to get realistic pre-industrial soil carbon stocks, see Sect.A4).

We have modified this paragraph to add a discussion and added a reference to the calibration procedure, detailed in Sect.A4 : *"GPP is overestimated in the Arctic and mid-latitudes compared to data-driven products, and within the observational range in the tropics (Fig.6 (b) and Fig.A14 (a)). This is likely due to IPSL-Perm-LandN simulating larger organic nitrogen stocks in the mid-latitudes and the Arctic than in the tropics, leading to higher mineralisation under warming, and therefore to greater sensitivity of nitrogen limitation to warming (Fig.A15). Compared to IPSL-CM6A-LR, GPP has largely increased in the Arctic (+3.3 PgC yr$^{-1}$) and mid-latitudes (+15.4 PgC yr$^{-1}$), and decreased in the tropics (-10.1 PgC yr$^{-1}$), resulting in an overall global increase of 8.6 PgC yr$^{-1}$. These differences are explained by the introduction of an explicit nitrogen cycle , which replaces an empirical GPP downregulation in IPSL-CM6A-LR (limitation of Vcmax under increasing atmospheric CO2 to mimic nutrient limitation without explicitly representing it), and to vertically-resolved soil biogeochemistry in IPSL-Perm-LandN. In addition, IPSL-CM6A-LR has been largely tuned using different data sources (FLUXNET, atmospheric CO2, NDVI, Peylin et al., 2016), while the new model including the nitrogen cycle has not been extensively calibrated (see Sect.A4)"*.

- **Lines 596-599: For both sentences, please add a discussion as to why this is/could be.**

We have modified this paragraph to add a discussion (see previous comment).

- **Line 604: 'the model' I assume this refers to IPSL-Perm-LandN? In the previous sentence you talk about IPSL-CM6A-LR.**

This is correct. We have replaced "the model" by "IPSL-Perm-LandN".

- **Line 606: 'These differences' – Which exactly? In the previous sentences you talk about similarities between the two model versions.**

We thank the reviewer for pointing out this inconsistency. This sentence refers to differences in global and regional GPP between IPSL-Perm-LandN and IPSL-CM6A-LR. We have moved it to the discussion l.596-599 (see previous comment).

- **Line 614: 'estimates' – which type of estimates? Observations, models? And why is this not in Figure 7 then?**

We thank the reviewer for pointing out this inconsistency. These estimates are the three observational products from Guenet et al. (2024). We have modified the text, Fig.7 and Fig.A14 to include the same observational products for consistency.

- **Line 617: 'The modeled RH…' – In IPSL-Perm-LandN I assume?**

This is correct. We have clarified *"The modeled RH for IPSL-Perm-LandN"*.

- **Line 620: 'reinforcing our confidence in these results' – I can't quite follow why this reinforces confidence in the results. In the previous section you have argued and shown that GPP and RH are well correlated, so both having the same bias compared to observations in not surprising and does not necessarily increase the robustness of the findings, depending on how large the offset to observations is. Please clarify.**

We agree that having similar spatial biases for GPP and RH was expected and that the robustness of the results depends on the magnitude of the bias. In addition, as global GPP is overestimated when compared to Jung RS and Jung RS-METEO, and global RH is within the range of observational products, the offset to observations is larger for GPP than RH. Therefore, we have modified this sentence : *"As expected, given their correlation, GPP and RH show the same regional biases when confronted with independent observational products."*

- **Lines 622-623: Worth mentioning that this sentences refers to model data and applies to both models**

We have specified *"For both models, the net…"*.

- **Lines 630-631: Sentence starting with 'This is…' – Does this mean modelled NBP is wrong by a factor of 2 because of that in all models?**

We acknowledge that the argument was too strong. Indeed, taking into account the river transport of carbon allows to bring closer together the estimates of the NBP simulated by land surface models to the net flux over land estimated by atmospheric inversions, as explained for instance in Ciais et al. (2021). The estimated pre-industrial river flux is around 0.65 PgC/yr (Regnier et al., 2022) with carbon transported from land to the ocean where a large fraction is emitted back to the atmosphere. This does not necessarily mean that the model NBP is two times smaller, as the CAMS inversion estimate is also

uncertain (especially the land - ocean flux partition). The 2023 GCB estimate for the mean of an ensemble of atmospheric inversions (including CAMS), after correction of the river flux, was 1.3 PgC/yr for the 2000s (between 0.7 and 2.0 PgC/yr; see Table 5 of Friedlingstein et al. 2023). Such mean estimate plus the pre-industrial river flux is thus closer to our estimate, although not exactly for the same period. This indicates that the CAMS flux estimate has a relatively large land sink (higher than the other inversions) and that the inversion estimates remain quite uncertain. In general, Land Surface Models do not simulate the pre-industrial land carbon sink corresponding to the pre-industrial river flux, as they are usually brought to equilibrium, following the TRENDY protocol (Sitch et al., 2024). Given that they lack the pre-industrial land sink, we can thus estimate that these models may underestimate the total land carbon uptake, although the uncertainties associated with these fluxes remain large. We have slightly modified the text to better account for the large uncertainties still associated with these numbers and the CAMS land sink estimate. The new text now reads: **"*This is due to the fact that atmospheric inversions account for lateral carbon fluxes (between the land and the ocean), whereas land surface models (and hence ESMs) typically do not model this flux and have a near-zero land-atmosphere carbon flux in the pre-industrial period. In contrast, the global pre-industrial river flux is estimated to be around 0.65 PgC yr$^{-1}$ (Regnier et al, 2022). Subtracting the contribution of lateral fluxes from the inversions generally helps to reconcile both approaches, leading to more comparable NBP values (Ciais et al, 2021). However, there is still significant uncertainty in these estimates and the 2023 CAMS estimate has a relatively large land sink (Friedlingstein et al, 2023)***".

- **Figures 8 and 10: Would be helpful to have a zero reference line in panel (a) in both figures.**

We have added the zero reference line in both figures.

- **Figures 8 and 10 captions: Worth mentioning that positive means land uptake in both figures.**

We have added : *"Positive (resp. negative) values correspond to a land carbon sink (resp. a source)."*

- **Line 682: Specify that this refers to IPSL-Perm-LandN. Also, 'historical period' –This is true for IPSL-CM6A-LR for the last decade but not the whole historical period, which can be misunderstood as such from the first half sentence.**

We agree that this needs clarification. We have added *"For IPSL-Perm-LandN, the permafrost region…"* and the period of the NBP quantification for IPSL-Perm-LandN in the following sentence : *"...than IPSL-CM6A-LR (0.24±0.04 PgC/yr for 2005-2014)"*. We hope this avoids the confusion with the temporal dynamics of IPSL-CM6A-LR.

- **Line 686: 'permafrost region may differ between models' – and even more so because their representation of permafrost and soil carbon differs substantially?!**

We thank the reviewer for highlighting this missing information. We have modified this statement : *"C4MIP models are divided into three groups and show a wide spread due to their differing representations of permafrost and soil carbon processes, as well as due to variations in the permafrost region between models."*

- **Line 699: 'medium SOC pool' – Weren't these introduced as active, slow and passive in Section 2.4.1? Where does the medium pool come from?**

This is a typo. The medium soil pool refers to CMIP6 nomenclature and corresponds to the slow pool in ORCHIDEE-v3. We have replaced "medium" by "slow". We have also modified similar inconsistencies in the rest of the text as well as the labels of figure A.16, A.19 and A.22.

- **Lines 732-733: subsentence in brackets – Would you therefore expect IPSL-CM6A-LR to drift away from what looks to be a very good agreement with GCB considering longer timescales (e.g. longer spin-up or future simulations)?**

At the timescale of the historical simulation (165 years), we only expect a small drift given that IPSL-CM6A-LR was run long-enough in pre-industrial conditions to ensure a relatively stable equilibrium. Such a drift could occur under much longer timescales (>1000 years), driven by ocean circulation.

- **Section 4.7: I am missing some discussion on the implications of the differences in the compatible CO2 emissions found between the two model versions. What does it tell us when, as in this case, cumulative compatible CO2 emissions are lower in IPSL-Perm-LandN? And what does this tell us about implications for the permafrost carbon-climate feedback?**

We have added the following discussion to Section 4.7 : " *Cumulative emissions are lower for IPSL-Perm-LandN than for IPSL-CM6A-LR (446 PgC) due to a lower historical total (land+ocean) carbon sink*" and "*In principle, the difference in cumulative compatible emissions (EgC) between IPSL-Perm-LandN and IPSL-CM6A-LR could be multiplied by the transient climate response to cumulative emissions (TCRE, °C.EgC$^{-1}$) of IPSL-Perm-LandN to infer the strength of the permafrost carbon-climate feedback (ΔT, °C) (assuming negligible change in the carbon-concentration feedback from the inclusion of permafrost). However, differences between IPSL-Perm-LandN and IPSL-CM6A-LR arise from both the inclusion of new permafrost processes and an explicit nitrogen cycle, leading to superimposed effects in the permafrost region, and to different carbon cycle dynamics in the tropics and mid-latitudes. Therefore, differences in cumulative emissions and TCRE between both versions are not solely due to the inclusion of permafrost in IPSL-Perm-LandN, which prevents a direct assessment of the historical permafrost carbon-climate feedback.*"

- **Lines 742-743: Sentence starting with 'They are…' – Wouldn't you expect that, given that those are concentration-driven experiments?**

This result was indeed expected for concentration-driven experiments that do not account for carbon cycle feedbacks. However, the close agreement between IPSL-Perm-LandN and GCB suggests that the model simulates the overall carbon sink (land + ocean) accurately. This is not the case for all C4MIP models over the 2005-2014 period, some of which underestimating the total carbon sink, and therefore compatible emissions (see Fig.11 (c)). We have modified this sentence to emphasize the role of carbon sinks for compatible emissions : "*For both models, they are very close to the fossil fuel emissions diagnosed by the Global Carbon Budget 2023 from different emission datasets, suggesting a relatively accurate simulation of the historical total (land+ocean) carbon sink, except for the simulated plateau in the 1940s.*"

- **Line 745: '(Rubino et al., 2013)' – Not sure I am seeing the stabilization in the reference. Can you indicate the figure here? Also, what alternative role do volcanic eruptions play here?**

The flattening of atmospheric CO2 concentrations from Law Dome measurement is shown in Fig.6 (b) of Rubino et al. (2013). Fig.1 from Bastos et al. (2016) shows a similar stabilisation using the same data. We have indicated the figure and added a reference to Bastos et al. (2016). We are not aware of any variability in volcanic eruptions that could explain this plateau.

- **Line 746: 'The models' – I assume this generally refers to CMIP models? Is this a consistent feature of all models?**

We thank the reviewer for highlighting the lack of clarity of this paragraph. We have clarified it : *"This plateau is due to the stabilisation of the atmospheric $CO_2$ concentration during this period (Bastos et al. (2016), Fig.1 and Rubino et al. (2013), Fig.6 (b)), which led to a decrease of $G_{ATM}$, causing a stagnation of $E_{FF}$ in concentration-driven C4MIP models. However no such stagnation is observed in $E_{FF}$ estimates from GCB, suggesting that a concomitant increase in carbon sinks occurred during this period (Liddicoat et al., 2021). No C4MIP model represents such an increase, and the dynamics of carbon sinks in this period is still not fully understood (Liddicoat et al., 2021; Bastos et al., 2016)."*.

- **Line 754: '21$^{st}$ century' – specify this refers only to up to 2014.**

We have specified *"(up to 2014)"*.

**Technical corrections**

- **Line 10: remove 'a' in front of 'depth-dependent decomposition dynamics'**
- **Line 103: 'used is' --> 'used in'**
- **Lines 117-118: 'ocean physics is … and is …' – should be 'are' in both cases, I think.**
- **Line 518: 'represented on' --> 'represented in'**
- **Figure 6 caption, line 3: 'observations' --> 'observational'**
- **Line 637: 'mid-high' --> 'mid-to-high'**

These technical issues have been corrected in the new version of the manuscript.

**References**

Arora et al.: Carbon–concentration and carbon–climate feedbacks in CMIP6 models and their comparison to CMIP5 models, Biogeosciences, 17, 4173–4222, https://doi.org/10.5194/bg-17-4173-2020, 2020.

Bastos et al.: Re-evaluating the 1940s $CO_2$ plateau, Biogeosciences, 13, 4877–4897, https://doi.org/10.5194/bg-13-4877-2016, 2016.

Chadburn et al. : An improved representation of physical permafrost dynamics in the JULES land-surface model, Geosci. Model Dev., 8, 1493–1508, https://doi.org/10.5194/gmd-8-1493-2015, 2015.

Ciais et al. Empirical estimates of regional carbon budgets imply reduced global soil heterotrophic respiration, National Science Review, Volume 8, Issue 2, February 2021, nwaa145, https://doi.org/10.1093/nsr/nwaa145

Intergovernmental Panel on Climate Change (IPCC). Climate Change 2021 – The Physical Science Basis: Working Group I Contribution to the Sixth Assessment Report of the Intergovernmental Panel on Climate Change. Chap.5 : Global Carbon and Other Biogeochemical Cycles and Feedbacks. Cambridge: Cambridge University Press, 2023.

Koven et al., 2013: Analysis of Permafrost Thermal Dynamics and Response to Climate Change in the CMIP5 Earth System Models. J. Climate, **26**, 1877–1900, https://doi.org/10.1175/JCLI-D-12-00228.1.

Lardy, R., Bellocchi G., and Soussana J. F.: A new method to determine soil organic carbon equilibrium, Environ. Modell. Softw., 26, 1759–1763, 2011.

Regnier et al., The land-to-ocean loops of the global carbon cycle. Nature. 2022 Mar;603(7901):401-410. doi: 10.1038/s41586-021-04339-9. Epub 2022 Mar 16. PMID: 35296840.

Rubino, M., et al. (2013), A revised 1000 year atmospheric $\delta^{13}$C-$CO_2$ record from Law Dome and South Pole, Antarctica, J. Geophys. Res. Atmos., 118, 8482–8499, doi:10.1002/jgrd.50668.

Sitch, et al, (2003), Evaluation of ecosystem dynamics, plant geography and terrestrial carbon cycling in the LPJ dynamic global vegetation model. Global Change Biology, 9: 161-185. https://doi.org/10.1046/j.1365-2486.2003.00569.x

Vuichard et al.: Accounting for carbon and nitrogen interactions in the global terrestrial ecosystem model ORCHIDEE (trunk version, rev 4999): multi-scale evaluation of gross primary production, Geosci. Model Dev., 12, 4751–4779, https://doi.org/10.5194/gmd-12-4751-2019, 2019.

Wu et al.: Integrating peatlands into the coupled Canadian Land Surface Scheme (CLASS) v3.6 and the Canadian Terrestrial Ecosystem Model (CTEM) v2.0, Geosci. Model Dev., 9, 2639–2663, https://doi.org/10.5194/gmd-9-2639-2016, 2016.

**Anonymous referee #2**

**This manuscript describes the Earth System Model IPSL-Perm-LandN, a development based on IPSL's CMIP6 model IPSL-CM6A-LR. Model improvements discussed here are based on a better physical representation of permafrost as well as an incorporation of vertically resolved soil carbon and an inclusion of soil nitrogen. A better representation of the terrestrial carbon cycle in general and permafrost carbon in particular in ESMs is becoming increasingly important especially since the CMIP framework is moving towards emission driven simulations instead of concentration driven simulations, and the permafrost carbon representation has been identified as a major source of uncertainty in climate projections. This study provides very relevant and important results on improvements mad in the IPSL's contribution to CMIP and addresses important gaps in the CMIP6 contribution.**

We thank the reviewer for pointing out the relevance of our effort to include permafrost processes in a CMIP ESM.

**The manuscript is very well written, and results are presented in a clear and understandable manner. It addresses aspects of overall model performance, including atmosphere and ocean variables, as the model is discussed as a whole, even though improvements were restricted to the land model. The introduction would profit from a slightly broader view at the importance of representing the terrestrial carbon cycle under emission driven simulations, and the discussion would profit from a broader look at limitations of the chosen modelling set up.**

We thank the reviewer for this positive assessment. We have added the following lines to the introduction : *"Reducing the uncertainties surrounding permafrost carbon cycle feedbacks is becoming especially important as ESMs move towards emission-driven simulations, in which the atmospheric CO2 concentrations will be largely determined by the simulated carbon cycle dynamics (Steinert. and Sanderson (2025), Park et al (2025), Sanderson et al (2024)). Such emission-driven simulations are particularly relevant for producing policy-oriented climate projections and for properly accounting for feedbacks between the carbon cycle and climate."*

We have also added a new section "Limitations of IPSL-Perm-LandN in simulating permafrost ecosystems" before the conclusion (Section 4.8).

**I would like to echo one of the questions of reviewer 1 here: Will this be the model set up for IPSL's contribution to CMIP7? If so, this should be emphasized in the conclusion section! Quite a number of LSMs that were used within the CMIP6 ESMs have had capabilities to represent land carbon processes better than it was done in CMIP6, but they were not included in the model set ups used for the simulations. A lot of LSM groups have made considerable improvements to their models since CMIP6, but to what degree these improvements will be part of the CMIP7 efforts remains to be seen, so if this model is IPSL's contribution to CMIP7, that would be a nice signal to the ESM community.**

Most of the modifications of IPSL-Perm-LandN will be included in the version of the IPSL ESM for CMIP7 Fast Track (CMIP7-FT), including the latent heat of soil water phase change, soil insulation by

soil carbon and surface organic layers, and an explicit nitrogen cycle (which is a significant step forward regarding the representation of land biogeochemistry in the IPSL ESM). However, the vertically-resolved soil biogeochemistry scheme might not be included because it requires a long spin-up phase that is not compatible with the relatively short deadline of the CMIP7-FT simulations (to be completed in 2026). Some developments are ongoing to solve this difficulty and vertically-resolved soil biogeochemistry should be included for the more general phase of CMIP7. The following lines have been added to the conclusion: *"Most of the new permafrost processes described in this study will be integrated into the IPSL ESM for CMIP7 Fast Track (CMIP7-FT), including the latent heat of soil water phase change, soil insulation by soil carbon and surface organic layers, and the explicit nitrogen cycle. The vertically-resolved soil biogeochemistry will likely only be included for the broader CMIP7 phase, due to the long spinup required and the time constraints of CMIP7-FT."*

**I have one general comment concerning the results and discussion section with regard to permafrost dynamics and permafrost carbon. You describe a number of choices in the moel description that probably have technical reasons, but don't seem straight forward. Their implications should be discussed in the results and discussion section.**

We thank the reviewer for this comment. We have added a new section "Limitations of IPSL-Perm-LandN in simulating permafrost ecosystems" before the conclusion (Section 4.8) to discuss the choices made in the model, the associated hypotheses and their implications and limitations. The next comments have been included in this section.

- **I assume there are technical reasons why there is only a 2m soil column for soil water, and that you wanted to do carbon and nitrogen pools for greater depth, but what are the consequences on latent heat and heat flux in general that soil moisture is only computed above 2m depths? ALT can be considerably deeper than 2m.**

We thank the reviewer for this comment. The restriction of soil hydrology to the upper 2 m of soil is a known technical limitation of ORCHIDEE-v3. Below 2 m, the calculation of soil thermal properties (and therefore of latent heat which is included in the apparent heat capacity) uses the soil water content of the deepest soil layer (1.5 – 2 m). This means that below 2 m, changes in the hydrology of the 1.5-2 m layer directly impact the soil thermal dynamics, whereas in the real world, soil water would have its own dynamics in these deeper layers. This can lead to unrealistically fast changes in soil thermal properties associated with changes of the water content of the 1.5 – 2 m, further impacting heat fluxes. This limitation obviously has an impact on ALT, especially in warmer permafrost regions where it can be greater than 2 m. It is probably one of the reasons for the relatively sharp transition in ALT on permafrost edges and for its overestimation in southern permafrost margins. Work is ongoing to have a spatially variable soil water depth in the model but it still requires substantial effort and calibration.

- **Why is there not representation of moss and lichens as PFTs? And what processes are missing because you only consider their physical properties, but no biology?**

The absence of moss and lichens PFT has several limitations. First, we do not represent the water fluxes associated with non vascular vegetation. In particular, mosses can act as sponges and absorb water, therefore regulating soil moisture (Turetsky et al. (2012)). Second, mosses are responsible for a large fraction of above-ground NPP (20% on average in boreal forests), and can take up carbon even

under very low temperatures, which is also missed by IPSL-Perm-LandN (Turetsky et al. (2010)). Some moss species can also fix atmospheric nitrogen and provide a source of nutrient for other plants, especially in generally nitrogen-limited boreal forests (Markham (2009)). Finally, mosses favor the development of peatlands through their influence on soil temperature and water content, and the slow burial of organic matter. These peatland ecosystems are not represented by IPSL-Perm-LandN.

A first attempt has been done with the inclusion of "moss and lichens" as a new PFT (for carbon, water and energy cycles) in a previous version of ORCHIDEE, together with a boreal shrub PFT (Druel et al., 2019). However, a separate PFT for moss and lichens is sub-optimal as they also constitute the understorey of forest ecosystems. A more comprehensive representation of moss and lichens thus requires mixed PFTs that can overlap horizontally with mosses and lichens being nearly ubiquitous in all arctic PFTs. Such functionality requires large structural changes that are progressively being integrated into the ORCHIDEE model, first with a more detailed description of the vertical structure of the ecosystems using a multi-layers energy budget (Ryder et al. 2016). In this context, the ORCHIDEE group has not integrated the initial development of Druel et al. (2029) for "moss and lichens" (done in a previous branch without the N cycle) into the main version of ORCHIDEE with the nitrogen cycle, but works on a more comprehensive approach.

- **The soil column has 18 layers extending to 90m depth, and soil moisture is only done for 11 layers in the upper 2m. You assume a soil moisture below that, how does that effect decomposition, and are these deep layers even relevant? Soil carbon is often assumed to be limited to around the upper 3m of the soil column, what do your carbon and nitrogen pools in the deepest layer look like? And I assume that the soil moisture assumption for layers below 2m is only used for calculating decomposition, so mass is conserved, but does that make any sense in 90m depth? What are the consequences of these assumptions for your top layer carbon and nitrogen pools?**

We thank the reviewer for this comment. Similarly to the calculation of soil thermal properties, below 2 m, the SOC decomposition function uses soil moisture from the deepest hydrological layer (1.5 – 2 m). In theory, this could result in overly fast changes to the decomposition function in deeper layers. However, the impact on soil carbon and nitrogen remains limited, since almost all of these stocks are located within the top 3 - 3.5 m (see Fig.A19). We agree that calculating soil carbon and nitrogen stocks for the entire 90 m column is irrelevant, and indeed almost no organic matter is found below 3.5 m in IPSL-Perm-LandN (deep deposits such as Yedoma are not represented).

**I have some more minor specific comments and questions to the authors listed below.**

**Specific Comments:**

**Lines 29-32: The permafrost-carbon feedback overall remains a major source of uncertainty, which is especially important in the light of the move towards emission driven simulations, where the actual $CO_2$ concentrations in the atmosphere will to a large degree be determined by ESM's own carbon cycles, which should be included here. See eg Steinert, N. J. and Sanderson, B. M.: Normalizing the permafrost carbon feedback contribution to the Transient Climate Response to Cumulative Carbon Emissions and the Zero Emissions Commitment, Earth Syst. Dynam., 16, 1711–1721, https://doi.org/10.5194/esd-16-1711-2025, 2025 and Sanderson, B. M., Booth, B. B. B.,**

Dunne, J., Eyring, V., Fisher, R. A., Friedlingstein, P., Gidden, M. J., Hajima, T., Jones, C. D., Jones, C. G., King, A., Koven, C. D., Lawrence, D. M., Lowe, J., Mengis, N., Peters, G. P., Rogelj, J., Smith, C., Snyder, A. C., Simpson, I. R., Swann, A. L. S., Tebaldi, C., Ilyina, T., Schleussner, C.-F., Séférian, R., Samset, B. H., van Vuuren, D., and Zaehle, S.: The need for carbon-emissions-driven climate projections in CMIP7, Geosci. Model Dev., 17, 8141–8172, https://doi.org/10.5194/gmd-17-8141-2024, 2024.

We thank the reviewer for this suggestion. We have added the following sentence later in the introduction on line 46, after ESMs have been introduced : *"Reducing the uncertainties surrounding permafrost carbon cycle feedbacks is becoming especially important as ESMs move towards emission-driven simulations, in which the atmospheric CO2 concentrations will be largely determined by the simulated carbon cycle dynamics (Steinert and Sanderson, 2025; Park et al., 2025; Sanderson et al., 2024)."*

**Line 44-46: There is an overview paper on the permafrost representation of the CMIP6 ESMs on a more general level than in Arora et al that should be mentioned here, Matthes, H., Damseaux, A., Westermann, S., Beer, C., Boone, A., Burke, E., ... & Wieder, W. R. (2025). Advances in Permafrost Representation: Biophysical Processes in Earth System Models and the Role of Offline Models. Permafrost and Periglacial Processes, 36(2), 302-318.**

We have added this reference.

**Line 56-59: The lack of resolving vertical soil carbon also prevents representation of abrupt permafrost thawing through eg fire or thaw slumps, which is estimated to be a major source of permafrost carbon loss (Turetsky, M.R., Abbott, B.W., Jones, M.C. et al. Carbon release through abrupt permafrost thaw. Nat. Geosci. 13, 138–143 (2020). https://doi.org/10.1038/s41561-019-0526-0), even though most LSMs in ESMs miss representation of the underlying processes as well. It would still be good to mention this here, since including vertically resolved soil carbon can be seen as creating a basis for inclusion of at least some of these fast processes.**

We have modified this sentence into : *"The lack of such a vertical soil carbon discretisation prevents most models from representing the large soil carbon content of the permafrost region as well as the effect of gradual and abrupt (e.g. through fire or thaw slumps) permafrost thaw on soil carbon dynamics and the permafrost carbon-climate feedback (Schädel et al., 2024; Gier et al., 2024; Varney et al., 2022; Turetsky et al., 2020)."*. We have also mentioned the absence of representation of abrupt thaw processes in IPSL-Perm-LandN and its implications in the new section 4.8 : *"IPSL-Perm-LandN only represents gradual thaw and misses abrupt thaw processes that could be a major source of permafrost carbon loss in the future (Turetsky et al., 2020). Such processes would necessitate the inclusion of excess ice and soil subsidence in IPSL-Perm-LandN, and could be inspired by developments made in CLM (Cai et al., 2020; Lee et al., 2014)"*.

**Line 179: Here, and everywhere else you show units, you separate them by a period. While that increases readability, I don't think it's standard notation. It should probably be changed.**

We have changed this in the whole paper.

**Line 305ff: Maybe I misunderstand what you are describing here. There is one mineral nitrogen pool, which is associated with the surface layer. If that pool is not sufficient, nitrogen is taken from the atmosphere. Nitrogen pools however are vertically resolved. So is mineralized nitrogen available to all soil layers out of the pool in the surface layer? And can nitrogen pools in all layers draw from the atmosphere through that mechanism as well? And if so, what consequences does that have?**

We thank the reviewer for this comment and agree this section should be clarified. Nitrogen uptake from the atmosphere only occurs when the organic matter decomposition (driven by eq.19) is limited by nitrogen (i.e. there is not enough mineral nitrogen available for the immobilisation associated with decomposition), which can be the case for litter decomposition. In that case, we choose not to limit decomposition and nitrogen is taken up from the atmosphere to match C:N ratios of receiving pools. This can artificially increase the soil nitrogen concentration of all layers in which this process occurs. However, this remains limited and generally only happens for surface layers during litter decomposition, which avoids the artificial increase of deeper organic nitrogen pools.

In addition, organic carbon and nitrogen are vertically discretised but this is not the case of the mineral nitrogen pool which is calculated using a bucket-like approach, and is available to all soil layers (see line 329-330). Therefore, all organic nitrogen pools can immobilise mineral nitrogen, regardless of their depth. This is a limitation of IPSL-Perm-LandN that is discussed in the new section 7.8 : *"In IPSL-Perm-LandN, soil organic carbon and nitrogen are vertically resolved but mineral nitrogen is not. Therefore, vegetation can access mineral nitrogen released throughout the soil column, regardless of the depth at which the release occurs. This will impact the future response of the permafrost carbon cycle, as deep nitrogen released at the thaw front will be made directly available for vegetation, possibly leading to overestimated plant nitrogen uptake and productivity."*.

We have also clarified this at line 330 *: "Soil mineral nitrogen, however, is not vertically resolved and remains represented on a single soil layer in each grid box. It can exchange nitrogen with all the organic nitrogen layers through mineralisation or immobilisation."*

**Line 355: Does that definition of the permafrost area used here mean that the area for which cryoturbation is calculated is dynamic?**

We assumed that the review refers to l.335. The area for which cryoturbation is calculated is indeed dynamic. This ensures that cryoturbation only happens in grid boxes where the simulated ALT is lower or equal to 3m, which is especially important in high-emission scenarios where the permafrost drastically decreases. We have added this precision on l.336 : *"Thus, the permafrost region where cryoturbation occurs is dynamic."*

**Line 360ff: If the SoilGrids data has vertical layers until 2m depth, how do you compute values for your layers below 2m? There is no information to interpolate, you could only extrapolate, and how does that make sense when there is hardly any carbon stored below 3m depth in most of the permafrost area?**

SOC and SON stocks below 2m have been initialised to zero to avoid an uncertain extrapolation. However, soil carbon can then be buried into deeper layers by bio- and cryoturbation. We have added a sentence in line 362 : *"Below 2m, initial organic carbon and nitrogen have been set to zero."*

**Line 369: This is more of a comment: The equilibrium approach is the one everybody uses, but even in 1850, the carbon pools were not in equilibrium.**

We thank the reviewer for pointing this out. Setting equilibrium as a target after spinup is indeed not relevant for all carbon pools/regions. We have mentioned this in line 369 : *"Before running IPSL-Perm-LandN under varying forcings, it is necessary to bring the carbon and nitrogen pools into equilibrium, although such a target is questionable given that parts of the carbon cycle were not at equilibrium in 1850 (e.g. permafrost soils and peatlands were accumulating carbon, Schuur et al., 2022)"*.

**Line 500: While the realistic warming rates are good, when looking at permafrost dynamics and thawing related carbon release, what matters at the end are absolute temperatures, since freeze/thaw processes are coupled to an absolute temperature. From your latitudinal plot, it does seem that the underestimation is mainly originating from the tropics, but I still wonder what implications are, and if there is a seasonality to the bias that could impact permafrost?**

We thank the reviewer for this comment. We have added a figure to the appendix (Fig.A5 in updated manuscript) to show the seasonal cycle of land surface air temperature (SAT) in the Arctic, mid-latitudes and the tropics for 2005-2014. The underestimation of global land SAT mainly comes from a systematic underestimation of land SAT in the tropics and mid-latitudes across all seasons. In the Arctic however, there is a cold bias in spring and a warm bias in autumn that partially cancel when looking at annual mean land SAT. Fig. A12 shows that for almost every season, cold and warm biases exist in the Arctic (compared to ERA5), although we clearly see the overall cold bias in May-June and the warm bias in October-November. These biases could impact the thermal state of permafrost, but are also impacted by the heat sink due to water thawing in spring and the heat source due to refreezing in autumn. We have modified the text on line 497 to mention this point : *"The cold land SAT bias in IPSL-Perm-LandN is mainly due to underestimated tropical and mid-latitude temperatures across all seasons while the Arctic land SAT is closer to ERA5 estimates, due to canceling cold and warm biases in spring and autumn, respectively (Fig.1 (b) and Fig.A5). These biases could impact permafrost freeze and thaw but are unevenly distributed across the region (Fig.A6)."*

**Line 520: Does the overestimation of snow fall actually lead to an overestimation of snow cover? And what about the seasonality? Could that impact your permafrost dynamics?**

We thank the review for this comment. We have compared the simulated snow cover to the ESA-CCI CryoClim observational product. There is no important bias in annual mean snow cover but an underestimation of snow cover in spring and autumn in the permafrost region. We have added the following lines at l.521 : *"This slight overestimation of Arctic snowfall does not lead to significant snow cover biases (Fig.A.10). However, snow cover is underestimated by 10 to 20% in the permafrost region in April-May and October-November, which could lead to reduced ground insulation and faster thawing and refreezing of permafrost in spring and autumn."*. We have also added a new figure showing the annual mean and monthly biases in snow cover in Appendix (new Fig.A10).

**Line 577: The overestimation of the southern boundary of permafrost might be associated with the underestimated surface air temperatures, depending on the seasonality of that bias.**

From Fig.A12, the (slightly) overestimated permafrost southern boundary is not obviously related to late summer temperature biases. This could be a legacy effect of the spring cold bias but remains uncertain. We have modified line 576-577 into : " *In Eurasia, the permafrost region compares well with the 50% permafrost contour from ESA-CCI observations, with a slight overestimation over the southern boundary, which could be due to a spring cold bias in this region (Fig.A6).* "

**Line 578ff: Is the underestimation of permafrost at the southern edge of the Canadian permafrost associated with maybe a spatial pattern in air temperature?**

We do not see any clear warm bias at the southern edge of the Canadian permafrost (except maybe November) that could explain such an underestimation of the permafrost region (Fig.A12). We have modified line 578 into *"In North America, simulated permafrost in IPSL-Perm-LandN is present in the north, but is absent at the southern edge, in Canada, which is not clearly related to a warm temperature bias (Fig.A6) but is a known bias in many CMIP6 models (Burke et al., 2020)."*.

***Line 603: I acknowledge that attribution of effects like this are difficult, but do you have any ideas why improving the terrestrial carbon cycle representation lead to a decrease in performance when it comes to GPP?***

We agree that the reasons for GPP overestimation in the Arctic and mid-latitudes should be given. This is due to 1) the inclusion of the nitrogen cycle in IPSL-Perm-LandN that replaces an empirical GPP downregulation in IPSL-CM6A-LR (limitation of Vcmax under increasing atmospheric $CO_2$ to mimic nutrient limitation without explicitly representing it), 2) the combined effect of nitrogen fertilisation and climate warming, in particular in permafrost regions (this point will be detailed in a forthcoming paper), and 3) the fact that IPSL-Perm-LandN has not been as extensively calibrated as IPSL-CM6A-LR regarding GPP (although the GPP of ORCHIDEE-V3 has been calibrated and evaluated offline in Vuichard et al. 2019, tuning has been adjusted in IPSL-Perm-LandN to get realistic pre-industrial soil carbon stocks, see Sect.A4).

We have modified this paragraph to add a discussion and added a reference to the calibration procedure, detailed in Sect.A4 : *"GPP is overestimated in the Arctic and mid-latitudes compared to data-driven products, and within the observational range in the tropics (Fig.6 (b) and Fig.A14 (a)). This is likely due to IPSL-Perm-LandN simulating larger organic nitrogen stocks in the mid-latitudes and the Arctic than in the tropics, leading to higher mineralisation under warming, and therefore to greater sensitivity of nitrogen limitation to warming (Fig.A15). Compared to IPSL-CM6A-LR, GPP has largely increased in the Arctic (+3.3 PgC yr-1) and mid-latitudes (+15.4 PgC yr-1), and decreased in the tropics (-10.1 PgC yr-1), resulting in an overall global increase of 8.6 PgC yr-1. These differences are explained by the introduction of an explicit nitrogen cycle , which replaces an empirical GPP downregulation in IPSL-CM6A-LR (limitation of Vcmax under increasing atmospheric $CO_2$ to mimic nutrient limitation without explicitly representing it), and to vertically-resolved soil biogeochemistry in IPSL-Perm-LandN. In addition, IPSL-CM6A-LR has been largely tuned using different data sources (FLUXNET, atmospheric $CO_2$, NDVI, Peylin et al., 2016), while the new model including the nitrogen cycle has not been extensively calibrated (see Sect.A4)".*

**References**

Druel, A., Ciais, P., Krinner, G., & Peylin, P. (2019). Modeling the vegetation dynamics of northern shrubs and mosses in the ORCHIDEE land surface model. Journal of Advances in Modeling Earth Systems, 11(7), 2020-2035.

Markham, J.H. Variation in moss-associated nitrogen fixation in boreal forest stands. Oecologia 161, 353–359 (2009). https://doi.org/10.1007/s00442-009-1391-0

Ryder, J., Polcher, J., Peylin, P., Ottlé, C., Chen, Y., van Gorsel, E., Haverd, V., McGrath, M. J., Naudts, K., Otto, J., Valade, A., and Luyssaert, S.: A multi-layer land surface energy budget model for implicit coupling with global atmospheric simulations, Geosci. Model Dev., 9, 223–245, https://doi.org/10.5194/gmd-9-223-2016, 2016.

Turetsky et al. (2012), The resilience and functional role of moss in boreal and arctic ecosystems. New Phytologist, 196: 49-67. https://doi.org/10.1111/j.1469-8137.2012.04254.x

Turetsky et al. 2010. The role of mosses in ecosystem succession and function in Alaska's boreal forest. Canadian Journal of Forest Research. 40(7): 1237-1264. https://doi.org/10.1139/X10-072

Vuichard et al.: Accounting for carbon and nitrogen interactions in the global terrestrial ecosystem model ORCHIDEE (trunk version, rev 4999): multi-scale evaluation of gross primary production, Geosci. Model Dev., 12, 4751–4779, https://doi.org/10.5194/gmd-12-4751-2019, 2019.